# Deep mutational scanning of the human insulin receptor ectodomain to inform precision therapy for insulin resistance

Vahid Aslanzadeh[1,2], Gemma V. Brierley[3], Rupa Kumar[4], Hasan Çubuk[2], Corinne Vigouroux [5,6], Kenneth A. Matreyek [7], Grzegorz Kudla [2] ✉ & Robert K. Semple [1,2,8] ✉

The insulin receptor entrains tissue growth and metabolism to nutritional conditions. Complete loss of function in humans leads to extreme insulin resistance and infantile mortality, while loss of 80-90% function permits longevity of decades. Even low-level activation of severely compromised receptors, for example by anti-receptor monoclonal antibodies, thus offers the potential for decisive clinical benefit. A barrier to genetic diagnosis and translational research is the increasing identification of variants of uncertain significance in the *INSR* gene, encoding the insulin receptor. By coupling saturation mutagenesis to flow-based assays, we stratified approximately 14,000 INSR extracellular missense variants by cell surface expression, insulin binding, and insulin- or monoclonal antibody-stimulated signalling. Resulting function scores correlate strongly with clinical syndromes, offer insights into dynamics of insulin binding, and reveal novel potential gain-of-function variants. This INSR sequence-function map has biochemical, diagnostic and translational utility, aiding rapid identification of variants amenable to activation by non-canonical INSR agonists.

Insulin promotes energy storage and tissue growth after food ingestion, and is essential for human life. Its effects are mediated by a ubiquitously expressed dimeric transmembrane receptor tyrosine kinase which autophosphorylates on insulin binding. This triggers a network of responses including activation of phosphoinositide 3-kinase, and phosphorylation and activation of downstream AKT kinases[1]. While structural studies have provided insights into the complex insulin binding kinetics and ensuing receptor activation[2,3], important structure-function questions remain, particularly relating to the conformation changes induced by insulin binding that activate trans-autophosphorylation. Receptor structures with multiple bound insulins have been described, however which are physiologically relevant remains to be fully determined[4–6].

More than 150 disease-causing mutations in the *INSR* gene, encoding the insulin receptor, are known[7,8]. Near complete loss of receptor function due to biallelic mutations causes Donohue syndrome, leading to mortality in early months of life, while in Rabson Mendenhall syndrome, another recessive syndrome, survival to adulthood is possible thanks to residual function[8]. Dominant INSR-related insulin resistance (IR), often called "Type A" IR, features up to 25% receptor function

[1]Institute for Neuroscience and Cardiovascular Research, University of Edinburgh, Edinburgh, UK. [2]MRC Human Genetics Unit, Institute for Genetics and Cancer, The University of Edinburgh, Edinburgh, UK. [3]Department of Comparative Biomedical Science, Royal Veterinary College, London, UK. [4]Medical Research Council (MRC) Metabolic Diseases Unit, Institute of Metabolic Science, University of Cambridge, Cambridge, UK. [5]National Reference Centre for Rare Diseases of Insulin Secretion and Insulin Sensitivity (PRISIS), Endocrinology Department, Saint–Antoine Hospital, Assistance Publique–Hôpitaux de Paris (AP-HP), Paris, France. [6]Inserm UMR_S 938, Saint–Antoine Research Centre, Sorbonne University, Paris, France. [7]School of Medicine, Case Western Reserve University, Cleveland, OH, USA. [8]National Severe Insulin Resistance Service, Cambridge University Hospitals NHS Foundation Trust, Hills Road, Cambridge CB2 0QQ, UK. ✉e-mail: Grzegorz.Kudla@ed.ac.uk; rsemple@ed.ac.uk

and is amenable to management with current therapies[8]. No clinical impact of 50% loss of receptor function is clearly documented.

Extracellular INSR mutations may change receptor abundance, localization, insulin binding, signalling, and/or recycling[9]. No single biochemical assay captures all these effects; many tens of INSR mutations have been studied using various in vitro and cellular approaches, but which assay best predicts clinical outcome is unclear. This is an acute problem as diagnostic testing commonly reveals novel "variants of unknown significance" (VUS), often in people of ethnicities poorly represented in large population datasets.

The initially steep relationship between INSR function and clinical outcome implies that even minor enhancement of signalling by severely impaired receptors could confer decisive benefit. One strategy is to use agonists with a mode of action distinct to that of insulin. Anti-receptor antibodies can activate wild-type receptor in cells[10] and in vivo[11,12], and sometimes also severely dysfunctional mutant receptors[13–15], exerting insulin mimetic metabolic actions in cells[13,14] and mice[14,16]. However, although this approach is validated for a small number of well-studied INSR mutations, the full repertoire of activatable mutations is unknown.

Multiplexed Assays of Variant Effect (MAVE) have emerged as a powerful approach in functional genomics[17]. By coupling comprehensive mutagenesis to pooled, sequencing-based assays, MAVE enable systematic evaluation of the functional consequences of many thousands of different amino acid changes simultaneously. The potential of this approach when applied to membrane-associated receptors is rapidly being demonstrated, having been used variously to delineate residues involved in ligand binding[18], to determine mechanisms of signal transduction[19,20], and to stratify pathogenic variants for potential targeted therapy[21].

We have now developed a MAVE by saturation mutagenesis of the extracellular, ligand-binding domain of the human insulin receptor, followed by massively parallel assays of variant effects on receptor expression, insulin and antibody binding, and insulin and antibody-stimulated signalling. This addresses the diagnostic challenge posed by VUS, will improve clinical prognostication, will enhance understanding of INSR genotype-function relationships, and improve stratification of loss-of-function (LoF) INSR mutations for novel agonist therapy.

## Results

### INSR mutation library creation and validation

We focused on the extracellular domain of the human INSR B isoform, which includes exon 11 and is regarded as the more "metabolic" INSR variant[22]. Two thirds of the full-length receptor are extracellular (Supplementary Fig. 1)[23], and this region harbours disease-related mutations compromising insulin binding and potentially activatable by novel ligands.

Using a plasmid-based approach we mutated residues 28–955, from the start of the first codon of the mature receptor to the start of the transmembrane domain. We subcloned the INSR gene into an entry plasmid with an interposed attB site to permit integration into a genomic landing pad and incorporated 30mer barcodes upstream from the open reading frame[24]. Mutations were then introduced into the previously barcoded INSR sequence by nicking mutagenesis based on a published method[25] to yield a mutagenised, barcoded library. Long read PacBio sequencing verified the library and determined pairing of barcodes and mutations (Fig. 1A, B; and Supplementary Fig. 2A-C). After removing barcodes with fewer than 150 reads across 4 bins, we observed 15,996 single missense and stop codon variants out of 18,560 possible (86% coverage), tagged by 80,956 unique barcodes. 81% of mutations associated with >1 barcode (Supplementary Fig. 2B). 35% of the barcodes aligned with unmutated INSR or synonymous mutations only, and were taken as wild-type (WT) in downstream analyses (Supplementary Fig. 2C, D, Supplementary Table 1).

Study of INSR variant function requires a background free of endogenous insulin receptor and the homologous insulin-like growth factor 1 (Igf1) receptor. We used a mouse embryo fibroblast model in which Igf1r knockout[26] is augmented by doxycycline-inducible, shRNA-mediated Insr knockdown[13]. We chose conditional knockdown as Insr knockout in this cell line reproducibly yielded cellular atypia, compromising viability. We lentivirally introduced a single copy of a published landing pad allowing efficient Bxb1-mediated integration of transfected plasmid[27]. Finally, the mutagenised, barcoded human INSR plasmid library was introduced to cells by electroporation (Fig. 1B). This generated a cellular system in which doxycycline simultaneously induced efficient endogenous mouse Insr knockdown and human INSR overexpression in an Igf1r null background (Supplementary Fig. 3), obviating concerns about clonal selection resulting from lethality in sustained culture of double Igf1r/Insr null cells.

### Design and validation of INSR functional assays

Having generated a conditionally expressible INSR variant library, we determined which mutations are expressed, which bind insulin, and which mediate insulin- or antibody-induced signalling by validated flow sorting assays (Fig. 1C, Supplementary Fig. 2E). To measure cell surface expression we used binding of two well characterised human INSR-specific monoclonal antibodies (mAbs 83-7 and 83-14) with distinct epitopes[28,29]. Both can bind mutated receptors, eliciting signalling even by some with severely attenuated insulin responsiveness[13]. To assess binding we used insulin or antibodies labelled with AlexaFluor 647 (AF647). Finally, to assess signalling induced by insulin or antibodies we probed permeabilised cells with an antibody against phosphorylated AKT[30] (Fig. 1C, Supplementary Fig. 2E).

Cells were flow-sorted into four bins per assay. From every bin DNA was extracted, barcodes were amplified, and high-throughput sequencing was used to quantify barcode frequency per bin (Fig. 1D). A weighted average was calculated for each barcode based on its distribution across bins, and a score calculated for each variant (Supplementary Data 1,2). Barcode score distributions for insulin binding and cell surface expression were superimposable and not significantly different for WT receptors and synonymous variants but skewed towards lower scores for missense variants (Fig. 1E). Both scores were lower for LoF variants A119V[31] and S350L[13,14,16,32], while only insulin binding was lower for D734A, as expected[13,16,33] (Fig. 1E).

False discovery rates were computed by a bootstrapping approach using all barcode scores per variant across replicate sorts of the same cell library. Replicates were finally averaged and displayed in array form in variant effect maps. (Fig. 1F, Supplementary Fig. 4A). Variant-effect maps and function scores were consistent across replicates (Supplementary Fig. 4A,B), and PCA revealed clusters corresponding to distinct experimental conditions (Supplementary Fig. 4C). Means of replicate scores were used in subsequent analyses.

### Variants reducing cell surface expression and/or insulin binding

Of 15,996 variants in the plasmid library, 13,638 were detected, on average, across experiments (79% of all possible missense variants, or 88% of the library (Supplementary Table 1)). Binding scores for insulin and antibodies showed distinct patterns (Fig. 2A). For most regions binding was broadly similar for each ligand, in keeping with loss of receptor expression, exemplified by cysteine-rich domain (CRD) cysteines involved in intramolecular disulphide bonds (Supplementary Fig. 5A). These stood out as blue columns in variant effect maps, indicating that any change from cysteine is highly deleterious to expression (Fig. 2A, Supplementary Fig. 4A). In contrast, but in keeping with prior literature, the consequence for cell surface expression and insulin binding of mutating residues subject to N- or O-linked glycosylation[34,35] were negligible when averaged across all missense mutations at each site, with the marginal exception of N364. Some individual variants at these sites do significantly alter expression and/

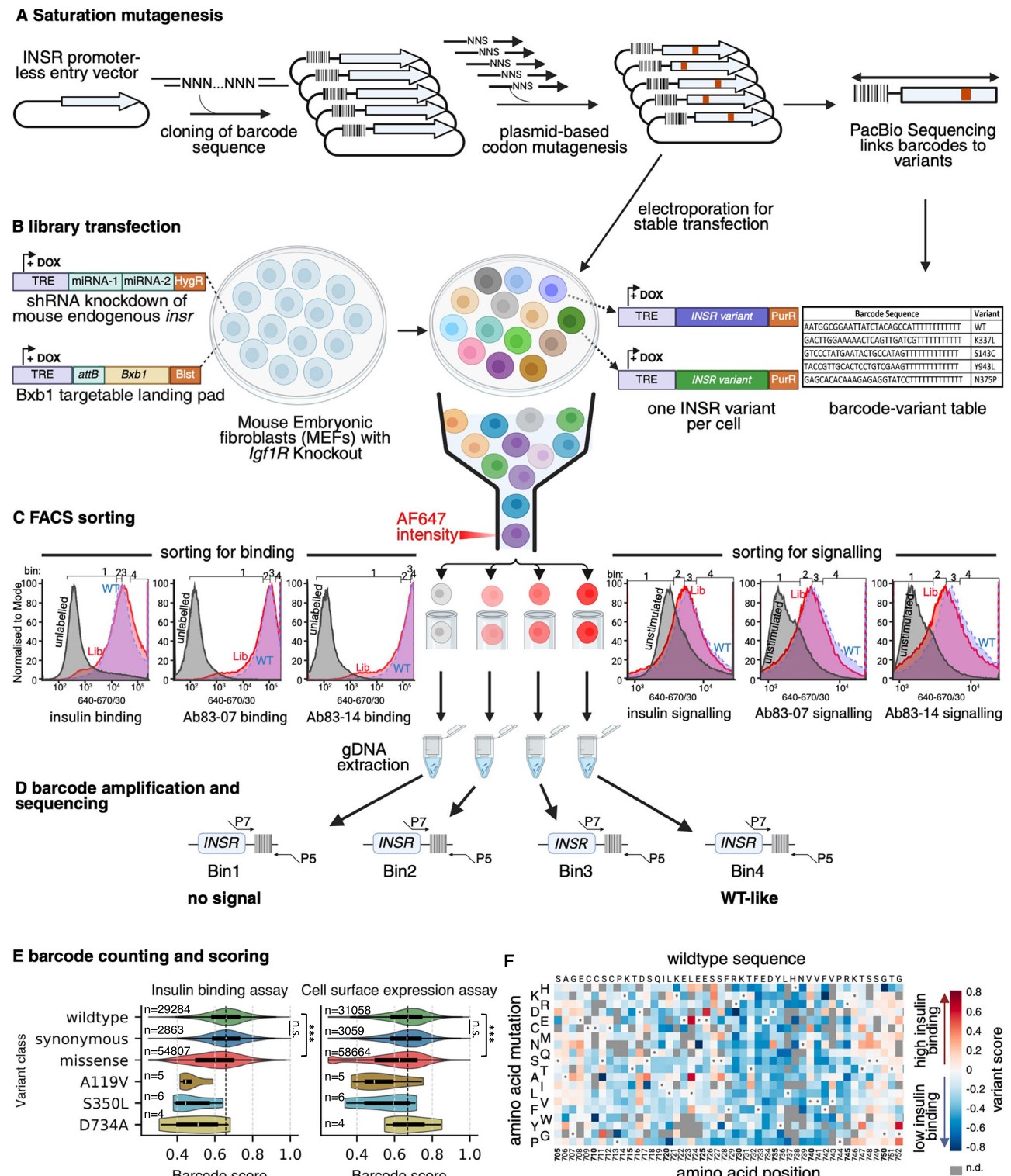

**A** Saturation mutagenesis

**B** library transfection

**C** FACS sorting

**D** barcode amplification and sequencing

**E** barcode counting and scoring

**F** wildtype sequence

or binding, but this is presumably a consequence of structural alterations specific to those variants, rather than relating to loss of glycosylation per se (Supplementary Fig. 6). Residues whose mutation reduced expression are illustrated in the cryoEM structure of the 4-insulin-bound "T" conformation of the receptor in Supplementary Fig. 5B.

As expected, binding of the antibodies correlated strongly (Fig. 2B, Supplementary Fig. 5C). Non correlating residues corresponded closely to epitopes for either mAb 83-14[28] (Fig. 2C), or mAb 83-7[28] (Fig. 2D), demonstrating the utility of the assay for epitope

mapping. For subsequent analyses, means of the scores for the two antibodies were taken as the (cell surface) "expression score", except for the 16 epitope residues with discordant scores (Fig. 2B-D), for which the single score from the antibody whose epitope was not mutated was used.

Important differences were apparent between insulin and antibody binding (Fig. 2A boxed regions). Most prominently, insulin binding was strongly, selectively disfavoured by mutations in the α-chain component of the insert domain, seen as a blue blush in the variant effect map in Fig. 2A. This region corresponds to the so called α

**Fig. 1 | Overview of workflow for massively parallel analysis of INSR variant effect. A** Schematic of saturation mutagenesis of the INSR ectodomain, including barcoding of an entry vector with 30 random nucleotides, PCR-based INSR ecto-domain mutagenesis to generate the barcode-variant library, and long read DNA sequencing to phase barcodes and mutations. **B** Transfection of the barcoded library into *Igf1r* knockout Mouse Embryo Fibroblasts (MEFs) with doxycycline-inducible INSR knockdown and a Bxb1-targetable landing pad to generate a large library of cells, each with a uniquely barcoded, conditionally expressible INSR variant. **C** Sorting of the cellular variant library into four bins based on binding assays (left) or signalling assays (right). Bin thresholds are shown above the plots, with WT (blue), variant libraries (red), and control (grey, no transfected MEFs) overlaid. At least 13 million cells were sorted into each bin. **D** Extraction of gDNA from each bin and amplification of barcodes with primers linked to Illumina p5 and p7 sequences. **E** Violin plots representing distribution of insulin binding (left panel)

and cell surface expression (right panel) barcode scores for WT, synonymous, missense, and three known pathogenic variants. Boxes = interquartile range (IQR), with median line. Whiskers denote 1.5 × IQR. Two-sided Mann–Whitney U tests were used to compare insulin binding score distributions between wild-type (WT) and missense variants ($p < 10^{-308}$), and between WT and synonymous variants ($p = 0.8$). The same testing was undertaken for WT and missense ($p < 10^{-308}$), and WT and synonymous ($p = 0.052$) expression scores. *** indicates $p < 0.005$; n.s. = not significant. **F** A detail of a sequence-function map representing variant scores for insulin binding for amino acids 705-751. Cell colour indicates the score for a single amino acid change. Variants are arrayed by position (column) and variant amino acid (row). Positive scores (in red) and negative scores (in blue) indicate better and worse function than wild-type in the assay, respectively. Grey cells are missing data and white cells with a dot in centre indicate the wild type residue at that position. Figure created in BioRender. Luijten, I. (https://BioRender.com/z6ak0zq).

−C-terminal (αCT) (residues 716–746), an α-helix forming a key part of insulin binding site 1, formed by the L1 domain of one INSR monomer and α-CT′ and FnIII-1′ loop of the other monomer[3,4].

We then compared median antibody binding scores with median insulin binding scores for 80 residues implicated structurally in insulin binding in sites 1 and 1′ (henceforth "site 1"), based on proximity to insulin in available structures or prior alanine scanning mutagenesis studies[4,5,36–41] (Supplementary Table 2), comparing them to cysteine-rich domain (CRD) cysteines (Fig. 2E). Mutation of site 1 residues slightly reduced cell surface INSR expression compared to WT receptor, while severely reducing insulin binding (Fig. 2E). This was in contrast to CRD cysteine mutations, which nearly all reduced expression and insulin binding (Supplementary Fig. 5A). This assessment was confirmed by agnostic statistical comparison of expression (mAb binding) and insulin binding scores, which revealed a close correlation between regions showing selective loss of insulin binding, and site 1 residues (Fig. 2A, dashed boxes). Mutation of amino acids implicated structurally in insulin binding at sites 2 and 2′ ("site 2")[4,5] (Supplementary Table 2) had no effect on receptor expression, and only mildly reduced insulin binding on average (Fig. 2F, and Supplementary Fig. 5D). Moreover although mutation of a small number of site 2 residues severely impaired both expression and binding (e.g., I512, L514, W516, F530, I561, K583), very little overlap was seen between regions with selective loss of insulin binding and Site 2-implicated residues (Fig. 2A, dashed boxes and blue asterisks respectively).

To assess residues important for insulin binding more agnostically, we coloured the INSR structure based on residues' mean insulin-binding scores, after excluding positions where mutations markedly reduce expression (Fig. 2G, and Supplementary Fig. 5E, F). This revealed dense clustering of the lowest binding scores in apposition to insulin, again in the αCT′ of one monomer and the L1 element of the other monomer that form insulin-binding site 1. Clusters of residues whose mutation selectively affected binding were also identified in interdomain hinge regions. Residues around insulin-binding site 2 did not stand out, however, as the small number of site 2 residues whose mutation severely reduces receptor expression were excluded from this representation due to their effect on expression.

Overall, 4,687 missense variants from 14,576 with scores (32%) significantly reduced expression, while 1,469/14,576 (10%) increased it (Fig. 2H). Approximately 4,623/14,243 (32%) reduced insulin binding and 961/14,243 (7%) consistently increased it (Fig. 2I). To validate findings, we curated all previous functional studies of individual variants, whether disease-associated or studied during structure-function investigations. Given variable experimental approaches used over 36 years we stratified variants pragmatically into 3 groups for each functional attribute: severe (<10% wild-type) or complete loss of function (LoF), intermediate LoF, and unimpaired function, also adjudicated on confidence in prior findings (Supplementary Data 3,4). INSR expression and insulin binding scores from MAVE and literature strongly agreed. For cell surface expression, 10/13 reported severe LoF variants

and 7/13 intermediate LoF variants were called as LoF with high confidence in the MAVE (Fig. 2H). The only reported severe LoF variants discordantly classified were W659R, reported in Donohue syndrome in trans with V1054M[42], C286Y, which had only a very low confidence score, and R762S, discussed below. Prior evidence for W659R LoF was a single immunoblot after transient overexpression, showing severely impaired prorotor processing[43]. No difference was found for insulin binding or signalling in the MAVE, however, despite at least 6 barcodes representing the variant in all experiments. The median binding score for the 19 substitutions evaluated at this position was moreover only −0.06 (interquartile range −0.15-0.07). Our consistent findings in orthogonal assays do not exclude pathogenicity, but suggest that any LoF for W659R is not captured in static expression and binding assays of the overexpressed variant in MEFs.

3/24 INSR variants (R525E, L579A and K730A) with reported normal expression showed loss of expression in this study (Fig. 2H). Prior evidence for all 3 variants came from transient overexpression and immunoblotting in a single study, however insulin signalling was reportedly reduced for all 3 variants in the same study[44]. They also all showed reduced insulin binding in the MAVE, suggesting that the prior immunoblotting assay may have been insufficiently sensitive to detect reduced expression.

For insulin binding, 10/12 reportedly severe LoF variants and 3/5 intermediate LoF variants with scores were called as LoF (Fig. 2I). One discordant reported severe LoF variant, P220L, had a low confidence LoF score in the MAVE, while the other - R762S – was clearly discrepant, being called as a gain-of-binding variant. R762 lies within the furin cleavage motif, and R762S was reported previously to be expressed but to show severely impaired insulin binding due to failed pro-receptor cleavage[45], a defect correctable by light trypsinisation. As trypsin was used in this study prior to flow sorting, it is likely that this remedied defective prorotor cleavage, restoring insulin binding.

Indirect evidence of loss of function was also obtained for the 11 documented pathogenic variants missing from the MAVE library: in Supplementary Fig. 7 function scores obtained for the other 77 pathogenic variants are plotted against the corresponding position median score, showing that median scores predict the effect of mutations with more than 50% accuracy. This means that even where specific variants of interest are missing from the MAVE, behaviour of different co-located variants still offers predictive value. Median scores per position for the pathogenic variants not directly assessed in this study are tabulated in Supplementary Table 3, showing that two residues show loss of function for one assay, four for two assays, and five for all three assays.

**Variants increasing insulin binding**
For some residues, most mutations increased insulin binding, sometimes without increasing expression. Most striking was G333, at which 16/17 substitutions increased binding, despite 14 also decreasing expression (Fig. 3A, B). For D627, 10 of 17 substitutions tested

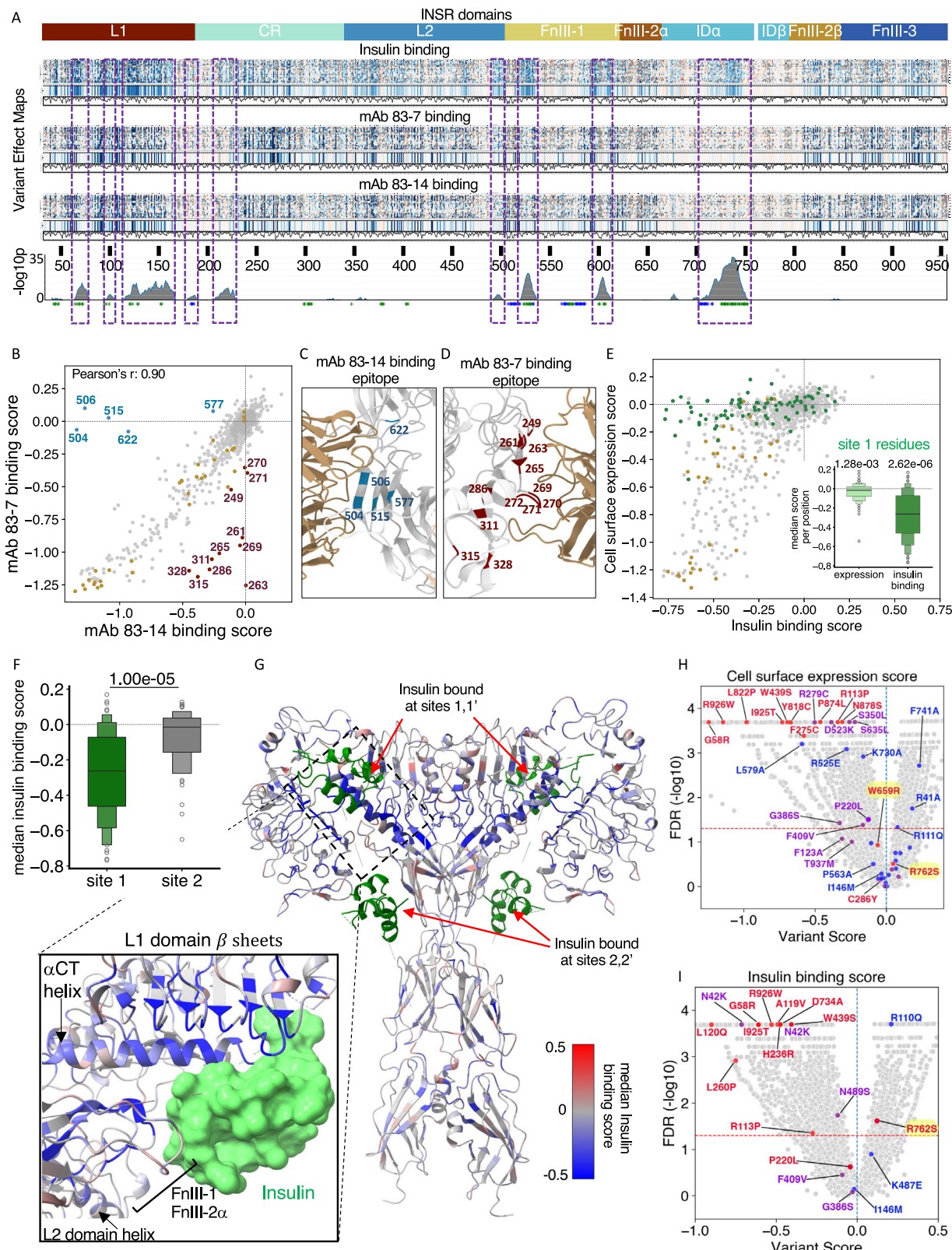

increased both binding and expression, with the greatest effect for D627A (Fig. 3A, B), in keeping with previous alanine scanning mutagenesis[46]. In vitro insulin binding assays confirmed increased insulin affinity and maximal binding for INSR G333Q (Fig. 3C), with a mild increase in cellular receptor protein on immunoblotting (Fig. 3D, E). INSR D627A, in contrast, showed reduced affinity and maximal

binding in vitro (Fig. 3C), but this was offset by strongly increased expression (Fig. 3D, E).

The insulin receptor adopts multiple conformations ranging from an inverted "V" conformation in the apo (insulin-unbound) state, through intermediate conformations with 1-3 bound insulin molecules to a "T" shaped conformation with 4 insulin molecules bound under

**Fig. 2 | Effects of mutations on cell surface expression and insulin binding.**
**A** Heatmap of scores for insulin, mAb 83-7 and mAb 83-14 binding, referenced to INSR domain architecture. Unscored variants are grey. 'median pp' (bar plots below heatmaps) indicates median variant scores per position and 'N variants pp' (line plots beneath bar plots) indicates number of variants scored per position. The bottom trace indicates -log10p from the Wilcoxon signed rank test (one sided) applied to sliding windows of 15 residues, testing the hypothesis that mAb and insulin binding scores are the same, with the alternative that binding scores are lower. Dashed purple boxes demarcate regions with most significant difference.
**B** Scatter plot of median scores per position for mAb 83-7 binding with dark-brown outliers and mAb 83-14 binding with light-blue outliers. Disulphide-bonded cysteines are golden brown. **C, D** Detail of INSR structure (PDB 4ZXB) showing mAb 83-14 and mAb 83-7 epitopes respectively, with outlier residues from (**B**) high-lighted. **E** Median scores for insulin binding and expression, calculated as mean of mAb-binding scores. Insulin binding site 1 residues ($n = 80$) are green. Disulphide-

bonded cysteines are golden brown. The inset boxenplot displays expression and binding scores for Site 1 residues. Inner boxes show interquartile ranges, and median line within, with sequential boxes showing 50% of remaining datapoints. A two-sided Mann-Whitney U Test was used to compare scores against those of all other residues. **F** Median insulin binding score distribution for site 1 and site 2 ($n = 47$) residues, shown as boxenplot as in (**E**), and compared by two-sided Mann Whitney U test. **G** 4 insulin-bound INSR structure (PDB: 6PXV) coloured by median insulin binding score, excluding poorly expressed residues (Supplementary Fig. 5E,F). Insulin molecules are green. **H, I** Volcano plots of expression or insulin binding score respectively for variants with scores from at least two experiments (x axis) against -log10 FDR (y axis). Annotated points are functionally studied variants Colours denote degree of functional impairment in prior assays. Red = severely impaired; Blue = little or no reported impairment; Purple = intermediate impair-ment (Supplementary Data 3). Dashed horizontal lines represent FDR of 0.05.

saturating conditions[4–6] (Supplementary Fig. 8). G333 lies in the CR:L2 hinge region, which remodels in transition from apo to insulin-bound receptor[5] (Supplementary Fig. 8). Substitution of glycine for bulkier amino acids may alter the energetics of this change. D627 lies in the FnIII-2α domain (Supplementary Fig. 8).

Most other missense variants that increased insulin binding also increased expression. Some stood out from others at the same amino acid, implying gain of function unique to that substitution (e.g., L1 domain: D169F; FnIII-1: N541D, N568C; FnIII-2α: T653N, K679C, T684W). Regions where most mutations increase insulin binding and/or receptor expression in the current MAVE (Fig. 3F) tend to be located at or near dynamic sites of interdomain interaction, for example between L1' and FnIII-2α in the unbound receptor, or in part of IDα that is unresolved in available structures and thus likely to be conformationally mobile.

## Effects of INSR mutations on insulin signalling
To assess maximal insulin signalling ($E_{max}$), we used a fluorescently labelled antibody to quantify Akt phosphorylation (on Ser473/474) in response to 100nM insulin. High level results are illustrated in Fig. 4A, with insulin binding results as a comparator. While there was broad similarity between binding and signalling maps, two hotspots where signalling was relatively more impaired than binding were seen, in the L2 and at the junction of the L2 and CR domains (orange hatched boxes, Fig. 4A). Several regions were also seen where binding was relatively more impaired than signalling, especially including elements of binding site 1 in the L1 and FnIII-1 domains (purple hatched boxes, Fig. 4A). The αCT helix was not prominent in this analysis, likely because the effect of its mutation on insulin binding is severe, with commensurate effects on signalling. Analysed together, mutations in insulin binding site 1 residues indeed had less effect on signalling than on binding (Fig. 4B). These findings are in keeping with the "spare receptor" concept, whereby maximal signalling is seen at relatively low levels of receptor occupancy[47]. Mutation of site 2 residues only mildly reduced binding and signalling in group-based analysis, with much more deleterious effects on binding and signalling for only a small subset of constituent residues (Fig. 4C). Mutation of most of these residues also reduced cell surface expression (Supplementary Fig. 5D and inset), although mutation at positions 529 and 530 had a selective effect on insulin binding. The effect of mutating site 2 mutations was thus much smaller than for site 1 mutations (Fig. 4D). Well-expressed variants with the largest loss of signalling affect residues clustered around site 1, with others in the "legs" of the receptor involved in unliganded receptor compaction (Fig. 4E).

5,358 individual variants from 14,544 with scores (37%) reduced signalling (Fig. 4A, F). All 8 variants shown empirically in previous studies to show severe loss of signalling, and 8/13 of all variants reported in human severe IR were called concordantly in the MAVE. Seven of the remaining variants had scores that were not significantly different to WT (FDR ≥0.05) (Fig. 4F). As for expression and binding

assays, indirect evidence of loss of signalling was also obtained even for pathogenic variants not directly studied in the MAVE, with 7 of the 11 residues affected yielding median signalling scores in the loss of function range (Supplementary Fig. 7, Supplementary Table 3).

1,501 variants increased peak signalling in the MAVE (Fig. 4A, F; and Supplementary Fig. 9). Mutation of D627 and to a lesser extent G333 increased signalling in the MAVE, although validation studies suggest this is likely a consequence of increased expression (Fig. 3C-E). Threonine 221 emerged as a residue of potential interest based on the median score for codon 221 variants (Supplementary Fig. 9A), but this was based on only 4 scores. Other variants warranting future assessment based on selectively increased signalling include residues forming part of insulin binding site 1 (L2 domain: E343I, N376W; FnIII-1: N568C, P576C), and residues involved in interdomain interactions that rearrange upon insulin binding (e.g., CR domain: C293F; FnIII-2α: R661W; FnIII-2β: Y839V; Supplementary Fig. 9A-G).

## Diagnostic performance of empirical functional results
We next asked whether INSR variant function scores can support genetic diagnosis. In variant-level analysis, MAVE scores broadly agreed with annotations in the ClinVar database, with known benign variants scoring higher than pathogenic variants (Fig. 5A). 102 of 127 (80%) variants recorded in Clinvar were classified as VUS, however, with a subset showing conflicting annotations. This underlines the potential value of MAVE data in refining genetic diagnosis. Indeed, on using scores for well studied variants to define loss of function cut offs (Supplementary Fig. 10), 31/83 variants designated VUS in ClinVar had loss of function in at least one MAVE assay, 19 in 2 assays, and 6 in all 3 assays, including variants altering critical disulphide binding cysteines (Supplementary Fig. 10B, C), strongly arguing for reassignment of these VUS as pathogenic. However the high rate of VUS in ClinVar blunted utility of these data as a gold standard against which to eval-uate MAVE performance.

To circumvent this problem we turned to analysis at the patient level, taking into account recessive inheritance and thus the presence of biallelic variants in most patients. We first reevaluated 125 reported cases of DS, RMS and Type A IR, as well as clinical and genetic findings from two large IR genetic testing laboratories, adjudicating on classi-fication of reported phenotypes to ensure consistency (Supplemen-tary Data 3). Altogether we identified 88 unique extracellular missense variants as pathogenic in 99 biallelic combinations, 43 of them homozygous. We calculated average function scores for the pairs of homozygous or compound heterozygous variants found in each patient. In some cases, extracellular missense mutations co-occurred with nonsense, frameshift or whole exon deletions in the extracellular domain, or intracellular domain mutations. These mutations were assigned low scores reflecting their anticipated effects (**Methods**). We found a clear correlation between biallelic empirical scores and clinical severity of IR, with the lowest scores in DS, intermediate in RMS and

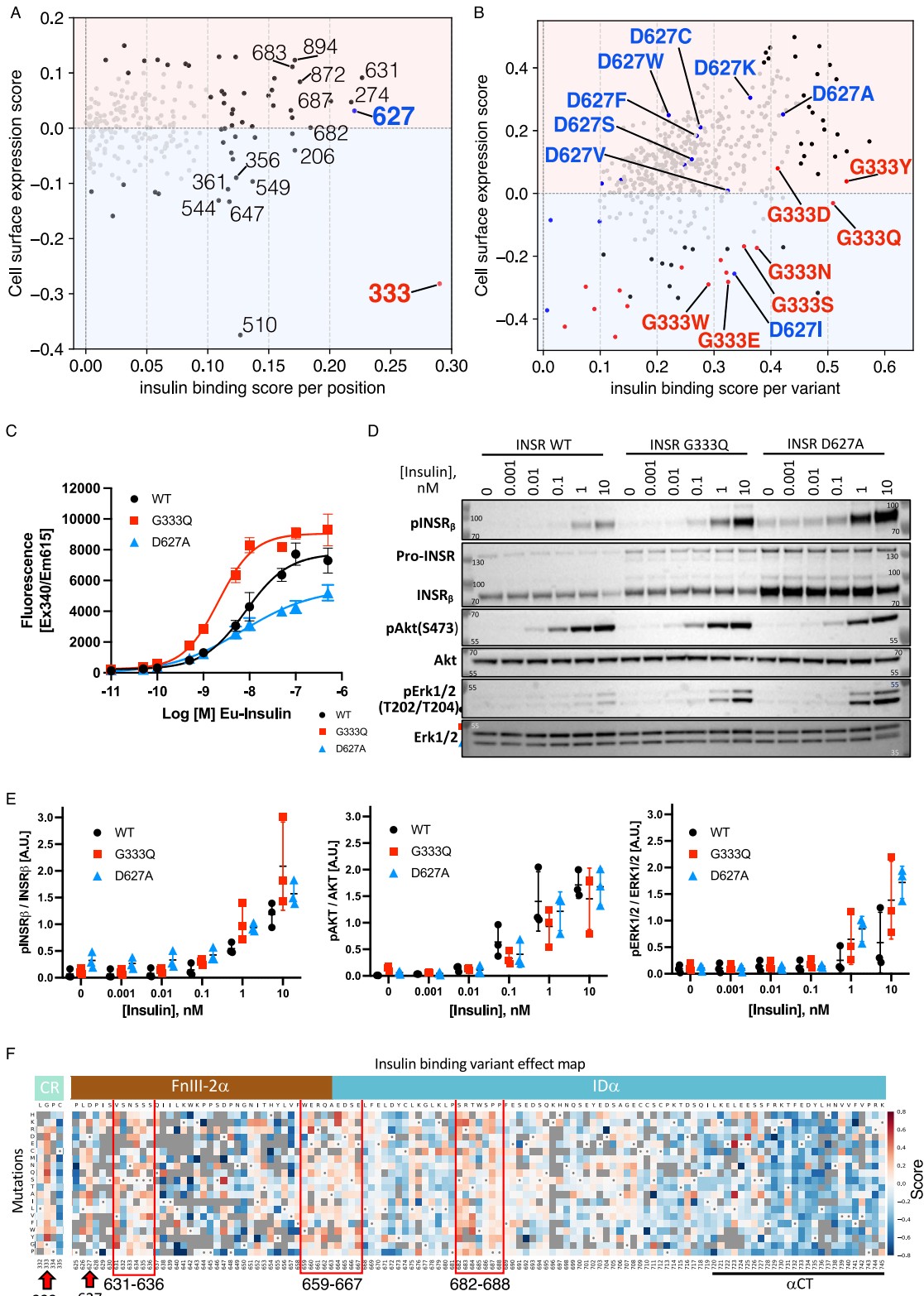

**Fig. 3 | Variants Increasing Insulin Binding. A** Scatter plot of median cell surface expression scores against insulin binding scores, focused only on gain of insulin binding (score > 0). **B** Similar scatter plot of individual missense variants with significantly increased insulin binding scores (score > 0 and FDR < 0.01). Variants at G333 and D627 are shown in red and blue respectively. **C** Binding of increasing concentrations of Eu-labelled Insulin to WT and mutant INSR over 16 hr at 4 °C. Data from 3 replicates are shown, represented as mean ± standard deviation (**D**) Representative immunoblot of lysates of MEFs expressing either WT or mutant INSR and stimulated for 10 min at 37 °C with increasing concentrations of insulin. **E** Densitometric analysis of Western blot images from 3 independent experiments, represented as mean ± standard deviation. **F** Variant effect map excerpt with highlighted gain-of-insulin-binding hotspots, visually ascertained.

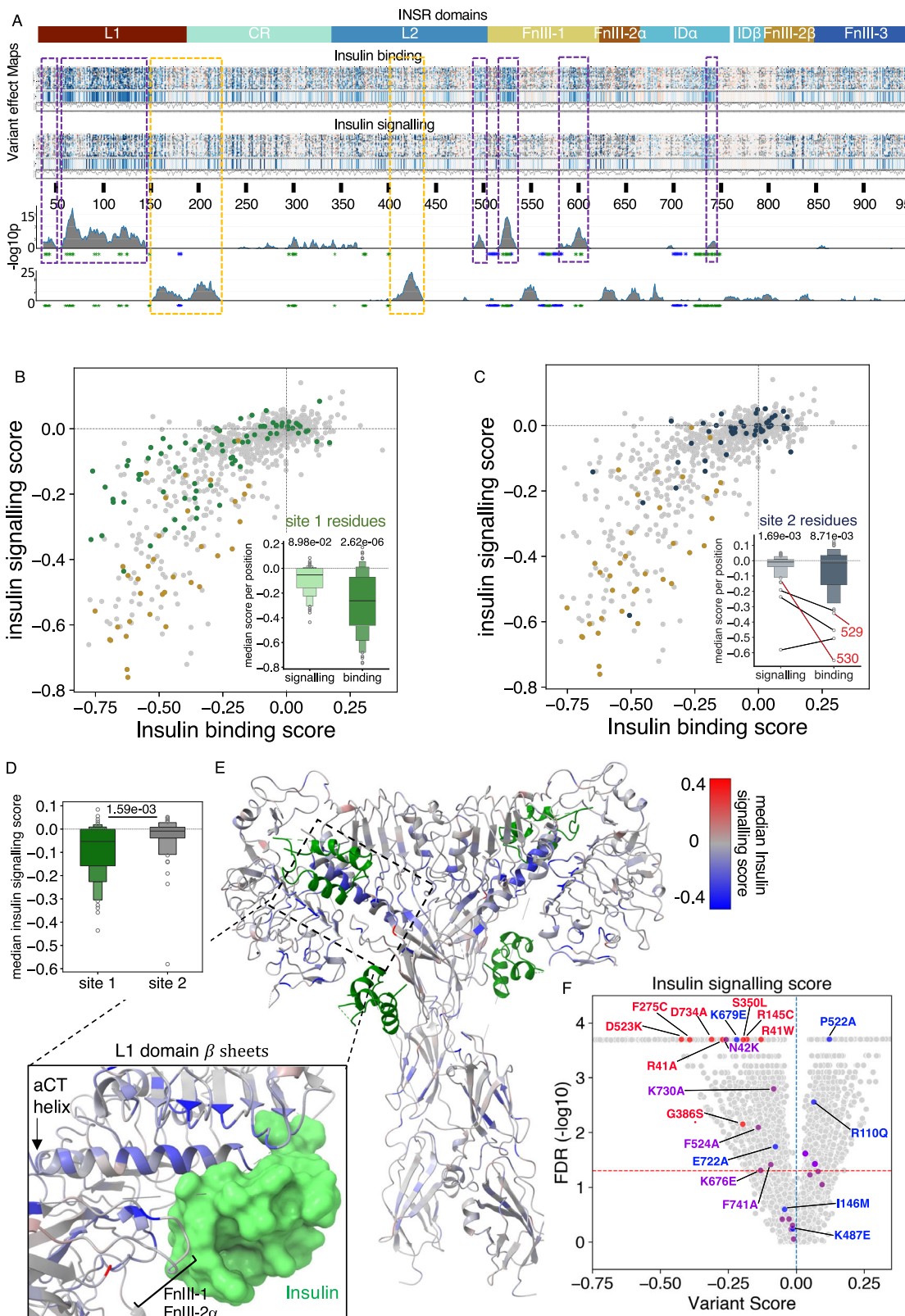

highest in type A IR (Fig. 5A). Scores in disease groups were strikingly lower than for the presumed benign homozygous variants in gnomAD, and with synonymous variants affecting the same codons, but overlapped with scores for all variants in gnomAD, as expected for a recessive condition.

We then evaluated the ability of biallelic scores to predict severe IR, taking pathogenic mutations from the curated, adjudicated set

(Supplementary Data 4), and defining homozygous variants in gnomAD and synonymous variants at the codons affected in pathogenic variants as benign.

Many variant effect predictors (VEPs) have been developed to aid in silico prediction of variant pathogenicity, latterly incorporating artificial intelligence. We compared performance of the MAVE against a basket of VEPs, using both results from individual MAVE experiments

**Fig. 4 | Effects of variants on maximal INSR signalling. A** Heatmap of variant scores for insulin binding (top) and signalling (bottom), referenced to INSR domain architecture. Unscored variants are grey. 'median pp' (bar plots below heatmaps) indicates median variant scores per position and 'N variants pp' (line plots below bar plots) indicates number of variants scored per position. The bottom two traces indicate -log10p arising the Wilcoxon signed rank test (one-sided) applied to sliding windows of 15 residues across the region studied, testing the hypothesis that binding and signalling scores are the same, with the alternative that binding scores are lower (top trace) or higher (bottom trace). Rectangular dashed purple or orange boxes demarcate regions showing the most significant difference. **B** Median scores for insulin binding against median signalling scores with site 1 residues highlighted in green. The inset boxenplot shows median insulin binding and signalling scores for site 1 residues ($n = 80$). Inner boxes show interquartile ranges, with median line at centre, with sequential nested boxes showing 50% of remaining datapoints A two-sided Mann-Whitney U Test was used to compare scores against those of all other residues. **C** Scatter plot of median insulin binding against median signalling scores, with dark-blue-highlighted site 2 residues. The inset boxplot, as before, displays median binding and signalling scores for site 2 residues ($n = 47$) with statistical comparison as for site 1. **D** Median insulin signalling score distribution shown as boxenplot for site 1 ($n = 80$) and site 2 ($n = 47$) residues, compared using two-sided Mann Whitney U test. **E** 4 insulin-bound INSR structure (PDB: 6PXV), coloured by median insulin signalling scores after excluding residues markedly lowering expression according to Supplementary Fig. 5E,F. Insulin molecules are green. **F**, Volcano plots of insulin signalling scores for all variants with scores from at least three experiments (x axis) against -log10 FDR (y axis). Annotated points are functionally studied variants. Red = severely impaired; blue = little or no reported impairment; purple = intermediate impairment (Supplementary Data 3). The dashed horizontal line denotes an FDR of 0.05.

(expression, binding and signalling) and of aggregated scores (lowest score across all three assays and sum of all three scores). We found near-perfect predictive performance of the current MAVE and the top VEPs, all with receiver operating characteristics (ROC) areas under the curve (AUC) above 0.96 both for all pathogenic variants, and for those seen only in severe recessive disease (Fig. 5B). Among MAVE scores, the sum of scores for each assay, the lowest score in any assay, and the insulin binding score performed best, followed by expression and signalling scores. Interestingly, despite the high predictive performance of MAVEs and VEPs, even the top VEP-derived scores correlated only moderately with MAVE data (range of correlation coefficients for the best performing MAVE score (sum of scores) and VEP scores 0.55 (AlphaMissense) to 0.16 (GRE + +)), whereas VEPs often correlated strongly with each other (correlation coefficients up to 0.98; Fig. 5C).

### Responsiveness of mutated receptors to antibody partial agonists

The translational aim of this study extended beyond addressing the diagnostic "VUS problem". We also sought to identify variants showing severely impaired insulin responsiveness that exhibit responsiveness to stimulation by the monoclonal antibodies tested, reasoning that patients with extreme IR harbouring at least one such variant may benefit from future antibody-based therapy, as previously suggested[13–16]. We tested this by assessing signalling responses of the whole library to 10 nM antibodies *in lieu* of insulin. Both antibodies elicited a pattern of signalling overlapping heavily with that of insulin (Fig. 6A), however key areas of difference were apparent. In total we identified 1193 and 1141 variants both poorly responsive to insulin and more responsive to mAb 83-7 and mAb 83-14 stimulation, respectively. Hotspots of relative antibody sensitivity were located in the αCT region noted previously, and in regions of the L1, CR and FnIII-1 domains (Fig. 6A). Most variants with relatively preserved antibody stimulation were responsive to both monoclonal antibodies, with differential responses only partly accounted for by residues forming parts of mapped antibody epitopes (Fig. 6B–E).

To focus on the variants with the strongest evidence for impaired signalling we selected variants with scores in all five insulin signalling assay replicates and with median score in the lowest 80% of the observed loss-of-function range. We further filtered these variants to include only those with scores in all three antibody signalling assay replicates. For each of these robust loss-of-insulin-signalling variants we then calculated the difference between antibody and insulin-stimulated signalling scores, which we call the "selective sensitivity score". These selective sensitivity scores are represented as Manhattan plots for each antibody in Fig. 7A and B, and mapped onto the 4 insulin-bound structure in Fig. 7C. Areas identified as showing selective activation by antibodies are boxed for reference, and the mean sensitivity scores for sliding windows of 15 residues are also plotted. D734A, an antibody-responsive variants that was the focus of previous proof-of-concept studies, was confirmed to signal on antibody exposure, but was far from the most antibody-sensitive variant (Fig. 7A,B). Selective sensitivity scores for all variants with positive scores for both antibodies tested are plotted in Fig. 7D, showing strong correlation and identifying variants with the most robust responses to both antibodies. The list of variants generated in this analysis offers good candidates for future translational studies of humanised versions of the antibodies studied (Supplementary Data 5).

## Discussion

The insulin receptor plays a critical role in diabetes and its treatment, in growth, development, cancer, and beyond. Yet despite decades of study[2,3], its complex insulin binding kinetics and signal transduction mechanism are incompletely understood. For people with extreme IR due to biallelic mutations in the *INSR* gene, therapeutic options remain poor.

Around 90 pathogenic missense variants have been described in the insulin-binding portion of the insulin receptor since first description of such variants[45,48], with more regularly found. Functional studies, using various cell types and assays, have been published for only around 40 of these. We now describe a MAVE of cell surface expression, insulin binding and insulin-induced signalling of around 14,000 such INSR variants—79% of all possible missense mutations.

The current MAVE accords with the large majority of prior functional studies, especially for severe LoF variants. Discrepancies are attributable in some cases to assay conditions (for example relating to "correction" of mutation effect by trypsin exposure), and perhaps to study of different receptor isoforms (which is not documented in all published studies). Some discordance may reflect uncertainty in MAVE results, but in many cases previous assays are likely the larger source of error. Prior assays generally involved pairwise comparison of variant and WT receptors, often expressed from randomly integrated constructs, sometimes with functional readouts of unproven linearity. In the MAVE, in contrast, most variants were represented by more than one barcode/clonal cell line, always with multiple biological replicates. Moreover all variants were compared simultaneously against each other and thousands of WT receptors, with a quantitative readout.

The results presented here also accord with structural studies of high affinity insulin binding site 1[2–5]. Many mutations of the residues involved severely reduce insulin binding, often with little or no reduction in receptor expression, and the MAVE alone would offer a highly suggestive view of the binding site. Binding site 2, in contrast, would not be predicted by the MAVE, with very little evidence that single mutations of the constituent residues affect insulin binding. However site 2 is a low affinity binding site, and moreover the 4 insulin-bound structures recently reported were only observed when saturating insulin concentrations (> 100 nM) were used[4,5]. Their relevance in vivo, where insulin concentrations range from around 0.01-10 nM, is doubtful. Moreover very few point mutations of site 2 residues thus

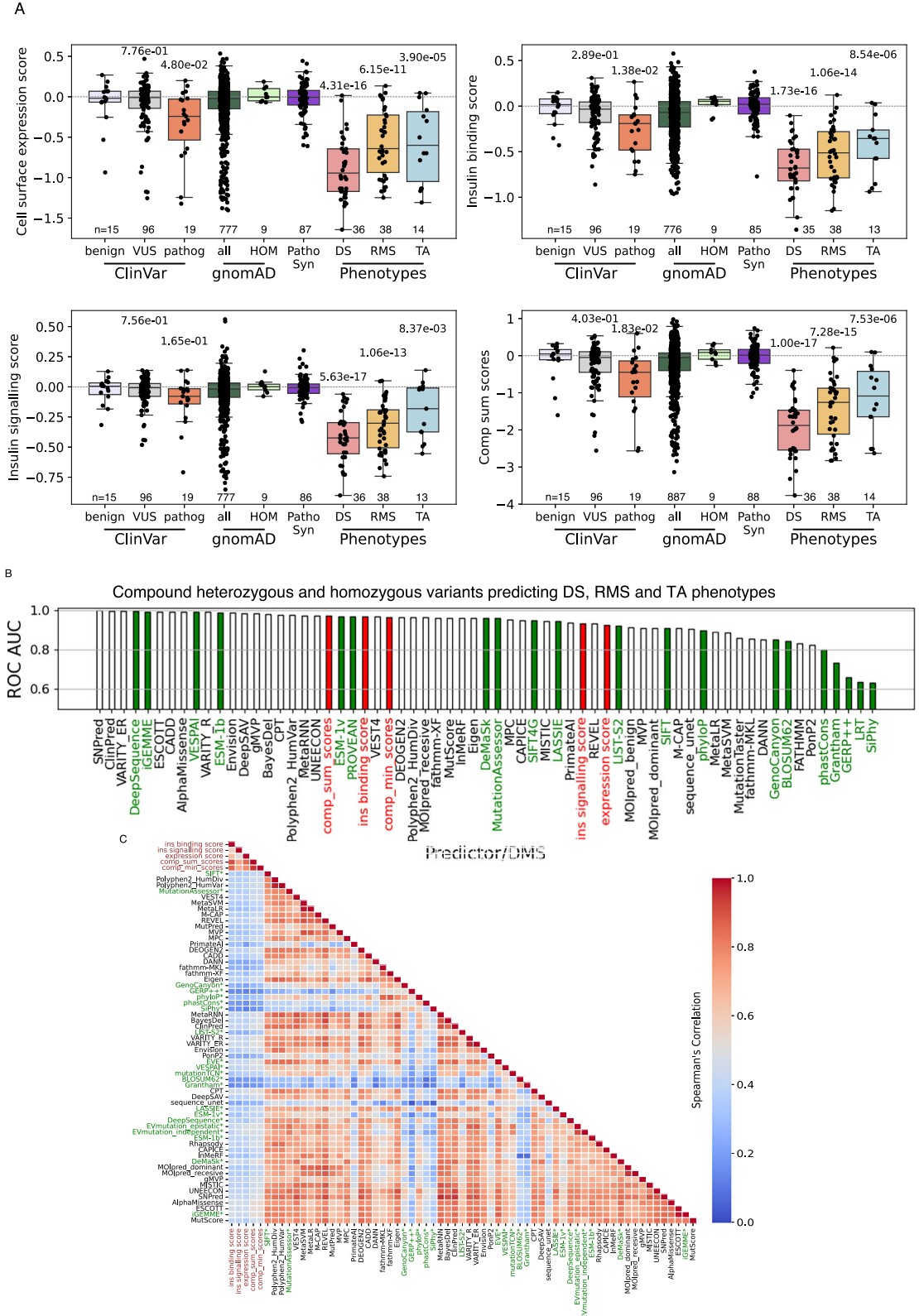

identified have been assessed for insulin binding. Whether our failure to detect deleterious effects on binding of individual site 2 mutations is a consequence of the insulin concentration employed, or reflects small contributions of individual residues to binding, requires further study.

The MAVE performed outstandingly in predicting pathogenicity, like the best in silico variant effect predictors (VEPs). Yet not all receptor attributes have been assayed in this study, and by

incorporating further assays, the ability of INSR MAVEs to discriminate pathogenic mutations may be enhanced further. The current MAVE validates the best current VEPs empirically, however, interestingly, correlation among scores from VEPs and MAVEs are modest (e.g., R = 0.53 between INSR sum of scores and AlphaMissense) but close to the average reported correlations between other MAVEs and VEPs (R 0.56–0.61 depending on assay[49]). Factors influencing these

**Fig. 5 | Evaluation of multiplexed assay results as an aid to genetic diagnosis.**
**A** Boxplots showing expression, insulin binding, and signalling score distribution, as well as composite sum of scores for different groups of variants. Boxes = interquartile range (IQR), with median line. Whiskers denote 1.5 × IQR. ClinVar variants designated benign or likely benign, as variants of uncertain significance (VUS), or as pathogenic or likely pathogenic are shown. Where conflicting dual designations are present in ClinVar both are shown. Data for all variants and homozygous variants from gnomAD are also shown, as well as scores for curated homozygous and compound heterozygous (CompHet) pathogenic variants associated with Donohue syndrome (DS), Rabson-Mendenhall syndrome (RMS), or type A insulin resistance (TA). Finally, scores for all synonymous variants at residues subject to pathogenic mutations are plotted as "Patho Syn". See "Methods" for average score computation for compound heterozygous variants. P-values from two-sided Mann-Whitney U tests are displayed above relevant boxplots comparing disease variants vs Patho Syn variants, and ClinVar VUS and pathogenic vs benign variants. **B** Receiver Operating Characteristic (ROC) analysis showing the Area Under the Curve (AUC) for all curated individual pathogenic variants for DS, RMS, and TA. The analysis compares predictions made using a set of supervised (light grey) and unsupervised (green) methods for variant effect prediction with those based on expression, insulin binding, and signalling scores alone, as well as composite minimum or sum of scores calculated in this study (red). Homozygous gnomAD variants were used as the set of benign variants. **C** Spearman's correlation matrix for scores from this study (red axis labels) and those from supervised predictors (black axis labels) and unsupervised predictors (green axis labels).

correlations are incompletely understood, but include the suitability of the experimental model, and number of replicates (on the MAVE side) and the quality of training data and amount of systematic bias (on the VEP side). Moreover MAVEs and VEPs often aim to estimate different outcomes (gene function in a cellular model *vs* pathogenicity at organismal level) so are not expected to correlate perfectly.

Beyond tackling the VUS diagnostic "roadblock" and generating a comprehensive genotype-function maps, we also demonstrate the utility of MAVEs for functional mapping of antibody epitopes. More importantly, we exploit the multidimensional nature of our assays to inform translational studies of insulin receptoropathy. Specifically, we stratify INSR variants for studies of novel receptor agonists by identifying which variants are expressed at the cell surface and amenable to activation by non-canonical ligands. By comparing antibody and insulin-induced signalling we further stratify mutations prospectively for future trials of humanised antibodies or other novel INSR agonists, including peptides and aptamers[50]. Poor prognosis and short lifespan in DS greatly constrains time to assess new variants and the prospective functional data we provide will aid early identification of infants with INSR defects potentially responsive to INSR ligands with a distinct mode of receptor engagement to insulin.

This study also identified novel variants that increase maximal insulin binding and/or signalling, particularly clustered in regions known to contribute directly to insulin binding or to be involved in dynamic remodelling of the receptor on ligand binding. The assays we employed did not assess insulin binding or signalling in the physiological insulin concentration range, and further study of these variants using such physiological concentrations in future will be of interest.

Our study has some limitations. Around 20% of the full repertoire of missense mutations have not been studied, although in many cases inferences can be drawn from other substitutions of the same residue. We also used a single very high insulin concentration, reporting only maximal response to receptor activation, or efficacy, and missing potentially clinically significant rightward shifts in the sigmoidal insulin dose-response curve. Loss of phosphoAkt signal during cell processing may further contribute to loss of dynamic range in signalling assays. Some variants that are convincingly pathogenic on genetic and clinical grounds show normal behaviours in this study and many prior studies. These include I146M[51] and K487E[52]. This reflects the inability of simple assays to capture the full range of receptor attributes important for in vivo function, including dynamics of receptor recycling after ligand stimulation. For I146M and K487E, defects in receptor function were only unmasked on more complex cellular studies of receptor recycling kinetics[51,52]. It will be of interest to use the cellular INSR library generated in this study to extend analysis to dose response relationships and recycling kinetics. Future work could also further screen different ligands, different downstream pathways, or INSR expressed as a hybrid with the homologous IGF1R.

In summary, we report a large, multidimensional MAVE of the extracellular INSR. This has translational utility in monogenic IR, is an important starting point for more sophisticated studies of variant receptor attributes, informs on complex structure-function relationships of the INSR, and serves as a paradigm for study of other RTKs of high biomedical importance.

## Methods

### Curation of prior clinical, genetic, structural and functional data
Pathogenic *INSR* mutations were ascertained from ClinVar Miner[53] (accessed 2.8.2024), from curation of published reports, and from unpublished diagnostic testing results from national referral centres for severe IR in Cambridge, UK and Paris, France (Supplementary Data 4). All available clinical data were examined and subjective adjudication was made on clinical diagnosis, irrespective of reported diagnostic labels, based on reported biochemical severity of IR, birthweight and other phenotypic features, and longevity. Donohue syndrome and Rabson Mendenhall syndrome cannot be demarcated precisely, with longevity (1-2 years for Donohue syndrome, >10 years for Rabson Mendenhall Syndrome) a key discriminator. Prior published functional studies of *INSR* variants, both naturally occurring and generated for experimental purposes, were curated and assessed. A wide variety of functional studies was reported, complicating simple comparison of findings, whose details are summarised in Supplementary Data 3. For each of expression, insulin binding and signalling, findings were stratified into severe loss of function (LoF)(approximately <10% wild-type function), moderate LoF (>10% wild-type function) and normal function. *INSR* variants in gnomAD were drawn from gnomAD v4.1.0, accessed 6.8.2024).

PDB identifiers for receptor structures assessed and discussed were 8U4B for the human IR-B receptor, 6PXV for the human IR-A receptor, and 7STI, 7STJ and 7STH for the murine IR-A receptor. Structures were accessed from PDB on 5th July 2024 and viewed using UCSF ChimeraX [version: 1.8rc202405230136 (2024-05-23)]

### General experimental reagents
Enzymes were purchased from New England Biolabs unless otherwise stated. Oligonucleotides were from IDT and are listed in Supplementary Data 6. *Igf1r*-/- mouse embryo fibroblasts (MEFs) were obtained from the Cosgrove laboratory CSIRO, Adelaide, Australia[54]. MEFs were cultured in Dulbecco's modified Eagle's medium (Thermofisher, 41965039) supplemented with 10% fetal bovine serum and Penicillin-Streptomycin (50 I.U./ml). Cells were passaged by trypsinization using 0.25% trypsin-EDTA (Millipore-Sigma) and tested negative for *Mycoplasma* before use. All antibodies used are detailed in Supplementary Table 4.

### Plasmid construction
The open reading frame of the B isoform (exon 11 + ) of the human *INSR* gene (UniProt P06213-2) was subcloned from pCDNA5/FRT/TO/h*INSR*[13]. attB_INSR_P2A_PuroR plasmids were created by restriction cloning, combining amplicons from attB_sGFP-PTEN-IRES-mCherry-P2A-PuroR (a gift from Douglas Fowler, University of Washington) amplified with primers VA01//VA02, and pCDNA5/FRT/TO/h*INSR* amplified with primers VA03//VA04 and VA05//VA06. To permit barcode cloning into the attB_INSR_P2A_PuroR plasmid, new restriction

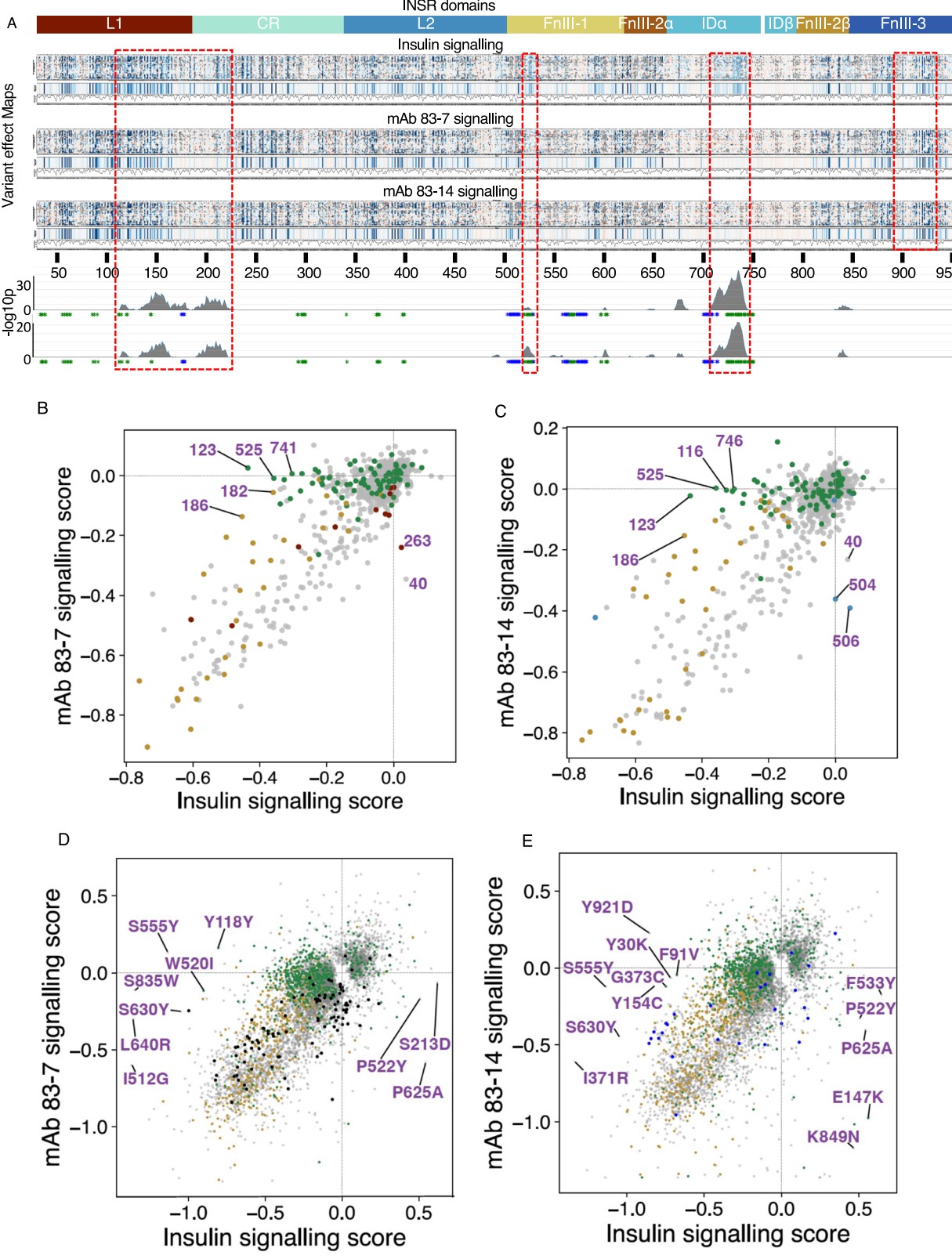

sites were created by digesting the plasmid with AfeI, dephosphorylating with Shrimp Alkaline Phosphatase (rSAP, M0371), and ligating an amplicon containing SbfI and MluI sites. The amplicon was made by annealing 10 mM of phosphorylated VA07 and VA08 primers before purification, ligation into the plasmid with T4 DNA ligase, and transformation into XL-10 gold cells. Correct insert orientation was confirmed by Sanger sequencing.

**Plasmid barcoding**

VA09 30mer barcoding oligos flanked by Sbf1 and AscI sites were synthesized and PAGE-purified before VA10 primer annealing and extension using Phusion HiFi PCR master mix (M0531S). 10 separate 50ul reactions, each using 0.5uM primer and 0.1uM barcoding oligo were mixed, purified using two MinElute PCR Purification columns (QIAGEN, 28004) and quantified with Nanodrop. 32 ng double

**Fig. 6 | INSR variants with impaired insulin binding that are potentially activatable by monoclonal antibodies. A** Heatmap of variant scores for maximal signalling induced by insulin, mAb 83-7 or mAb 83-14, referenced to extracellular domain architecture. Variants that were not scored are coloured grey. 'median pp' = median variant scores per position and 'N variants pp' = number of variants scored per position. The bottom two traces indicate -log10p arising from application of the Wilcoxon signed rank test (one-sided) to sliding windows of 15 residues across the whole region studied, testing the hypothesis that signalling scores for insulin and mAb 83-7 (top trace) or 83-14 (bottom trace) are the same, with the alternative hypothesis that scores for mAbs are higher than for insulin. Rectangular dashed red boxes demarcate the regions showing the most significant difference. **B**, **C** scatter plots of median signalling scores per residue for (**B**) insulin and mAb 83-7 with epitope residues in black, or (**C**) insulin and mAb 83-14 signalling, with epitope residues in light blue. Insulin binding site 1 residues are coloured green, and cysteine residues involved in disulphide bond formation golden brown in both plots. Selected residues are labelled by number. **D**, **E** show similar scatter plots, but this time showing all variants for which FDR < 0.05 in at least one assay.

stranded barcoding amplicon was digested overnight at 37°C in a 500ul reaction containing10ul Sbfl-HF and 10ul AscI, and purified with two MinElute columns. In a separate reaction, 7ug attB_INSR_P2A_-PuroR plasmid was digested for 2 hr in a 300ul reaction supplemented with 5ul Sbf1-HF and 5ul MluI-HF followed by 30 minutes dephosphorylation with Shrimp Alkaline Phosphatase (rSAP, M0371) and purification with QIAquick PCR Purification columns (QIAGEN, 28104). Digested barcoding amplicon was cloned into digested attB_INSR_-P2A_PuroR plasmid with T4 DNA ligase in 5 separate 3 ml reactions (each containing 600 ng digested plasmid, 5.25 ng digested barcoding amplicon, 300ul T4 DNA ligase buffer and 5ul T4 DNA ligase) with 4 hr room temperature incubation. Reactions were supplemented with 2.5ul each of the isoschizomers MluI-HF and AscI to remove any background from plasmid self-ligation. Ligation of MluI-HF and AscI-derived fragments abolishes the recognition sequences. 15 ml Qiagen PB buffer was added to each reaction, then eluted in 15ul using a DNA Clean & Concentrator column (Zymo Research, D4013). Eluted samples were pooled and digested by MluI-HF and AscI for 1 hr to remove residual background. 9ul of this reaction was transformed into 100 of XL10-Gold Ultracompetent Cells in 10 separate reactions (Agilent Technologies, 200315). Before plating, transformation mixes were pooled in a total volume of 10 ml, 11 ml sterile water was added and 400ul of the mixture was quickly spread on 52 150 mm agar plates with 100ug/ml ampicillin, using glass beads. Dilution plates were made with 10ul, 25ul and 50ul of the mixture to allow estimation of transformation efficiency and number of barcoded plasmids. All plates were incubated at 37 °C overnight. Colonies were scraped, pooled, and aliquots stored as pellets at −80 °C.

## Multiplexed codon mutagenesis

The library generation and sequencing workflow is outlined in Fig. 1. First, previously described primer design software[55] was used to generate a pool of 60mer oligonucleotides, consisting of 928 subpools, each including an NNS codon at one of the 928 codons of the extracellular receptor, flanked by 28 and 29 bp of template-homologous sequence before and after the mismatched codon, respectively (Supplementary Data 6). The NNS codon encodes all 20 amino acids and one stop codon. To minimise unevenness of mutagenesis, oligonucleotides were organized into 47 pools, each covering 20 sequential codons, and separate reactions were run for each. Each oligonucleotide mixture was phosphorylated by adding 20 ul 10 uM oligonucleotides to 2.4 µl T4 Polynucleotide Kinase Buffer, 1 ul 10 mM ATP, 1 ul T4 Polynucleotide Kinase (10 U/µL) into a PCR tube and incubation at 37 °C for 60 minutes. Phosphorylated oligos were stored at −20 °C and diluted 1:300 in nuclease-free H$_2$O on the day of mutagenesis. 7 µl 100 µM universal secondary primer (VA11) was phosphorylated similarly in a final reaction volume of 30 µl, and diluted 1:20 on the day of mutagenesis.

We applied a reported nicking mutagenesis strategy[25] to construct the *INSR* variant library. The barcoded attB_INSR_P2A_PuroR plasmid library was freshly purified from a bacterial pellet using the Qiagen Mini-Prep Kit (Cat. 27106 × 4) for mutagenesis before column purification, elution in 10 µL nuclease-free H$_2$O and transformation into 100 ul XL10-gold ultracompetent cells (Agilent, 200315). Transformation mix volumes were adjusted to 850 µL with sterile LB medium and

800 µL spread on two 150 mm agar plates containing 100 mg/ml Ampicillin. Serial dilutions from the remaining 50ul reaction mixture were prepared to permit calculation of transformation efficiencies. After overnight incubation at 37 °C, colonies were scraped into 10 mL LB at an estimated bottleneck of 130,000 (whole library) or 2750 (individual mutagenesis blocks) colony forming units to limit library size. 3 ml scraped cell suspensions were prepared using a Qiagen Mini-Prep Kit and 2ug purified plasmid DNA from each block was pooled to prepare the final barcoded, mutagenised INSR library.

## PacBio sequencing of INSR variant library for barcode-variant mapping

PacBio sequencing was used to phase barcodes and mutations through long sequence reads spanning barcode and INSR extracellular domain sequence. To eliminate strand exchange during PCR, PacBio sequencing templates were prepared from purified plasmid library by SbfI-HF and PspOMI digest. Digested product containing 3 kb template fragment (barcode and ectodomain sequence) and 5 kb plasmid backbone was cleaned using the QIAquick PCR Purification Kit and submitted to University of California, Davis PacBio Sequencing Services for size selection with BluePippin (Sage Science) followed by library prep and sequencing on a single SMRT cell with Sequel II, using 30-hr movie collection times. The Sequel II system performs on-instrument data processing and delivers accurate long reads (HiFi reads) through building circular consensus sequences (CCSs) from subreads (raw sequencing files available in GEO; accession number GSE277112). We processed HiFi reads using alignparse[56,57] version 0.6.3, which applies minimap2[58] version 2.24 for long read alignment to identify and call mutations in the INSR sequence and to phase them with barcodes. 3,568,566 of 4,016,907 HiFi reads (89%) mapped to the target (available in GEO; accession number GSE277112), and the output was used to generate a codon-variant lookup table. We first retained mapped HiFi reads with sequencing accuracy reported by the PacBio ccs program in both INSR ectodomain sequence and barcode to be at least 99.99%. 70% of all reads passed these filters. As PacBio sequencing produced on average 19 reads per amplicon (Supplementary Fig. 1A), we next calculated the empirical accuracy of retained reads[57] to assess the reliability of barcode-variant phasing, i.e., how often the reads linked with a specific barcode are identical. We observed 99.4% empirical accuracy for each HiFi read, excluding reads with indels. Since 96% of the barcodes were associated with two or more HiFi reads we generated a cumulative consensus of HiFi reads within these barcodes, so that consensus accuracy would exceed calculated empirical accuracy for individual reads. Barcodes differing by only two bases but associated with different variants were omitted from the codon-variant table. Selection criteria were designed to exclude barcodes with lower CCS read counts, while retaining those with higher read counts. As a result, 52 conflicting barcodes, mostly with single reads, were removed. Barcodes associated with insertions, deletions or more than one variant were also excluded. Overall 126,272 barcodes were retained, of which 80,972 associated with 15,996 missense variants, 3539 barcodes with 813 nonsense variants, 4,023 and 41,232 barcodes with synonymous variants and wild type sequence, respectively (Supplementary Fig. 2B-D; Supplementary Table 1). The code used to generate the final barcode-variant table is available on (https://doi.org/10.

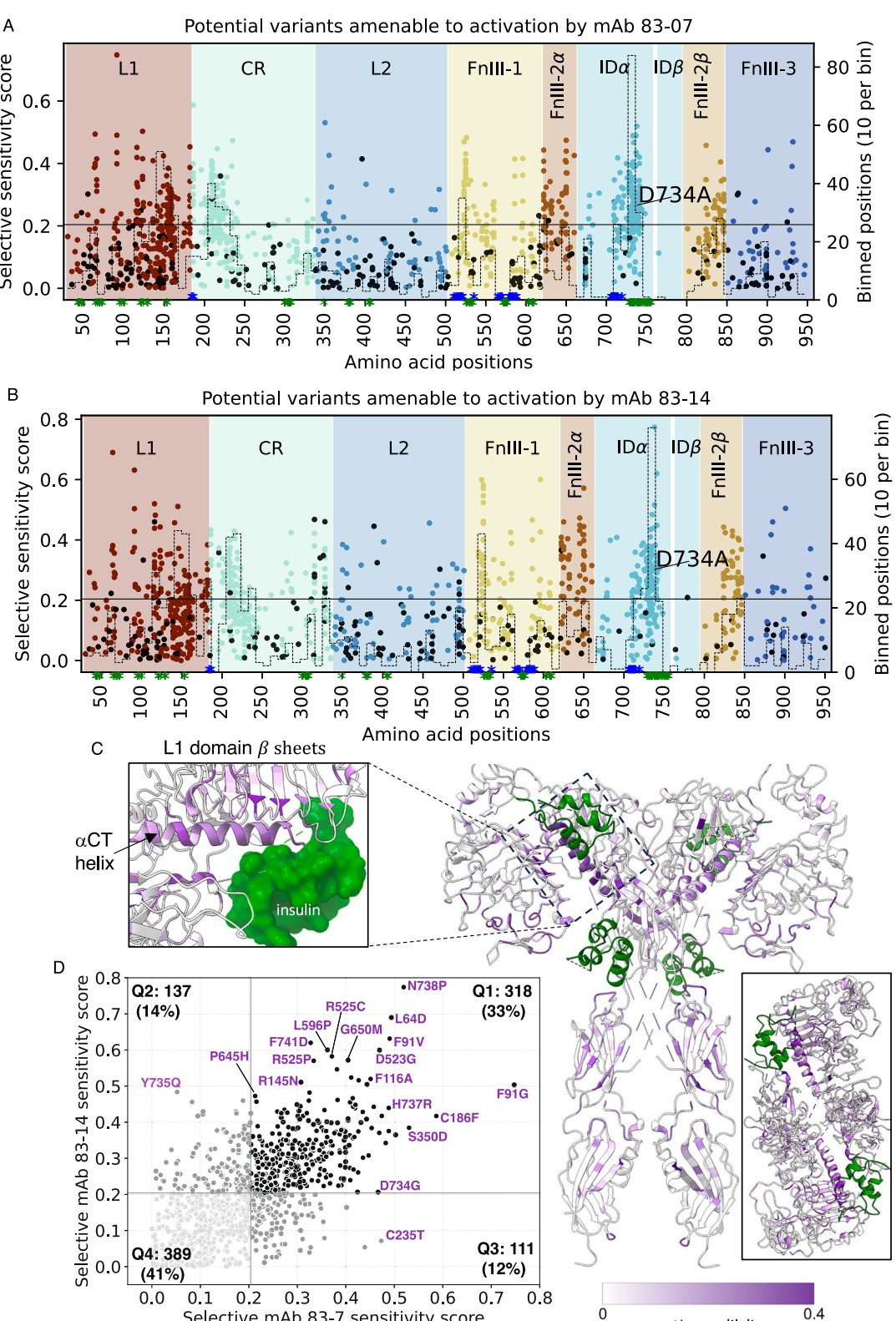

**A** Potential variants amenable to activation by mAb 83-07

**B** Potential variants amenable to activation by mAb 83-14

**C** L1 domain β sheets

**D**

Generation of R-MmINSR KD cells containing single landing pads

Murine embryonic fibroblasts (MEFs) derived from the *Igf1R⁻ʹ⁻* mouse line[54] were infected at low multiplicity of infection with lentivirus

encoding concatenated shRNAs targeting the *Insr* gene[13], and exposed to 500ug/ml hygromycin-B for 2 weeks. Clonal cell lines were isolated by limiting serial dilution and screened for GFP expression upon doxycycline (DOX) addition. Four clones demonstrated strong inducible knockdown of endogenous *Insr*, one of which was used for all subsequent experiments (Supplementary Fig. 3).

5281/zenodo.16788684) and includes additional plots illustrating library structure.

**Fig. 7 | Location of INSR variants with impaired insulin binding amenable to monoclonal antibody activation. A, B** Manhattan plots displaying differential signalling scores for (**A**) mAb 83-7 and (**B**) mAb 83-14 compared to insulin, for all variants whose mutation severely impairs insulin signalling. Only loss-of-signalling variants with antibody signalling results from all three replicates and which scored in all five insulin signalling replicates are plotted. Black dots denote variants activatable only by the single mAb plotted. The horizontal line represents a selective sensitivity score of +0.204. D734A, a variant previously extensively studied, is labelled. Green or blue stars on X axes indicate site 1 and site 2 residues respectively. Areas identified on variant effect maps as showing selective activation by antibodies are boxed for reference, and a frequency histogram indicating counts of potential target variants in a sliding 10 residue window is superimposed. **C** 4 insulin-bound INSR structure (PDB: 6PXV), coloured by mean sensitivity scores for mAb 83-7 and mAb 83-14. Insulin molecules are green. The left hand box shows a zoomed in view of insulin bound to binding site 1, and the right hand box shows the view of the structure from above. **D** Scatter plot comparing selective sensitivity score for mAbs 83-7 and 83-14, divided into quadrants according to the arbitrary threshold score +0.204 used in (**A, B**). Selected variants are labelled, and proportions of variants in each quadrant are indicated.

A previously described DOX-inducible BxB1 DNA recombinase landing pad (Tet-coBxb1-2A-BFP_IRES-iCasp9-2A-Blast_rtTA3, Addgene 171588) was introduced into R-MmINSR KD cells using the Lenti-X Packaging Single Shot (VSV-G) (Cat 631275) system. Briefly, 7 µg of pLenti_Tet-coBxb1-2A-BFP_IRES-iCasp9-2A-Blast_rtTA3 lentiviral vector in 600ul water was mixed with Lenti-X Packaging plasmid and incubated at room temperature for 10 minutes. The mixture was transferred dropwise onto cultured Lenti-X 293 T cells on a 10 cm plate and incubated at 37 °C in 5% CO2. Medium was changed the next day and supernatant collected after 48 and 72 hrs and pooled. Collected medium was centrifuged ($300 \times g$, 5 min), and the supernatant filtered through a 0.45µm filter to remove debris. The landing pad construct expressed doxycycline-inducible blue fluorescent protein (BFP) and a Blasticidin resistance gene, which were used to confirm landing pad insertion. 100 µl to 1 ml lentiviral supernatant was used, and assessment of BFP expression 48 hr after infection was used to identify lines with an MOI < 1. After treatment with 6 µg/ml Blasticidin for one week, cells with the highest BFP fluorescence were sorted into single cells in 96-well plates using a BD FACS Aria II (405-450/50 nm laser). Surviving clones were transferred into a 24-well plate and later 6-well plates. 400,000 of the 3 selected clonal lines were seeded on a 6 well plate and transfected with 2.5ug of attB-miRFP670 using Lipofectamine 3000. The medium was changed and 2ug/ml Puromycin added 48 hr after transfection. Cells were assessed for recombination after a week of antibiotic selection with a BD LSR Fortessa flow cytometer. One clone showing uniform loss of BFP and gain of miRFP670, indicative of a single landing pad was selected for use in subsequent experiments.

### Transfecting cells with the INSR mutation library

Landing pad-containing R-MmINSR KD cells at 80% confluence were trypsinised and pelleted at $300 \times g$ for 4 minutes at room temperature, washed with 20 mL PBS, counted, and resuspended in OptiMEM at $1.17 \times 10^8$ cells/mL. In a separate tube, 3 ml of cells ($3.5 \times 10^8$ cells) were mixed with 2.8 mg WT plasmid or plasmid library resuspended in 500ul nuclease-free $H_2O$. The mix was loaded on a MaxCyte CL-1.1 processing assembly (PA) using a 10 ml luer-lock syringe, and electroporated twice using the MaxCyte GTx 0-6 setting. The electroporated cells were immediately unloaded using a syringe and the PA washed twice with 3.5 ml OptiMEM to collect remaining cells. Cells were allowed to recover at 37 °C for 40 minutes before 75 mL warm growth media was added. The cell suspension was introduced to 15 5-layer T175 flasks to a density of $4.7 \times 10^8$ cells/175 cm2 and incubated under standard conditions. Selection with 1.5ug/ml puromycin was started 48 ho after transfection and the medium was changed every 2-3 days until colonies appeared and grew to high confluency for harvesting. Harvested cells from all flasks were pooled and aliquoted for cryopreservation.

### Validation of cellular model

To confirm efficient induction of expression of myc-tagged human INSR on exposure to doxycycline, and doxycycline-induced knockdown of endogenous mouse Insr, R- (Igf1r$^{-/-}$), R-MmInsrKD, and R-MmInsrKD + HsINSRmyc cells were cultured in the presence or absence of 1ug/ml Doxycycline for 3 days at 37 °C/5%CO2. Cells were washed, snap frozen, and lysed on ice in lysis buffer (20 mM HEPES, 150 mM NaCl, 1.2 mM MgCl2, 1 mM EGTA, 1 mM PMST, 1 mM Na3VO4, 10% v/v glycerol, 1% v/v Triton-X-100, Roche complete-EDTA protease inhibitors). Insoluble material was pelleted by centrifugation (10,000xg for 10 min at 4 °C) and supernatant quantified by BCA assay (ThermoFisher). 15 µg lysate per lane was resolved on NuPAGE 4-12% Bis-Tris gels (ThermoFisher) and transferred to nitrocellulose by iBlotII (ThermoFisher). Membranes were blocked with 3% w/v BSA/TBST for 1 hr at room temperature before overnight incubation at 4 °C with primary antibodies from Cell Signalling Technologies (3025, 2276, 9750, 4967). Membranes were washed four times with 1xTBST prior to incubation with horseradish peroxidase (HRP)-conjugated secondary antibodies (Cell Signalling Technologies: 7076, 7074). Immobilon Western Chemiluminescent HRP substrate (Millipore) was used to detect protein-antibody complexes and grey-scale tag image formats (TIFFs) captured utilising an iBright FL1500 Imaging System (ThermoFisher).

### Preparing fluorescent insulin and antibody conjugates

We used the Alexa Fluor 647 NHS Ester (Thermofisher, A37573) kit to label recombinant human insulin (Sigma, 91077 C). Immediately before use, the amine-reactive compound was dissolved in anhydrous dimethylsulfoxide (DMSO) to 10 mg/mL and insulin in 0.1 M sodium bicarbonate buffer [pH 8.3] to prepare 10 mg/ml insulin solution. 100ul reactive dye solution was slowly added to 266ul of this insulin solution and the reaction incubated for 1 h at room temperature with continuous stirring. Labelled insulin conjugate was recovered by dialysis using Slide-A-Lyzer MINI Dialysis Devices, 3.5 K MWCO (Thermofisher, 69550).

83-7 and 83-14 antibodies (gifts from Prof. Kenneth Siddle) were labelled with AF647 dye using Zip Alexa Fluor Rapid Antibody Labelling kit (Thermofisher, Z11235) following the manufacturer's protocol.

### Multiplexed assay of insulin or antibody binding

Cells were recovered from liquid nitrogen, expanded for at least two passages, washed twice with PBS and serum starved (DMEM without FCS) for 16 h. $1.5 \times 10^8$ cells were used for each replicate. Cells were washed once with PBS, detached with trypsin, quenched with an equal volume of growth medium, and centrifuged at $400 \times g$ for 4 minutes before washing with FACS buffer and resuspension at $1 \times 10^7$/ml in FACS buffer. Cells were incubated with AF647_insulin (100nmol/L, AF647_83-7 or AF647_83-14 (1 mg/ml) conjugates for 1 h on ice with periodic mixing. Stained cells were pelleted, washed 4 times with FACS buffer and fixed in 3% methanol-free PFA (43368, ThemoFisher) for 10 minutes on ice. Finally, cells were pelleted, washed once with PBS, and resuspended in PBS before sorting on a FACS Aria II within 48 h.

### Multiplexed assay of insulin- or antibody-induced Akt phosphorylation

Cells were prepared as for binding assays until quenching of trypsinised cells with an equal volume of warm growth medium this time containing 100 nM human recombinant insulin (Sigma, 91077 C) or

10 nM anti-INSR antibody. After 10 minutes incubation at 37 °C, cells were centrifuged at $400 \times g$ for 4 minutes and washed with FACS buffer before fixing with 4% methanol-free PFA (Thermofisher, 28908) for 10 minutes on ice with periodic mixing. Fixed cells were washed once with FACS buffer and resuspended in Fc block (FACS buffer with 10% FCS) containing 0.1% saponin and incubated on ice for 15 minutes to permeabilise cells and block Fc receptors. Fixed cells were spun, resuspended at $10^7$ cells per ml in FACS buffer containing 0.1% saponin, and stained with AlexaFluor 647-conjugated anti-Phospho-Akt (Ser473/474) antibody (Cell Signalling, Cat No. 4075, Lot 22) at 1/200 dilution for 1 h on ice with periodic mixing. Stained cells were pelleted, washed 4x with FACS buffer and fixed with 3% methanol-free PFA for 10 minutes on ice. Finally, cells were pelleted, washed once with PBS, and resuspended in PBS before sorting on a FACS Aria II within 48 hours.

### Validation of FACS-based assays of INSR function

To validate FACS-based assays of INSR expression and function, we used non transfected cells, and cells transfected with either wild-type (WT) INSR or the mutant INSR library. Both monoclonal antibodies used robustly detected surface INSR expression in WT and library cells compared to untransfected cells (Fig. 1C, and S1F). Similarly, binding of labelled insulin produced comparable profiles for WT and library cells which were clearly distinct from the profile of untransfected cells, although some insulin binding was detected in control cells, consistent with low level residual expression of murine Insr.

A smaller dynamic range was observed in assays for phosphorylated AKT (pAKT) (Fig. 1C, Supplementary Fig. 2E), as expected for a labile intracellular antigen. Shifts in pAKT FACS profile were seen on stimulation by insulin, 83-7 or 83-14 antibodies for both WT and library cells, but some pAKT was also detected on insulin stimulation of untransfected cells. The lack of pAkt detection in untransfected cells stimulated with human INSR-specific antibodies further supports this being due to binding to endogenous mouse Insr, in keeping with the residual insulin binding noted above. We also observed that overexpression of either WT or mutated human INSR induced some AKT phosphorylation even without stimulation. However, despite these limitations of the cellular system used, we consistently observed much stronger binding of and signalling by insulin and antibodies than background.

### Variant library FACS sorting and barcode detection

A BD FACS Aria II sorter was gated for singlets, and cells were sorted into four bins by fluorescence in the 640-670/14 A channel. At least 13 million cells were collected per bin. Sorted cells were collected by centrifugation and stored at −20 °C before gDNA extraction and library preparation.

Genomic DNA was prepared from cells in each bin by phenol-chloroform extraction. Cell pellets were resuspended in 500 µl lysis buffer (100 mM Tris [pH 8.5], 0.5 mM EDTA, 0.2% SDS and 200 mM NaCl, 25ug RNAase A) and incubated at 37 °C for 1 h before adding 700ug proteinase K (New England Biolabs, P8107S) and incubated at 56 °C overnight. An equal volume of Phenol:Chloroform:Isoamyl (PCI) alcohol 25:24:1 was added to each reaction, vortexed vigorously and centrifuged at high speed for 5 minutes. The aqueous layer was transferred to a clean 1.5 ml tube, mixed with equal amount of PCI, vortexed vigorously and centrifuged at high speed for 5 minutes. The aqueous layer was again transferred into a new tube and DNA precipitated with ethanol and resuspended in 400ul nuclease-free $H_2O$. Because extracted gDNA was viscous, samples were sonicated on a Diagenode Bioruptor Plus on high power with 30 seconds on/off for 5 cycles.

PCR was performed to amplify gDNA barcodes. Optimisation of DNA preparation and PCR was required to enable efficient amplification of barcodes even from signalling assays, which employed PFA fixation prior to FACS. We used overnight proteinase K digestion to break down crosslinked proteins, and traditional Phenol:Chloroform:Isoamyl alcohol (25:24:1) extraction rather than kit-based extraction. Phenol helps to denature proteins further, releasing DNA, while the short PCR amplicon of 120 bp also increased likelihood of amplification. This approach led to a reliably improved PCR yield. To avoid amplifying barcodes from any persisting extragenomic INSR plasmid, the reverse primer was designed to anneal to a gDNA sequence downstream from the plasmid integration site. For every bin, we distributed purified gDNA across multiple 25 µL PCR reactions. Each included 12.5 µL Phusion high-fidelity PCR master mix with HF buffer (NEB, M0531S), 500 ng genomic gDNA, 10 µM reverse primer with Illumina P5 sequence (VA12), and 10 µM indexed forward primer including an Illumina P7 sequence (VA13) binding upstream from the barcode. PCR began with a step of 96 °C for 2 minutes and ended with 72 °C for 1 minutes, with 24 amplification cycles (96 °C for 30 seconds; 65 °C for 30 seconds; 72 °C for 40 seconds) in between.

PCR products for each bin were pooled and 300ul purified with MinElute PCR Purification Kit and resuspended in 50ul of water. The eluate was size selected (expected size of 283nts) on a 2% agarose gel (Monarch DNA Gel Extraction Kit, T1020S) and quantified by Bioanalyser DNA1000 chip. Quantified samples were mixed together in equimolar ratios and submitted for 50 bp single end sequencing using a custom sequencing primer (VA14) on an Illumina NextSeq. Demultiplexed reads were trimmed using trimmomatic software such that only 30nt barcode sequences was left. Sequences then aligned to the barcode sequence library determined from PacBio sequencing by Enrich2, yielding a count of the number of times each library barcode was sequenced in each FACS bin. Raw illumina sequencing and processed files are available on GEO; accession number GSE277112. Scripts are available at (https://zenodo.org/records/16788684).

### Generating G333Q, D627A and D743A single substitution cell lines

Primers (Supplementary Data 6) were designed to generate substitution mutations in attB_INSR_P2A_PuroR plasmid using the Q5 Site-Directed mutagenesis kit following manufacturer's recommendations. Products were amplified in *E. coli* strains and substitutions confirmed by whole plasmid sequencing. 10ug of variant plasmids were transfected into to 10*10^6 landing pad-containing R-MmINSR KD cells using MaxCyte PA R-50 × 3 before selection and study by flow cytometry.

### In vitro assay of insulin binding to G333Q and D627A mutant receptors

To generate europium-labelled insulin, 7 mg Insulin (200 IU Actrapid, Novo Nordisk) was dialysed using 3.5MWCO Slide-A-Lyzer mini dialysis devices (ThermoFisher, 88400) against three buffer changes of 50 mM bicarbonate buffer, pH9.6 over 16 h at 4 °C. Dialysed Insulin was then concentrated utilising 3 K MWCO protein concentrator (Thermo-Fisher, 88515) according to the manufacturer's protocol. 2 mg concentrated, buffer-exchanged insulin was then labelled with europium using the DELFIA Eu-labelling kit (0.4 mg, Revvity Health, 1244-302), again per manufacturer's protocol. To separate labelled insulin from unreacted Eu-chelate, NAP-5 columns containing Sephadex G-25 resin were used with 50 mM Tris-HCl pH 7.8 containing 150 mM NaCl, and 0.05% sodium azide as the elution buffer.

To enable capture of myc-tagged insulin receptors, white Lumitrac 600 96-well plates (Greiner, 655074) were coated overnight at 4 °C with 20ng/well anti-myc antibody clone 4A6 (Merk, 05-724) diluted in 50mM bicarbonate buffer, pH 9.6. Plates were then blocked for 2 hr at room temperature with 0.5% Bovine serum albumin (BSA, Sigma Aldrich A7030) diluted in 1x Tris buffered saline (TBS) containing

0.05% Tween-20 (1x TBST). R-MmINSRKD cells expressing either WT, G333Q, or D627A mutant INSR were grown to confluence in T75 flasks prior to 5 h serum-starvation. Serum-starved monolayers were washed in PBS and lysed by the addition of 5 ml/T75 flask lysis buffer (20 mM HEPES, 150 mM NaCl, 1.2 mM $MgCl_2$, 1 mM EGTA, 1 mM PMST, 1 mM $Na_3VO_4$, 10% v/v glycerol, 1% v/v Triton-X-100, Roche complete-EDTA protease inhibitors, Roche phosSTOP phosphatase inhibitors) and incubated on ice for 1 hr to solubilise membranes. Cellular debris was pelleted by centrifugation at 8,000xg for 15 min at 4°C and 100 μl/well supernatant added to the anti-myc antibody coated 96 well plate. Receptors were captured overnight at 4°C. Plates were washed four times with 1xTBST before adding increasing concentrations of Eu-labelled insulin at 100 μl/well and incubating overnight at 4°C. Unbound Eu-insulin was removed by washing six times with 1xTBST. Bound Eu-insulin was detected by adding 100ul/ well DELFIA Enhancement Solution (Revvity Health, 1244-104) and recording time-resolved fluorescence after 30 minutes using a Tecan Infinite 200 Pro plate reader (340 nm excitation, 612 emission, 400μs lag time).

## Low throughput assay of receptor signalling by Western blotting

R-MmINSRKD cells expressing WT, G333Q, or D627A mutant INSR were serum-starved for 5 hr prior to stimulation with indicated concentrations of insulin for 10 min at 37°C/5%CO2. Cells were washed, snap frozen, and lysed on ice in lysis buffer (20 mM HEPES, 150 mM NaCl, 1.2 mM $MgCl_2$, 1 mM EGTA, 1 mM PMST, 1 mM $Na_3VO_4$, 10% v/v glycerol, 1% v/v Triton-X-100, Roche complete-EDTA protease inhibitors, Roche phosSTOP phosphatase inhibitors). Insoluble material was pelleted by centrifugation (10,000xg for 10 min at 4 °C) and supernatant quantified by BCA assay (ThermoFisher). Lysate, 10 μg/lane, was resolved on NuPAGE 4–12% Bis-Tris gels (ThermoFisher) and transferred to nitrocellulose by iBlotII (ThermoFisher). Membranes were blocked with 3% w/v BSA/TBST for 1 hr at room temperature before overnight incubation at 4 °C with primary antibodies. Primary antibodies were obtained from Cell Signalling Technologies (catalogue numbers: 3024, 3025, 4060, 2920, 5726, 9102). Membranes were washed four times with 1xTBST prior to incubation with horseradish peroxidase (HRP)-conjugated secondary antibodies (Cell Signalling Technologies, catalogue numbers: 7076, 7074). Immobilon Western Chemiluminescent HRP substrate (Millipore) was used to detect protein-antibody complexes and grey-scale tag image file formats (TIFFs) captured utilising an iBright FL1500 Imaging System (ThermoFisher). ImageJ v1.54 f was used to perform densitometry analysis of pixel density within the TIFF images.

## Quantification and statistical analysis

**Calculation of variant function scores and false discovery rates.** The puromycin selection marker is located downstream from the INSR gene, with a P2A self-cleaving sequence in between. Any stop codons in the INSR gene would stop translation of the Puromycin resistance gene and therefore cells carrying stop codons will be removed during antibiotic selection after transfection unless stop codon readthrough occurs. We expected no reads for such barcodes on Illumina sequencing but observed reads for 7% of the possible stop codon barcodes. We considered the possibility that these barcodes represent erroneous barcode-variant assignments in PacBio data analysis. To test this, we compared the counts of Illumina sequencing reads representing different classes of variants in FACS based assays. We found that barcodes associated with stop codons show greatly depleted read counts in the FACS data, compared to barcodes associated with wild-type, synonymous or missense variants. This strongly suggests that these barcodes represent bona fide stop codons that were only partially depleted during puromycin selection, perhaps thanks to stop codon

readthrough, and supports the accuracy of our barcode-variant phasing approach.

We used Enrich2[59] to count number of occurrences of each barcode in each bin. Enrich2 configuration files are available at (https://doi.org/10.5281/zenodo.16787091). We then used a previously published method for calculating barcodes and variant scores with some modifications[60]. The count for each barcode in a bin was divided by the sum of counts recorded in that bin to obtain the frequency of each barcode ($F_b$) within that bin. This calculation was repeated for every bin in each replicate experiment.

$$F_{b,binN} = \frac{C_{b,binN}}{\sum C_{binN}} \quad \text{for } N = 1 \text{ to } 4 \tag{1}$$

Barcodes with fewer than 150 reads across 4 bins in each experiment were filtered out. Next, for each barcode and each replicate experiment, a barcode score ($S_b$) was calculated using the following equation:

$$S_b = \frac{(F_{b,bin1}\,0.25) + (F_{b,bin2}\,0.5) + (F_{b,bin3}\,0.75) + (F_{b,bin4}\,1)}{(F_{b,bin1} + F_{b,bin2} + F_{b,bin3} + F_{b,bin4})} \tag{2}$$

Finally, for each replicate experiment, a variant score ($S_v$) was calculated by dividing the mean barcode score corresponding to that variant by the mean barcode score of wild-type INSR, as follows:

$$S_v^{replicate} = \frac{\bar{S}_{b,v}}{\bar{S}_{b,wt}} \tag{3}$$

After obtaining the variant scores ($S_v^{replicate}$) for all replicates of a given experimental condition, the mean log2 variant score ($\bar{S}_v$) was calculated across replicates:

$$\bar{S}_v = \log2 \left( \frac{1}{n} \sum_{i=1}^{N} S_v^{replicate} \right) \tag{4}$$

Here, $N$ is the number of replicates. $\bar{S}_v$ score used for all plotting.

To quantify confidence that a variant's score differs from the score of wild-type INSR, we used a bootstrapping approach to calculate, for each variant, a false discovery rate (FDR). This approach assigns FDR scores to variants associated with any number of barcodes, while avoiding assumptions about the distribution of scores and their measurement errors.

For variant $v$ with $k$ barcodes across all biological replicates, barcode score $B_v$ was calculated for each variant as:

$$B_v = \frac{1}{k} \sum_{b=1}^{k} S_b \tag{5}$$

We then generated weighted averages of 10,000 randomly sampled sets of $k$ wild-type barcodes ($B_{wt}$) and calculated the wild-type barcode score $B_{wt}$ of each set as follows:

$$B_{wt,i} = \frac{1}{k} \sum_{b=1}^{k} B_{wt} \quad \text{for } i = 1 \text{ to } 10,000 \tag{6}$$

Finally, we computed the false discovery rate ($FDR_v$) for variant $v$ from:

$$FDR_v = \begin{cases} N(B_{wt} < B_v)/N(B_{wt} < 0)\, if\, B_v < 0 \\ \\ N(B_{wt} > B_v)/N(B_{wt} > 0)\, if\, B_v > 0 \end{cases} \tag{7}$$

Where $N(B_{wt} < x)$ is the number of times $B_{wt} < x$ across the set of 10,000 bootstrapped wild-type scores. A pseudocount of 1 is added if $N(B_{wt} < B_v)$ or $N(B_{wt} > B_v)$ is equal to 0.

**Computation of biallelic function scores.** Nonsense, frameshift or whole exon deletions, for which no empirical function scores were generated, were assumed to be complete or near complete LoF variants, and were thus assigned the lowest high confidence score seen in the MAVE (for W516R); intracellular mutations, whether missense, frameshift or nonsense, were assigned this score multiplied by 1.5, representing a punitive factor to reflect the well documented dominant negativity of such mutations[61].

**Regional comparison of variant scores between different assays.** To compare consequences for different functional readouts of mutating different regions of the receptor, we applied the Wilcoxon signed-rank test. Prior to statistical comparison, scores from each dataset, which show different ranges between assays, were normalized using the empirical cumulative distribution function (ECDF), which transforms scores into quantile ranks (ranging from 0 to 1), enabling robust, distribution-free comparisons between scores from different assays. We aligned the two datasets being compared, matching amino acid substitution and position, before applying the Wilcoxon signed-rank test to a 15-residue sliding window. Windows were tested only if they contained at least five variants. The test assessed whether the distribution of ECDF-normalized scores in condition A (e.g., insulin binding) was significantly lower than in condition B (e.g., expression), i.e., whether mutations had more deleterious effects in assay A than assay B, as indicated in figure legends. To aid interpretability, only one-sided tests were used (alternative = 'less'), focusing on reduced variant tolerance in the first dataset. The resulting $-\log_{10}$(p-value) values were plotted against the window centre position to visualize regions of statistically significant sensitivity shifts. Regions for which p < 0.05 are shown as shaded areas on the plots.

**Visualisation and interrogation of function scores**
Fully analysed data are interrogatable in the curated MaveDB community database of MAVE data. Data can be found by searching for *INSR* from the MaveDB landing page (https://www.mavedb.org/). The Universal Resource Name (URN) for the dataset is urn:mavedb:00001239-a. Clicking on this URN leads to a project summary page that includes detailed metadata and links to individual score sets, each representing a distinct assay. Within each score set page, a heatmap is displayed showing variant scores spanning residues 28 to 955. In this heatmap, blue indicates variants with scores lower than wild type (WT), red indicates higher-than-WT scores, white represents scores approximately equal to WT (near zero), yellow marks the WT amino acid at each position, and black denotes positions for which no data are available. Above the heatmap, a histogram illustrates the distribution of scores for the corresponding assay. By default, MaveDB sets the midpoint of its heatmap colour scale to the median of the score distribution and interprets this value as the WT. Variants are then coloured on a red-white-blue gradient relative to this median. However, because our score distributions are asymmetric around zero (defined as WT), this default behaviour distorts the heatmap representation. To ensure accurate visualization, we rescaled the positive variant scores to match the scale of the negative scores, forcing the median to align with our WT at zero. This adjustment ensures scores below zero are correctly coloured blue, and scores above zero are consistently coloured red. To retrieve the score for a specific variant, users can enter the mutation using the three letter amino acid code (e.g., Cys35Lys) into the query box. This highlights the corresponding position in both the heatmap and distribution plot and displays its exact score. Each score set page also provides downloadable versions of the heatmap, histogram, and the full score file. The score files include not only the variant effect scores but also associated false discovery rate (FDR) values and replicate scores. Supplementary Table 5 provides a summary of the score set URNs, their corresponding assays, and direct links (active as of July 2025).

**Reporting summary**
Further information on research design is available in the Nature Portfolio Reporting Summary linked to this article.

## Data availability
The processed PacBio and illumina sequencing data are available at (https://zenodo.org/records/16787091). Raw PacBio and illumina sequencing data generated in this study have been deposited in the GEO database under accession code GSE277112. Fully analysed data are also interrogatable in the curated MaveDB community database under accession code 00001239. Source data are provided with this paper.

## Code availability
The code utilized in this study can be found at (https://github.com/vaslanzadeh/INSR_MAVE[62]).

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

## Acknowledgements

RKS and GK are supported by the Wellcome Trust [grants 210752 and 207507]. RKS is additionally supported by the BHF Centre for Research Excellence Award III [RE/18/5/34216], and GK by UK Medical Research Council (MRC) University Unit programme MC_UU_00035/8. GVB & RKS are supported by Diabetes UK [grant 22/0006407]. CV has been funded via the PRISIS rare disease reference center, funded by the French Ministry of Health and Assistance-Publique Hôpitaux de Paris, and is a member of the European Reference Network on Rare Endocrine Conditions (EndoERN Project ID n° 739527). We are grateful to Ben Livesey and Joe Marsh (University of Edinburgh/MRC Human Genetics Unit) for advice on ROC analysis methodology, and thank Elisabeth Freyer and Michael Rennie (Flow cytometry facility, University of Edinburgh Institute of Genetics and Cancer) for support with extensive cell sorting. We thank Stephen O'Rahilly (University of Cambridge/MRC Metabolic Diseases Unit), Olivier Lascols (Department of Molecular Biology and Genetics, Saint-Antoine Hospital, Assistance Publique-Hôpitaux de Paris, Paris, France) and Drs Philip Gorden and Rebecca Brown (National Institute for Diabetes, Digestive and Kidney Disease, Bethesda, Maryland, USA) for access to *INSR* variant details and associated clinical data from their respective centres, and all clinicians referring patients for genetic studies. Supplementary Fig. 1B was generated using UCSF ChimeraX, developed by the Resource for Biocomputing, Visualization, and Informatics at the University of California, San Francisco, with support from National Institutes of Health R01-GM129325 and the Office of Cyber Infrastructure and Computational Biology, National Institute of Allergy and Infectious Diseases.

## Author contributions

Conceptualization: V.A.,R.K.S.,G.K.; Data curation: V.A., R.K., H.Ç., G.V.B., R.K.,C.V.; Formal analysis: V.A., G.V.B.; Funding acquisition: R.K.S.,G.K.; Investigation: V.A.; Methodology: V.A.,G.V.B.; Project administration: V.A.,R.K.S.; Resources: V.A.,R.K.S.,G.K.,G.V.B.,K.M.; Supervision: R.K.S.,G.K.; Validation: V.A.,G.V.B.; Visualization: V.A.,G.V.B.; Writing—original draft: V.A.,R.K.S.,G.K.,G.V.B.; Writing—review and editing: all co-authors.

## Competing interests

R.K.S. has received consulting fees from Novartis, Astra Zeneca, and Alnylam, research contribution in kind from Pfizer, and speaking fees from Novo Nordisk, Eli Lilly, and Amryt. CV serves as investigator in the APL-22 clinical study sponsored by Chiesi Farmaceutici and in the REGN4461-PLD-20100 sponsored by Regeneron Pharmaceuticals, and has served as speaker and received support for attending meetings from Amryt Pharmaceuticals (now Chiesi Farmaceutici) and Sanofi. The remaining authors declare no competing interests.
