## [Transparent Peer Review file · Nature Communications]

Deep mutational scanning of the human insulin receptor ectodomain to inform precision therapy for insulin resistance

Corresponding Author: Professor Robert Semple

Version 0:

Reviewer comments:

Reviewer #1

(Remarks to the Author)

This manuscript describes a deep mutational scanning study of the human insulin receptor ectodomain, with the aim of providing data that have the potential to inform both clinical diagnosis as well as therapeutic development. The study is timely in that it complements the recent advances in structural studies of the receptor ectodomain and the associated mechanisms of receptor activation by insulin. To aid their study, the authors have also undertaken a comprehensive collation of known human insulin receptor mutations that will be of benefit to those working in the broader field.

I find the study to be on-the-whole well-presented and carefully undertaken, but there are shortcomings that I would require to be addressed prior to any acceptance for publication. These are of varying importance and are presented below in no particular order:

Extended Data Figure 2: the panel captions do not match the panels themselves, and likewise the references to panels in this figure in the main text appear incorrect. This makes it difficult to disentangle, and there is likely a panel missing from this Figure.

I am also concerned about the authors' use of incredibly small font in some of the Figures, there may be editorial requirements in this regard.

Glycosylation. The authors do not discuss the impact of variants on receptor glycosylation. Obviously, mutation of residues involved in N- and O-linked glycosylation (either the residues that form the linkage to the sugar or those that form part of the associated motif) will remove glycosylation at the relevant site. It would be of interest to see whether, for example, mutation of the asparagine residues involved in N-linkage of sugars has impact on folding and/or insulin binding and whether such impact concurs with literature in this regard.

L81. The authors should also reference the class of receptor activating peptides pioneered by Schäffer et al. (2003, <https://doi.org/10.1073/pnas.0830026100>), which have also more recently been subject to structural studies to elucidate their mode of action. The latter could also be referenced at L365 (p15).

L99. For clarity, should state "...from the start of the first codon of the mature receptor to the start of the transmembrane domain."

Extended Data Figure 1A. There is something amiss in coloring of the schematic on the right-hand side of this panel. In particular, the two FnIII-2 domains (in the middle of the structure, second domains up from the bottom) show a multitude of colors and not the brown color indicated on the left-hand side. There are also "brown" strands within the yellow FnIII-1 domains. Furthermore, there is no indication in the panel or in its caption that the four insulins are colored green. I recommend that the panel be redone in clearer and more contrasting colors. Also, in the left-hand component of the panel, the authors should label as such the respective α - and β -chains, to assist subsequent interpretation of Fig. 2A, as well as label the chain components as ID α , ID β , FnIII-2 α and FnIII- β for consistency with the text.

Extended Data Figure 1C. In the version that I downloaded, Panel C is of unacceptably low resolution (cf. Panels B, D & E, which are of high resolution).

Extended Data Figure 1D. The abscissa label is incomplete.

Reference 23. First author is wrong and reference incomplete. Is it Longo et al. ?

Reference list. The citation style varies considerably across the reference list, e.g., sometimes journal names are spelt out in full, other times abbreviated, plus many other variations and shortcomings. Some references are duplicated (refs 34 and 37), others have multiple year dates (ref. 38). The authors should check all references carefully to correct these errors.

L154. Should state "...selectively disfavoured by mutations in the α -chain component of the insert domain...", as almost no effect is apparent in the β -chain component of the ID.

L158-160. The sentence needs rewording: it is not clear what exactly is being compared to what.

L158. The authors should consider including a Supplementary Table that lists the 80 residues they considered as part of Site 1, and to remove any doubt be clear that the 80 refers to 80 for each of site 1 and site 1' (if that is indeed what they mean).

L159. The references cited should include Menting et al. (2013) Nature 493, 241, the first 3D structure of insulin interacting with Site 1.

Extended Data Figure 2: the issues with this have already been stated above.

Extended Data Figure 2B (as it appears in the Figure itself, not the caption). The authors need to make clear the four insulins and label these as binding respectively to sites 1, 1', 2 and 2' for clarity.

Figure 2G inset. The authors need to label the domain on the lower left to inform the reader which it is.

L181/2. Remove paragraph break at this point.

L185-191. Can the authors deduce any structural impact of W659R, or would likely be innocuous from a structural perspective?

Figure 2H. The authors should highlight the location of W695R, P220L and R762S in the plot, given that they are discussed in L182-L203.

L152. Whereas I concur that the mAb score could be used as a surrogate for the relative degree of cell surface expression, the maximum of the two scores for the antibodies might be better than their mean in this regard; my reasoning being that the mutation of a single residue may compromise binding of one antibody and not the other and such binding compromise might be quite marked, without necessarily affecting receptor folding or expression. The authors should present their rationale for use of the mean.

L1056. Replace substitution with substitution.

Extended Data Figure 3. The preamble to the individual panel captions is not clear: all residues in the panels and the captions would have the same numbering irrespective of the isoform (as they lie upstream of the exon 11 product), so why are the authors drawing attention to the numbering difference when there isn't one in this instance?

L219-L215. FnIII-2a and FnIII-2A should be replaced by FnIII-2 α (three instances).

L225-L226. Replace "ID domain" with "part of ID α ". It is also premature to conclude that there is an "unresolved interaction" between ID α and the "interior of the receptor", given that the segment of ID α in question is indeed unresolved and therefore unlikely to be in any form of defined interaction with the remainder of the receptor; it is more likely to be conformationally mobile. It is also not clear what the authors mean by the "interior of the receptor". This discussion should be revised.

L251-L254. A number of the residue location descriptions here are wrong. For example, I see no "remodelling" of Y839 upon overlay of PDB structures 4ZXB (apo IR ectodomain) and 6PXV (four-insulin-bound IR ectodomain); note that Y839 is equivalent to Y800 in the PDB files as the latter number from the start of the mature protein and are based on IR-A, the exon 11 minus isoform. Likewise, the authors state that F862 is "not resolved": this residue is equivalent to F823 in the above PDB entries and it is indeed present in both, i.e., not "not resolved". It is also present in PDB file 8U4B (the IR-B apo ectodomain structure). The authors should therefore check (and correct if necessary) all their residue location claims throughout the manuscript.

Also, in L252/L253, FnIII-2A and FnIII-2B should read FnIII-2 α and FnIII-2 β , respectively – if indeed these are retained upon re-evaluation.

Finally, if the authors do wish to conclude that some residues are "not resolved in available structures", then they should within the Methods section state which structures they assessed (or the nature and date of the search they conducted) to reach this conclusion.

L1080: Single-letter amino acid code is missing for residue 183.

L290-294, L355-357. The relatively low correlation between the MAVE study reported here and in silico prediction tools is quite striking. Can the authors offer any further insight into this, even if speculative?

(Remarks on code availability)

Reviewer #2

(Remarks to the Author)

In this manuscript, Aslanzadeh, et al. have developed a high-throughput Multiplexed Assay of Variant Effect (MAVE) system to test the pathogenicity of mutations of the human insulin receptor (INSR) gene. To do so, they performed a massively parallel shotgun-type approach to introduce roughly 14,000 single-codon variants covering the extracellular domain of INSR. The authors took a massively parallel approach to characterizing the expression, binding affinity, and function (i.e., AKT phosphorylation) of each mutant INSR in response to high-dose insulin and two anti-INSR antibodies in MEFs lacking endogenous InsR and Igf1R. They harness this intricate system to make a number of interesting observations about the structure-function relationship of particular INSR mutations, including those that confer both loss- and gain-of-function and those associated with discrepant effects. The authors consider the translational significance of the MAVE system to be its potential to better characterize INSR variants of unknown significance (VUS) that have long bedeviled clinicians, thereby aiding rational selection of insulin or insulinomimetics for treatment of patients with INSR-related extreme insulin resistance.

This study is a novel and interesting approach to tackling the “VUS problem.” Its strengths include the large number of mutations tested, the differentiation of mutations’ effects on INSR expression vs. binding vs. function, and the inclusion of both insulin and anti-INSR antibodies as INSR ligands. However, the paper reads as meandering – dabbling episodically in various aspects of the INSR structure-function relationship without providing a compelling through line of their medical significance. DS and RMS are extremely rare conditions (TA less so, but still rare). Thus, although the MAVE’s precision-medicine implications are undoubtedly important for DS/RMS/TA patients, this system’s potential relevance to the overwhelming majority of insulin-resistant patients without fixed defects in INSR is not explored. The authors might therefore burnish the study’s salience by more clearly tying their data into type 2 diabetes pathophysiology or treatment. Of course, generating transgenic mice to test their model’s predictive success in guiding “rational” treatment selection (insulin vs. activating insulin autoantibody) is beyond the scope of the current study. Yet, the authors should consider incorporating an in vitro metabolic endpoint to supplement AKT phosphorylation, such as insulin-stimulated glucose uptake, for at least some of the INSR variants highlighted in the paper (e.g., G333Q, D627A). It would be better still, for future studies if not for the current one, to develop a high-throughput glucose-disposal metric (e.g., 2NBDG uptake) to complement their AKT assay.

On a more specific point, this reader was left wondering if all known human INSR extracellular pathogenic missense mutations are represented in the MAVE. Although this is implied in the text, stating it explicitly would better cement its translational significance. Table S3 may help to answer this question but the table is not well explained, including the meaning of the gray shading (seemingly denoting extracellular missense mutations) and of “no data” in the columns marked “experimental scoring (this study)” for particular mutations. Although frameshift, nonsense, and intracellular-domain mutations listed in the table would not be tested in the MAVE as a matter of course, some extracellular missense mutations are also listed as associated with “no [experimental] data” (e.g. V93A, T211I, C293R, N489D, W642R). To this reader, “no data” suggests that that particular mutation was not tested in the MAVE system; if this impression is incorrect, it would be helpful for the authors to clarify the definition. It is important that all known INSR extracellular pathogenic missense mutations are tested because these variants are the true positive controls that back the in-vivo verisimilitude of the study’s in vitro findings and thus are essential to evaluation of the MAVE. Similarly, it would be helpful to include known pathogenic (e.g., DS-/RMS-/TA-associated) mutations affecting receptor expression, binding, and/or function in the “manual” assays of Fig. 3C-E to better contextualize the data presented on G333Q and D627A.

Some additional points to consider include:

- Although it is alluded to in the materials section, the absence of both Igf1R and murine InsR in the transfected MEFs should be presented at the protein (+/- RNA) levels in order to formally exclude confounding of the binding and functional assays by the endogenous receptors.
- The seemingly discrepant binding of D627A in high-throughput binding assay (relatively high, Fig. 3B) vs. the gold-standard “manual” binding assay (relatively low, Fig. 3C) is concerning for the reliability of the system as a whole. How do the authors reconcile this as a non-systemic issue? Moreover, the highlighted G333Q appears not especially representative of most of the other G333 mutations, which seemed to have lower cell-surface expression and insulin binding. Was the goal of focusing extensively on G333Q in Fig. 3C-E to better discern an effect on binding independent of cell-surface expression?
- Fig. 5A would benefit from addition of at least a subset of WT INSR alongside DS/RMS/TA under Phenotypes. This might include, for example, synonymous mutations at the same residues associated with DS/RMS/TA.
- The many small heat maps are difficult to read and do not convey the figure’s “message” to advantage. It would be more helpful perhaps to focus only on the boxed/highlighted areas of interest.
- Figure quality is suboptimal in general. Although publication-quality figures are not required at this stage, some of them were difficult even to review. For example, Extended Data 1C is quite pixelated, the labels within the many plots and heat maps are so small as to require 200x (or greater) magnification to read unambiguously, and in some cases even the color-coded dots are difficult to tell apart at normal magnification (e.g., Extended Data 2C).

- The proliferation of mutation labels associated with the dots in, e.g., Fig. 3A-B are jumbled or even overlapping, diluting the point the figure is trying to make. It would be helpful to better highlight the mutations specifically referred to in the text within these figures. This reader had to spend a good deal of time sorting through, e.g., Fig. 3B to find the exact missense mutations cited in the text (D627A, G333Q).
- Some figure cross-references are inaccurate (e.g., line 169 reference to Extended 2F) and should be checked.

(Remarks on code availability)

Reviewer #3

(Remarks to the Author)

Aslanzadeh V, et al report a large scale deep mutational scan of the insulin receptor (INSR). Mutations within INSR disrupt this important proteins role leading to a rare genetic disease. There are many different mechanisms by which mutations can have their effects, including by altering expression, insulin binding, and downstream signaling. The authors here develop assays and measure the effects of ~14k variants on these phenotypes. In addition, partial agonist antibodies are used to study signaling. Overall, the manuscript reports a massive body of work that will be of broad interest.

Major points

Strengths:

1. The authors do well measuring a series of important phenotypes. I was particularly excited to see the data collection on partial agonist antibodies' binding and signaling effects. These data could answer historically intractable problems in understanding protein structure-function mechanisms. This data and system, the authors set up, could be used to uncover all sorts of interesting areas of INSR biology.
2. This dataset will be critically useful for predicting the effects of variants in people. The multi-phenotype data will help clinical geneticists make more confident decisions. The authors do well in comparing the effects of the data in Figure 5 to these effects and beautifully show the impact of measuring different phenotypes as well. Further, the authors do well to honestly present the data in direct comparison with variant effect predictors. Overall, this work will have a major impact on the VUS problem.

Opportunities for improvement:

1. The authors could improve the manuscript by including more statistical tests and direct quantification of differences they see. There is a lot of comparison where the effects are described as mild or strong but these qualitative terms can be hard to follow. In addition there are rough estimations throughout of the overall effects of mutations that do not include specific numbers or percentages to give a sense of the impact of those effects. I encourage the authors to be more quantitative in their presentation of the data.
2. I love the experiments with the antibodies and think they provide an opportunity for directly comparing the effects of binding and signaling across a range of ligands. However, these comparisons were not done, and in general, while there are some comparisons between insulin and antibodies – there are very few comparisons across the antibodies themselves and none in the context of the binding. This all leads this experiment to feel somewhat under-analyzed and does not feel like it does the beautiful work justice. I encourage the authors to dig a little deeper into the data.

Minor Points

Line 86: The authors mention conducting a multi-dimensional MAVE or deep mutational scan. They do not cite critical past studies in mutational, multi-dimensional, or mutational scans conducted on membrane proteins for signaling. I encourage the authors to cite and give credit to past relevant work in this space, as there are hundreds of prior studies they are building upon.

L102: The authors mention that mutations were made by nicking mutagenesis, which yields a mutagenesis and barcoded library. To my understanding, this method involves two steps: nicking mutagenesis, which makes the mutations, and then introducing barcodes in a following step. I would suggest slightly altering the language to represent this nuance.

L107: Authors combined numerated INSR and synonymous mutations. I think it would be worth considering separating these out if possible - the effects of synonymous mutations are not well explored and here is a nice opportunity to directly compare them.

L112 The authors mention using doxycycline-inducible shRNA *Insr* knockdown in the background of a knockout. Simultaneously the authors included the INSR plasmid library in a landing pad - which requires doxycycline induction. This is a clever complementation strategy - perhaps worth explaining this a bit more as it allows doing a DMS in an essential gene which can be tricky.

L125 - The authors assessed signaling on permeabilized cells with a phosphor-specific antibody. It's well known that PFA fixation makes sequencing quite difficult as a result most people do not do assays such as this. I did not notice a specific discussion of this issue in the methods. Was there specific optimization done to get this to work? If so it would be good to share this. L141 - The authors discuss the coverage in their library as 88% of variants detected. However, it's probably

worth including a threshold of reads that variants are detected at. For instance, it is not mentioned what percentage of variants have sufficient coverage to call a variant effect confidently or that are detected across sufficient number of samples to get a score.

L153: The authors compare insulin and antibody binding. They discuss areas where this is different as is apparent due to the 'blue blush.' While I appreciate the beautiful language I think it would be good to have a quantitative comparison here as well.

L175: The authors mention that overall they see 5,000 variants significantly decrease expression and 1,500 increase it. I think more specific numbers are useful here as are giving percentages.

L229 The authors mention their signaling assay had far less dynamic range than was previously seen. What was the difference? Why might that be?

L235: The authors compare different effects of variants based on their location in site 2. They discuss the effects of mutations in 'outlying' amino acids. It is unclear what these are referring to. Perhaps consider making a

L242 The authors say around 5,000 individual variants reduced signaling. In general I think it would be useful to include specific numbers (throughout the manuscript) but also put this into context of the number of variants that have scores. Finally, what is reduced signaling? What was the cutoff that was used to make this call, and what was its rationale?

L244: The authors mention lower confidence score - what confidence is this? L248: Authors mentioned T221 emerging as a residue of interest based on the average positional score then say but it's only based on 4 mutations. It's unclear what I am supposed to learn from this. In addition, it would be good to be a bit more specific about what average positional score the authors are referring to. Additionally, in the insets the score is referred to the median positional scores not the mean positional score.

L299: The authors compare the effects of partial agonist antibodies to insulin at 10nM. It is challenging to directly compare one compound to another because the effective concentration may be different. Why was this concentration chosen? Why is it an order magnitude less than insulin?

Figures—The figure labels are small, making interpreting them hard. The figure labels in the main figures should be more consistent.

Figure 6: I would have loved to see some of this data mapped on the structures. I had a really hard time interpreting the data here which is a shame as I think it's the most interesting experiment don't in the manuscript.

Figures 6B-D: The authors show the median score per position across the different signaling datasets. However, for most of the comparisons, they discuss individual variants. I think it would be worthwhile to show the variant distributions.

Figure 6E—It's unclear why this panel is not included in an earlier figure, as this figure's primary focus is the partial agonist antibodies. It seems this is here mostly to explain figures 6F-G (as described below I found this extremely confusing).

Figures 6F-G: The authors show the effects of an antibody signaling-insulin signaling. The y-axis denotes this as (average mab signaling)-(insulin signaling). But I am a bit confused by what is being calculated - is this the average across replicates? Is there no average in the insulin score? In the corresponding figure legend it says that only certain variants were included. I commend the authors in trying to be rigorous but I ended up very confused. I encourage them to include a section on the methods on this section of the manuscript. Figure 6H: The authors have a Venn diagram comparing the two mAb partial agonists but not insulin. It would be worthwhile having the venndiagram also include insulin. Instead, I think this is based on the data from the Manhattan plots which is mab-insulin.

By Willow Coyote-Maestas

(Remarks on code availability)

Version 1:

Reviewer comments:

Reviewer #1

(Remarks to the Author)

The authors have of the whole addressed the concerns I raised in my first review, and the manuscript is now largely suitable for publication. I will leave assessment of the authors' response to the other reviewers to the latter's and the Editor's discretion.

• However, there remain a few (minor) points which need to be addressed by the authors or which I recommend be subject to editorial discretion. These are detailed below, wherein I copy my original query and then state my response to that of the

authors.

• Responses by the authors to my remaining queries are acceptable and are not discussed further herein. I do not need to see a further corrected manuscript.

Original Query by Reviewer #1:

Extended Data Figure 2: the panel captions do not match the panels themselves, and likewise the references to panels in this figure in the main text appear incorrect. This makes it difficult to disentangle, and there is likely a panel missing from this Figure.

Reviewer's comment on Authors' response:

The authors' corrections are adequate, except that Panel E is now labelled "F" in the caption, this needs to be corrected.

Original Query by Reviewer #1:

I am also concerned about the authors' use of incredibly small font in some of the Figures, there may be editorial requirements in this regard.

Reviewer's comment on Authors' response:

The authors have improved the font size. I will leave it to editorial discretion as to whether further improvements are needed or can be achieved.

Original Query by Reviewer #1:

The authors do not discuss the impact of variants on receptor glycosylation. Obviously, mutation of residues involved in N- and O-linked glycosylation (either the residues that form the linkage to the sugar or those that form part of the associated motif) will remove glycosylation at the relevant site. It would be of interest to see whether, for example, mutation of the asparagine residues involved in N-linkage of sugars has impact on folding and/or insulin binding and whether such impact concurs with literature in this regard.

Reviewer's comment on Authors' response:

The authors have addressed this question in detail in their amended Results and are to be commended for doing so. For completeness, I recommend inclusion in the final manuscript of the four additional panels that they offer as additional Extended Data Figures / Panels, together with appropriate cross-referencing of these panels in the main text.

Original Query by Reviewer #1:

L81. The authors should also reference the class of receptor activating peptides pioneered by Schäffer et al. (2003, <https://doi.org/10.1073/pnas.0830026100>), which have also more recently been subject to structural studies to elucidate their mode of action. The latter could also be referenced at L365 (p15).

Reviewer's comment on Authors' response:

The authors add an appropriate reference at original line L365 to the Schäffer manuscript (authors' reference #13 in the current draft). However, the inclusion of the reference at original L81 (now line L83) does not make sense, as it suggests that the Schäffer study refers to activating antibodies, whereas it deals with insulin mimetic peptides. This needs to be corrected.

Original Query by Reviewer #1:

Extended Data Figure 1A. There is something amiss in coloring of the schematic on the right-hand side of this panel. In particular, the two FnIII-2 domains (in the middle of the structure, second domains up from the bottom) show a multitude of colors and not the brown color indicated on the left-hand side. There are also "brown" strands within the yellow FnIII-1 domains. Furthermore, there is no indication in the panel or in its caption that the four insulins are colored green. I recommend that the panel be redone in clearer and more contrasting colors. Also, in the left-hand component of the panel, the authors should label as such the respective α - and β -chains, to assist subsequent interpretation of Fig. 2A, as well as label the chain components as ID α , ID β FnIII-2 α and FnIII-2 β for consistency with the text.

Reviewer's comment on Authors' response:

The Figure is improved. However, the label "L2" appears detached above the right-hand side part of Panel B, I suspect that it is intended to be part of the left-hand part of Panel B, this needs to be corrected. Also, in Panel A, I require explicit inclusion of segments α CT and α CT', colored as in Panel B.

Original Query by Reviewer #1:

L158. The authors should consider including a Supplementary Table that lists the 80 residues they considered as part of Site 1, and to remove any doubt be clear that the 80 refers to 80 for each of site 1 and site 1' (if that is indeed what they mean).

Reviewer's comment on Authors' response:

The authors have now included such a Table. I request that the table include in addition to the residue numbers their respective three-letter amino acid codes to help prevent any confusion.

Original Query by Reviewer #1:

L251-L254. A number of the residue location descriptions here are wrong. For example, I see no "remodelling" of Y839 upon overlay of PDB structures 4ZXB (apo IR ectodomain) and 6PXV (four-insulin-bound IR ectodomain); note that Y839 is

equivalent to Y800 in the PDB files as the latter number from the start of the mature protein and are based on IR-A, the exon 11 minus isoform. Likewise, the authors state that F862 is “not resolved”: this residue is equivalent to F823 in the above PDB files and it is indeed present in both, i.e., not “not resolved”. It is also present in PDB file 8U4B (the IR-B apo ectodomain structure). The authors should therefore check (and correct if necessary) all of their residue locations claims throughout the manuscript.

Reviewer’s comment on Authors’ response:

The authors now have the following text:

START:

Threonine 221 emerged as a residue of interest based on the average positional score (Supplementary Figure 8), but this was based on only 4 variant scores in the MAVE. Other variants warranting future assessment based on selectively increased signalling include residues forming part of insulin binding site 1 (L2 domain: E343I, N376W; FnIII-1: N568C, P576C), residues involved in interdomain interactions that change on insulin binding (e.g. CR domain: C293F; FnIII-2a: R661W; FnIII-2b: Y839V; Extended Data Figure 4A-G), and residues not resolved in available structures (e.g. FnIII-3: F862Y).

:END

I note in the last part of the last line above that F862 is still claimed to be “not resolved”, even though the authors now agree that it is “resolved”. This needs to be fixed.

In terms of Y839, I agree that we had a different interpretation of their word “remodelling”, but I recommend as a fix that the authors use the more informative word “re-arrange” as opposed to the non-descript word “change” in the above text.

-ends-

(Remarks on code availability)

I have not reviewed the code as assessing it is outside of my expertise.

Reviewer #2

(Remarks to the Author)

I feel that the authors have thoroughly and successfully addressed the concerns I raised in my initial review. I have no additional comments.

(Remarks on code availability)

Reviewer #3

(Remarks to the Author)

Great job revising the article. I appreciated the increased number of citations, I think the manuscript now reads more clearly, and it’s easier for me to follow the figures.

Willow Coyote-Maestas

(Remarks on code availability)

Responses to reviewers' comments

Reviewer #1

Extended Data Figure 2: the panel captions do not match the panels themselves, and likewise the references to panels in this figure in the main text appear incorrect. This makes it difficult to disentangle, and there is likely a panel missing from this Figure.

AUTHORS' RESPONSE: Our apologies. These mismatches have been corrected so that captions and panels match, and so that callouts in text are now accurate. We confirm that the problem was only mislabelling, not the missed panel suspected by the reviewer.

I am also concerned about the authors' use of incredibly small font in some of the Figures, there may be editorial requirements in this regard.

AUTHORS' RESPONSE: This is a fair criticism (although we had sought to ensure that images on first submission were of sufficient resolution to allow zooming). We have endeavoured to maximise clarity by now increasing fonts wherever possible and useful. We will be happy to respond as required to any residual concerns.

The authors do not discuss the impact of variants on receptor glycosylation. Obviously, mutation of residues involved in N- and O-linked glycosylation (either the residues that form the linkage to the sugar or those that form part of the associated motif) will remove glycosylation at the relevant site. It would be of interest to see whether, for example, mutation of the asparagine residues involved in N-linkage of sugars has impact on folding and/or insulin binding and whether such impact concurs with literature in this regard.

AUTHORS' RESPONSE: This is an interesting question which had obviously also occurred to us. As a broad brush summary, published studies that have mutated glycosylation sites have shown that single mutations tend to be tolerated (i.e they result in no discernible change in cell surface expression, processing, transport, insulin binding, or autophosphorylation). Double, triple, or quadruple mutations, in contrast, generally do have an impact on processing and function. Our study is of single mutations, and in keeping with the literature we see no

evidence of major effects of mutating individual N- or O-glycosylated residues on either cell surface expression or insulin binding. We have amended *Results* as follows:

Of 15,996 variants in the plasmid library, 13,638 were detected, on average, across experiments (79% of all possible missense variants, or 88% of the library (Table S1)). Binding scores for insulin and antibodies showed distinct patterns (Figure 2A). For most regions binding was broadly similar for each ligand, in keeping with loss of receptor expression, exemplified by cysteine-rich domain (CRD) cysteines involved in intramolecular disulphide bonds (Extended Data Figure 2A). These stood out as blue columns in variant effect maps, indicating that any change from cysteine is highly deleterious to expression (Figure 2A). In contrast, but in keeping with prior literature, the consequence for cell surface expression and insulin binding of mutating residues subject to N- or O-linked glycosylation^{32,33} are negligible when averaged across all missense mutations at each site, with the marginal exception of N364. Some individual variants at these sites do significantly alter expression and/or binding, but this is presumably a consequence of structural alterations specific to those variants, rather than relating to loss of glycosylation *per se*. Residues whose mutation reduced expression are illustrated in the cryoEM structure of the 4-insulin-bound “T” conformation of the receptor in Extended Data Figure 2B.

As we now have a large number of supplementary figures, we have elected not to include further figures showing glycosylation site analysis, however for the reviewers’ interest, and also to illustrate the potential value of our dataset as a resource for those with similar structure-function questions, we further offer a variety of plots analysing binding and expression of mutations of all annotated residues subject to either N-linked or O-linked glycosylation, with a summary of supporting literature, below. If the editors prefer us to include these as formal supplementary figures we should be happy to incorporate these:

A. Functional effects of mutations of each **N-glycosylated asparagines** 43, 52, 138, 242, 282, 322, 364, 414, 445, 541, 633, 651, 698, 769, 782, 920, 933 (identified as N-glycosylation sites in *Sparrow (2008) DOI: 10.1002/prot.21768*). Mean scores for all mutations at the indicated position are shown.

B. Functional effects of individual mutations of **N-glycosylated asparagines**:

Supporting literature on glycosylation sites reviewed (N.B. all mature IR-B or IR-A numbering below depending on study; we have converted to proreceptor numbering in plots above here and elsewhere to align with our usage and HGVS nomenclature):

A. N-linked:

Sparrow (2008; DOI: 10.1002/prot.21768) experimentally confirmed 17 of 19 predicted N-linked glycosylation sites in CHO-K1 cells:

- Complex glycans at N25,255,295,418,606,624,742,755,893 (mature IR-B numbering):
- Multiple species of high-mannose glycans at N111,514
- Single species of complex glycan at N671
- Single species of high-mannose glycan at N215
- Mixture of complex, hybrid, and high-mannose glycan species at N16
- N387,906 confirmed by amino acid sequencing to be glycosylated:
- N337 confirmed by electron density maps to be glycosylated (= N364 by our numbering)
- [N78,282 predicted to be glycosylated not experimentally confirmed]

They summarise functional studies by observing redundancies in INSR glycosylation, noting that many sites can be mutated individually without detriment to cell surface expression, receptor processing or ligand binding. They further observe that when combinations are examined, each of L1, CR, and L2 domains requires at least one intact glycosylation site to ensure correct folding and processing.

Elleman (2000; PMID 10769182): mutated 15/18 sites (excluding N25, 397, and 894, all important for biosynthesis or function). N>Q of 742, 755, 893, and 906 had no effect on processing, intracellular transport, or insulin binding but reduced autophosphorylation and signal transduction. Mutation of 16, 25, 78, and 111 abolished proreceptor processing/ cell surface expression. Single mutations at 16 or 25 had little effect compared to double mutation. Glycosylation at 397 or 418 (but not both) is required for normal biosynthesis.

Leconte (1992; DOI: 10.1016/S0021-9258(18)41942-4): Mutating N742, N755, N893, N906 has no effect on cell surface expression or affinity for insulin but abolishes insulin-stimulated autophosphorylation

Collier (1993; DOI: 10.1021/bi00081a029): Mutating the first 4 N-linked glycosylation sites (N16, 25, 78, 111) prevents pro-receptor processing and cell surface expression

Caro (1994; DOI: 10.2337/diab.43.2.240): Mutated N16, N25, N78, N111 singly or in combination. N16 and N25 single mutations were processed to mature α - and β -subunits, but less efficiently than WT. Double mutants tested impaired proreceptor processing and/or function except N16/N25 double mutation.

Bastian (1993; DOI: 10.2337/diab.42.7.966): Substituting N397 or N418 for Q does not affect cell surface expression, binding or autophosphorylation, but double mutation abolishes cell surface expression

Klaver (2019; DOI: 10.1242/dmm.039602): Cell surface INSR expression reduced in oligosaccharyltransferase-null cells due to decreased N-linked glycan occupancy, impaired processing and increased ER retention

B. O-linked:

Sparrow et al (2007); doi 10.1002/prot.21261): experimentally confirmed O-glycosylation at T744,749,759 and 763; S757,758 (mature IR-B). These sites occur in a 20-residue segment that begins nine residues downstream from the start of the beta-chain. This region also includes N-linked glycosylation sites. This region of O-glycosylation could be deleted from an IR ectodomain construct without deleterious effects on receptor assembly or insulin binding.

Gutmann et al (2019); doi 10.1083/jcb.201907210, for example, corroborates the above findings structurally

Collier & Gorden (1991); doi 10.2337/diab.40.2.197 first showed that INSR O-linked oligosaccharide chains were attached to the N terminal tryptic peptide of the beta-subunit. Binding of insulin and INSR autophosphorylation were not dependent on O-linked glycosylation, as cells grown in the presence of O-glycosylation inhibitor exhibited normal response to insulin.

L81. The authors should also reference the class of receptor activating peptides pioneered by Schäffer et al. (2003, <https://doi.org/10.1073/pnas.0830026100>), which have also more recently been subject to structural studies to elucidate their mode of action. The latter could also be referenced at L365 (p15).

AUTHORS' RESPONSE: Thank you, we agree this is a good addition. This work is now cited.

L99. For clarity, should state "...from the start of the first codon of the mature receptor to the start of the transmembrane domain."

AUTHORS' RESPONSE: Change made.

Extended Data Figure 1A. There is something amiss in coloring of the schematic on the right-hand side of this panel. In particular, the two FnIII-2 domains (in the middle of the structure, second domains up from the bottom) show a multitude of colors and not the brown color indicated on the left-hand side. There are also "brown" strands within the yellow FnIII-1 domains. Furthermore, there is no indication in the panel or in its caption that the four insulins are colored green. I recommend that the panel be redone in clearer and more contrasting colors. Also, in the left-hand component of the panel, the authors should label as such the respective α - and β -chains, to assist subsequent interpretation of Fig. 2A, as well as label the chain components as $ID\alpha$, $ID\beta$, $FnIII-2\alpha$ and $FnIII-\beta$ for consistency with the text.

AUTHORS' RESPONSE: We agree that something appeared awry. We have tried to enhance ease of visual inspection and interpretation now with recoloring.

Extended Data Figure 1C. In the version that I downloaded, Panel C is of unacceptably low resolution (cf. Panels B, D & E, which are of high resolution).

AUTHORS' RESPONSE: Corrected.

Extended Data Figure 1D. The abscissa label is incomplete.

AUTHORS' RESPONSE: Corrected

Reference 23. First author is wrong and reference incomplete. Is it Longo et al?

AUTHORS' RESPONSE: Corrected

Reference list. The citation style varies considerably across the reference list, e.g., sometimes journal names are spelt out in full, other times abbreviated, plus many other variations and shortcomings. Some references are duplicated (refs 34 and 37), others have multiple year dates (ref. 38). The authors should check all references carefully to correct these errors.

AUTHORS' RESPONSE: Corrected

L154. Should state "...selectively disfavoured by mutations in the α -chain component of the insert domain...", as almost no effect is apparent in the β -chain component of the ID.

AUTHORS' RESPONSE: This change has been made

L158-160. The sentence needs rewording: it is not clear what exactly is being compared to what.

AUTHORS' RESPONSE: To our eyes the wording was already clear, but we have made this more explicit still (see new text in response to next comment)

L158. The authors should consider including a Supplementary Table that lists the 80 residues they considered as part of Site 1, and to remove any doubt be clear that the 80 refers to 80 for each of site 1 and site 1' (if that is indeed what they mean).

AUTHORS' RESPONSE: We have now created a supplementary table of these 80 residues as requested (new **Supplementary Table 2**). We further confirm that we mean 80 for each of Site 1 and Site 1'. These 80 residues represent our broad view of residues potentially contributing thermodynamically to insulin binding, based on proximity to insulin in available structures. Results text edited accordingly:

We then compared median antibody **binding scores with median insulin binding scores** for 80 residues implicated structurally in insulin binding in sites 1 **and 1'** (henceforth "site 1"), **based on proximity to insulin in available structures or prior alanine scanning mutagenesis studies**^{4,5,27-32}(**Supplementary Table 2**), comparing them to cysteine-rich domain (CRD) cysteines (Figure 2E). Mutation of site 1 residues slightly reduced cell surface INSR expression compared to **WT receptor**, while severely reducing insulin binding (Figure 2E, inset). This was in contrast to CRD cysteine mutations, which nearly all reduced expression and insulin binding (Supplementary Figure 5A). Mutation of amino acids implicated structurally in insulin binding at sites 2 and 2' ("site 2")^{4,5}(**Supplementary Table 2**) had no effect on receptor expression, and only mildly reduced insulin binding on average, however mutation of a small number of residues severely impaired both (e.g. I512, L514, W516, F530, I561, K583) (Figures 2F, Supplementary Figure 5D).

[to remind the reviewer, these are the corresponding references, already cited:

4. Uchikawa et al. *Elife* 8, (2019).
5. Gutmann et al. *J Cell Biol* 219, (2020).
26. Williams et al *J Biol Chem* 270, 3012–3016 (1995).
27. Whittaker et al *J Biol Chem* 277, 47380–47384 (2002).
28. Mynarcik et al *J Biol Chem* 272, 2077–2081 (1997).
29. Mynarcik et al *J Biol Chem* 271, 2439–2442 (1996).
30. Andersen et al *PLoS One* 12, e0178885 (2017).]

L159. The references cited should include Menting et al. (2013) *Nature* 493, 241, the first 3D structure of insulin interacting with Site 1.

AUTHORS' RESPONSE: We agree. Apologies for the oversight. Now added.

Extended Data Figure 2B (as it appears in the Figure itself, not the caption). The authors need to make clear the four insulins and label these as binding respectively to sites 1, 1', 2 and 2' for clarity.

AUTHORS' RESPONSE: We have changed the figure to highlight this. We hope this offers sufficient clarity.

Figure 2G inset. The authors need to label the domain on the lower left to inform the reader which it is.

AUTHORS' RESPONSE: The “domain pictured on the lower left actually features structural elements of FnIII-1 and FnIII-2 α intertwined, while the α -helix pictured is part of the L2 domain. It is not easy to label these features individually, but we have endeavoured to do so as follows:

L181/2. Remove paragraph break at this point.

AUTHORS' RESPONSE: Change made

L185-191. Can the authors deduce any structural impact of W659R, or would likely be innocuous from a structural perspective?

AUTHORS' RESPONSE: Wild-type W659 is deeply buried and likely contributes to the hydrophobic core of the β -sheet. Replacing it with arginine introduces a steric and electrostatic

shift. The arginine side chain extends differently, possibly destabilizing local packing or interacting unfavourably with the surrounding environment. This change is reflected in the $\Delta\Delta G$ computed by FoldX (+4.3 kcal/mol), indicating a significant energetic penalty, consistent with a destabilizing effect on the receptor's structure. This is consistent with the structural modelling reported previously in the paper cited in the current text.

W659R was reported in a Chinese case report in trans with a very convincing loss-of-function intracellular mutation. In aggregate it does indeed seem highly likely that it is pathogenic, in our view, but as we explain it convincingly does not show impaired binding or signalling in the current assay despite assay of multiple independent variants in multiple experiments. Indeed, as noted elsewhere in the text, a small number of other convincing pathogenic variants do not show obvious abnormalities in static cellular assays, emphasising the point that no singly cellular model replicates all *in vivo* relevant aspects of INSR function. We have slightly edited the relevant paragraph as follows:

The only reported severe LoF variant discordantly classified was W659R, reported in Donohue syndrome in trans with V1054M³¹. Prior evidence for W659R LoF was a single immunoblot after transient overexpression, **showing severely impaired proreceptor processing**³². No difference was found for insulin binding or signalling in the MAVE, **however**, despite at least 6 barcodes representing the variant in all experiments. The median binding score for the 19 substitutions evaluated at this position was moreover only -0.06 (interquartile range -0.15-0.07). Our consistent findings in orthogonal assays do not **rule out** pathogenicity, but suggest that any LoF for W659R is not captured in static expression and binding assays of the overexpressed variant **in MEFs**.

Figure 2H. The authors should highlight the location of W659R, P220L and R762S in the plot, given that they are discussed in L182-L203.

AUTHORS' RESPONSE: This has been done

L152. Whereas I concur that the mAb score could be used as a surrogate for the relative degree of cell surface expression, the maximum of the two scores for the antibodies might be better than their mean in this regard; my reasoning being that the mutation of a single residue may compromise binding of one antibody and not the other and such binding compromise might be quite marked, without necessarily affecting receptor folding or expression. The authors should present their rationale for use of the mean.

AUTHORS' RESPONSE: We demonstrate the extremely strong concordance between mAb binding results in Figure 2B-D, the only outlying residues in scatterplots being those known to form part of the one of the Ab epitopes, as discussed. These 16 outlying residues are the only ones for which the mean Ab binding score would underestimate binding. In consequence, for each of these 16 residues we used the uncompromised single mAb binding score instead of the sum. This corrects for the problem the reviewer identifies. We have not altered the prior analysis, but we now add an explicit note about treatment of these epitope residues as follows:

As expected, binding of the antibodies correlated strongly (Figure 2B, Extended Data Figure 2C). Non correlating residues corresponded closely to epitopes for either mAb 83-14²¹ (Figure 2C), or mAb 83-7²¹ (Figure 2D), demonstrating the utility of the assay for epitope mapping. For subsequent analyses, means of the scores for the two antibodies were taken as the (cell surface) "expression score",

except for the 16 epitope residues with discordant scores (Figure 2B-D), for which the single score from the antibody whose epitope was not mutated were used.

L1056. Replace *substation* with *substitution*.

AUTHORS' RESPONSE: Corrected

Extended Data Figure 3. The preamble to the individual panel captions is not clear: all residues in the panels and the captions would have the same numbering irrespective of the isoform (as they lie upstream of the exon 11 product), so why are the authors drawing attention to the numbering difference when there isn't one in this instance?

AUTHORS' RESPONSE: We agree. This qualifier is unnecessary and is deleted in the revised text:

Supplementary Figure 7. INSR residues identified to result in gain of function upon substitution shown mapped onto the cryo-EM structures of various insulin-bound states. Monomers are coloured different shades of grey. Insulin is represented by green ribbon structures. Yellow depicts residues where multiple substitutions increase insulin binding (G183, H274, G333, D627, V631, N633, S634, S635, S636, W659, E660, R661, Q662, A663, E664, D665, S666, E667, S682, R683, T684, S686, P687, P688). Blue depicts residues where a unique substitution increases binding (D169, N541, N568, T653, K679). ~~Structures are of the A isoform but residues highlighted are numbered according to the B isoform pro-receptor for consistency.~~ Some residues highlighted may be involved in interactions with parts of the receptor not resolved in available structures. A, Unbound, INSR B-isoform (PDB: 8U4B). The highlighted section is rotated counterclockwise 45° to the full view of the receptor. B, Single insulin bound (green ribbon) INSR isoform A (PDB: 7STI). PDB structure 7STI is the A isoform of the *Mus musculus* orthologue, but for consistency residues highlighted are numbered according to the human B isoform pro-receptor with the corresponding mouse residue in brackets. The highlighted section is rotated counterclockwise 90°. C and D, Two insulin bound (green ribbons) INSR (PDB: 7STJ (asymmetrical), and 7STH (symmetrical)). These structures are the A isoform of the mouse orthologue, but for consistency residues highlighted are again numbered according to the human B isoform pro-receptor with the corresponding mouse residue in brackets. Highlighted sections are rotated counterclockwise 90°. E) Saturated, four insulin bound INSR (PDB: 6PXV). The highlighted section is rotated counterclockwise 90° to the full view.

L219-L215. *FnIII-2a* and *FnIII-2A* should be replaced by *FnIII-2α* (three instances).

AUTHORS' RESPONSE: Corrected

L225-L226. Replace “ID domain” with “part of IDα”. It is also premature to conclude that there is an “unresolved interaction” between IDα and the “interior of the receptor”, given that the segment of IDα in question is indeed unresolved and therefore unlikely to be in any form of defined interaction with the remainder of the receptor; it is more likely to be conformationally mobile. It is also not clear what the authors mean by the “interior of the receptor”. This discussion should be revised.

AUTHORS' RESPONSE: We have edited our wording to reflect the uncertainty noted by the reviewer as follows:

Regions where most mutations increase insulin binding and/or receptor expression in the current MAVE (Figure 3F) tend to be located at or near dynamic sites of interdomain interaction, for example between L1' and FnIII-2a in the unbound conformation, **or in part of ID α that is unresolved in available structures and thus likely to be conformationally mobile** (Supplementary Figure 7)."

L251-L254. A number of the residue location descriptions here are wrong. For example, I see no "remodelling" of Y839 upon overlay of PDB structures 4ZXB (apo IR ectodomain) and 6PXV (four-insulin-bound IR ectodomain); note that Y839 is equivalent to Y800 in the PDB files as the latter number from the start of the mature protein and are based on IR-A, the exon 11 minus isoform. Likewise, the authors state that F862 is "not resolved": this residue is equivalent to F823 in the above PDB entries and it is indeed present in both, i.e., not "not resolved". It is also present in PDB file 8U4B (the IR-B apo ectodomain structure). The authors should therefore check (and correct if necessary) all their residue location claims throughout the manuscript.

AUTHORS' RESPONSE: Thank you for this careful assessment which has helped us to identify and correct an error, namely our claim that F862 is not resolved – this claim arose from a simple misreading of a spreadsheet. In contrast we stand by our discussion of Y839, while endeavouring to use more precise language. If we have misinterpreted the reviewer's comment then we will be happy to address this further.

More specifically, Y839 shown in red below in structures 8U4B (apo; Choi lab 2024) and 6PXV (4 ins bound; Bai lab 2019). We used 8U4B instead of 4ZXB (Ward/Lawrence; IR-A) as it related to the B isoform.

Even cursory inspection indicates that Y839 moves from being at the interface of the protomers in the apo conformation, to being widely separated in the bound conformation. We hypothesise that it may play a role in stabilisation of the apo conformation, and intended to imply no more than this by use of the word "remodelling". We infer that this word has been interpreted by the reviewer to suggest re-organisation/re-orientation of this residue within the domain. To reflect our meaning more precisely we have edited "remodel" to the more neutral "change" as follows:

Threonine 221 emerged as a residue of interest based on the average positional score (Supplementary Figure 8), but this was based on only 4 variant scores in the MAVE. Other variants warranting future assessment based on selectively increased signalling include residues forming

part of insulin binding site 1 (L2 domain: E343I, N376W; FnIII-1: N568C, P576C), residues involved in interdomain interactions that **change** on insulin binding (e.g. CR domain: C293F; FnIII-2a: R661W; FnIII-2b: Y839V; Extended Data Figure 4A-G), and residues not resolved in available structures (e.g. FnIII-3: F862Y).

Further to reassure the reviewer we offer an excerpt from a spreadsheet in which we have curated structural and functional data and alignments:

MOUSE					Human					Hs IR-B					Mouse IR-A				Hs IR-A	
Pro-IR-B	Mat-IR-B	Pro-IR-A	Mat-IR-A	QD	QD	Pro-IR-B	Mat-IR-B	Pro-IR-A	Mat-IR-A	Domain	8U4B	7STI	7STJ	7STH	6PXV					
169	142	169	142	D	D	169	142	169	142	L1	D	D	D	D	D					Not visible in structures
183	156	183	156	G	G	183	156	183	156	L1	G	G	G	G	G					
258	231	258	231	F	F	258	231	258	231	CR	F	F	F	F	F					GoBinding Multi Subs
274	247	274	247	H	H	274	247	274	247	CR	H	H	H	H	H					GoBinding Specific Subs
293	266	293	266	C	C	293	266	293	266	CR	C	C	C	C	C					Gain of Signalling
333	306	333	306	G	G	333	306	333	306	CR	G	G	G	G	G					Gain of Bind. and Sig.
343	316	343	316	E	E	343	316	343	316	L2	E	E	E	E	E					
376	349	376	349	N	N	376	349	376	349	L2	N	N	N	N	N					
541	514	541	514	N	N	541	514	541	514	FnIII-1	N	N	N	N	N					
568	541	568	541	N	N	568	541	568	541	FnIII-1	N	N	N	N	N					
578	551	578	551	P	P	578	549	578	549	FnIII-1	P	P	P	P	P					
629	602	629	602	D	D	627	600	627	600	FnIII-2a	D	D	D	D	D					
633	606	633	606	V	V	631	604	631	604	FnIII-2a	V	V	V	V	V					
635	608	635	608	N	N	633	606	633	606	FnIII-2a	N	N	N	N	N					
636	609	636	609	S	S	634	607	634	607	FnIII-2a	S	S	S	S	S					
637	610	637	610	S	S	635	608	635	608	FnIII-2a	S	S	S	S	S					
638	611	638	611	S	S	636	609	636	609	FnIII-2a	S	S	S	S	S					
655	628	655	628	T	T	653	626	653	626	FnIII-2a	T	T	T	T	T					
661	634	661	634	W	W	659	632	659	632	FnIII-2a	W	W	W	W	W					
662	635	662	635	E	E	660	633	660	633	FnIII-2a	E	E	E	E	E					
663	636	663	636	R	R	661	634	661	634	FnIII-2a	R	R	R	R	R					
664	637	664	637	Q	Q	662	635	662	635	FnIII-2a	Q	Q	Q	Q	Q					
665	638	665	638	A	A	663	636	663	636	FnIII-2a	A	A	A	A	A					
666	639	666	639	E	E	664	637	664	637	FnIII-2a	E	E	E	E	E					
667	640	667	640	D	D	665	638	665	638	FnIII-2a	D	D	D	D	D					
668	641	668	641	S	S	666	639	666	639	FnIII-2a	S	S	S	S	S					
669	642	669	642	E	E	667	640	667	640	FnIII-2a	E	E	E	E	E					
681	654	681	654	K	K	679	652	679	652	FnIII-2a	K	K	K	K	K					
684	657	684	657	S	S	682	655	682	655	FnIII-2a	S	S	S	S	S					
685	658	685	658	R	R	683	656	683	656	FnIII-2a	R	R	R	R	R					
686	659	686	659	T	T	684	657	684	657	FnIII-2a	T	T	T	T	T					
688	661	688	661	S	S	686	659	686	659	ID	S	S	S	S	S					
689	662	689	662	P	P	687	660	687	660	ID	P	P	P	P	P					
690	663	690	663	P	P	688	661	688	661	ID	P	P	P	P	P					
841	814	829	802	Y	Y	839	812	827	800	FnIII-2b	Y	Y	Y	Y	Y					
864	837	852	825	F	F	862	835	850	823	FnIII-3	F	F	F	F	F					
943	916	931	904	Y	Y	941	914	929	902	FnIII-3	Y	Y	Y	Y	Y					

Also, in L252/L253, FnIII-2A and FnIII-2B should read FnIII-2 α and FnIII-2 β , respectively – if indeed these are retained upon re-evaluation.

AUTHORS' RESPONSE: Corrected as suggested.

Finally, if the authors do wish to conclude that some residues are “not resolved in available structures”, then they should within the Methods section state which structures they assessed (or the nature and date of the search they conducted) to reach this conclusion.

AUTHORS' RESPONSE: For the reviewer's interest, these were the residues we found not to be resolved, with associated structures:

8U4B = Hs IR-B Proreceptor: 1-27, 201-203, 546-554, 567-573, 685-715, 743-794, 947-1382
7STI = Mm IR-A Proreceptor: 1-27, 190-194, 298-300, 549-552, 567-574, 686-710, 748-784, 935-1372
7STJ = Mm IR-A Proreceptor: 1-27, 190-194, 298-300, 546-552, 567-574, 686-719, 747-784, 935-1372
7STH = Mm IR-A Proreceptor: 1-27, 190-194, 298-300, 546-554, 567-575, 686-712, 748-784, 938-1372
6PXV = Hs IR-A Proreceptor: 1-27, 190-194, 298-300, 546-554, 684-717, 745-780, 938-1370
Accessed on PDB, 5th July 2024; viewed on UCSF ChimeraX [version: 1.8rc202405230136 (2024-05-23)]

We have added the following text to Methods:

PDB identifiers for receptor structures assessed and discussed were 8U4B for the human IR-B receptor, 6PXV for the human IR-A receptor, and 7STI, 7STJ and 7STH for the murine IR-A receptor. Structures were accessed from PDB on 5th July 2024 and viewed using UCSF ChimeraX [version: 1.8rc202405230136 (2024-05-23)]

L1080: Single-letter amino acid code is missing for residue 183.

AUTHORS' RESPONSE: Corrected to "G183"

L290-294, L355-357. The relatively low correlation between the MAVE study reported here and in silico prediction tools is quite striking. Can the authors offer any further insight into this, even if speculative?

AUTHORS' RESPONSE: The correlation of scores from our INSR MAVE and *in silico* VEPs (R=0.53 between INSR sum of scores and AlphaMissense) is close to the average reported correlations between other MAVEs and VEPs [R 0.56-0.61 depending on assay type (*Livesey & Marsh; Variant effect predictor correlation with functional assays is reflective of clinical classification performance. Genome Biol 26, 104 (2025). <https://doi.org/10.1186/s13059-025-03575-w>*]. Factors that influence these correlations are not fully understood, but include the suitability of the experimental model, number of experimental replicates (on the MAVE side) and the quality of training data and amount of systematic bias (on the VEP side). In addition, MAVEs and VEPs often aim to estimate different outcomes (gene function in a cellular model vs pathogenicity at the organismal level) so are not expected to correlate perfectly.

Notably, several top VEPs correlate very strongly with each other (e.g. R=0.98, iGEMME vs ESCOTT). As these are black box, AI-based methods, it is difficult to interpret this result, but we speculate that this represents a limitation of current VEPs trained on a similar pool of available data (mainly evolutionary conservation and protein structure information) and that don't capture functional information that can only be measured experimentally.

We have edited the relevant paragraph of the discussion as follows:

The MAVE performed outstandingly in predicting pathogenicity, like the best *in silico* variant effect predictors (VEPs). Yet not all receptor attributes have been assayed in this study, and by incorporating further assays, the ability of INSR MAVEs to discriminate pathogenic mutations may be enhanced further. The current MAVE validates the best current VEPs empirically, however, interestingly, correlation among scores from VEPs and MAVEs are modest (e.g. R=0.53 between INSR sum of scores and AlphaMissense) but close to the average reported correlations between other MAVEs and VEPs (R 0.56-0.61 depending on assay⁴⁶). Factors influencing these correlations are incompletely understood, but include the suitability of the experimental model, and number of replicates (on the MAVE side) and the quality of training data and amount of systematic bias (on the VEP side). Moreover MAVEs and VEPs often aim to estimate different outcomes (gene function in a cellular model vs pathogenicity at organismal level) so are not expected to correlate perfectly.

Reviewer #2

In this manuscript, Aslanzadeh, et al. have developed a high-throughput Multiplexed Assay of Variant Effect (MAVE) system to test the pathogenicity of mutations of the human insulin receptor (INSR) gene. To do so, they performed a massively parallel shotgun-type approach to introduce roughly 14,000 single-codon variants covering the extracellular domain of INSR. The authors took a massively parallel approach to characterizing the expression, binding affinity, and function (i.e. AKT phosphorylation) of each mutant INSR in response to high-dose insulin and two anti-INSR antibodies in MEFs lacking endogenous InsR and Igf1R. They harness this intricate system to make a number of interesting observations about the structure-function relationship of particular INSR mutations, including those that confer both loss- and gain-of-function and those associated with discrepant effects. The authors consider the translational significance of the MAVE system to be its potential to better characterize INSR variants of unknown significance (VUS) that have long bedeviled clinicians, thereby aiding rational selection of insulin or insulinomimetics for treatment of patients with INSR-related extreme insulin resistance.

This study is a novel and interesting approach to tackling the “VUS problem” Its strengths include the large number of mutations tested, the differentiation of mutations’ effects on INSR expression vs. binding vs. function, and the inclusion of both insulin and anti-INSR antibodies as INSR ligands. However, the paper reads as meandering – dabbling episodically in various aspects of the INSR structure-function relationship without providing a compelling through line of their medical significance.

AUTHORS’ RESPONSE: We note the reviewer’s concern that the narrative is “meandering” and that the structure-function aspect of the discussion lacks a fully developed “through line” to the “medical significance”. While we do accept that we cannot fully develop this element of the study, this is entirely a reflection of space constraints, and, crucially, of the need to offer enough technical details of a MAVE undertaken on an enormous scale for this to be useful to the MAVE community, and accepted in relevant data repositories such as MaveDB. We highlight to this reviewer the array of questions related to technical aspects of the MAVE from reviewer 3 in this regard. We believe that successful achievement of the primary purpose of this study – to offer an account of the functional effects of as many INSR missense variants as possible – should not be downplayed because of concern that we cannot extend discussion of interesting variants fully. The number of missense variants we study here is more than 2 orders of magnitude higher than the number of variants studies in the field to date. We could, of course, simply have neglected to offer much discussion of individual variants, but we felt that the INSR structure-function field would be very interested in this study, and so we preferred to offer some interesting vignettes to highlight aspects of the vast dataset that can be dived into by the community. We hope that the reviewer can appreciate this pragmatic stance. We anticipate several downstream studies (indeed these are underway) where the sorts of analysis the reviewer laments the lack of will be undertaken.

DS and RMS are extremely rare conditions (TA less so, but still rare). Thus, although the MAVE’s precision-medicine implications are undoubtedly important for DS/RMS/TA patients, this system’s potential relevance to the overwhelming majority of insulin-resistant patients without fixed defects in INSR is not explored. The authors might therefore burnish the study’s salience by more clearly tying their data into type 2 diabetes pathophysiology or treatment.

AUTHORS’ RESPONSE: Again, we have very little space to develop these wider more speculative elements in the discussion. While we hold that the importance for rare patients with genetic receptoropathy alone lends the study sufficient impact to be of broad general interest (mindful of its obvious human relevance, which is commonly more debatable for the many studies reporting findings solely in rodent models, for example), we are ever interested in wider implications beyond rare disease. The argument is a little more nuanced than implied by the reviewer, however. While insulin resistance (IR) is indeed a nearly obligate feature of

type 2 diabetes (T2DM), we have shown clearly, later confirmed by others, that the IR seen in receptoropathy (no fatty liver, normolipidaemia, preserved or high adiponectin) is a different IR subphenotype to that seen in obesity/T2DM (fatty liver, dyslipidaemia, low adiponectin)(e.g. PMID 19164855, 18299442). It would thus be disingenuous to overplay relevance to common T2DM. On the other hand it is our hope that insulin receptoropathy could be used as a rare disease stepping stone (with attendant licensing benefits) into common disease for INSR mimetics such as mAbs. We are currently further engineering Abs to this end to overcome the receptor desensitisation that we have previously described in mice and that reduces potency of receptor agonism in vivo (PMID 32816962). A final interesting aspect of INSRopathy is that reverse genetic studies in populations such as UK Biobank has indicated that heterozygous INSR LoF mutations actually confer a benign metabolic profile (e.g. PMID 34875679). We are also currently utilising our experimental function scores to stratify the population-wide extracellular variants to explore this further.

All these further ventures are of great interest, but are incredibly difficult to do justice to given stringent space constraints in the current manuscript, and we prefer not to be overly superficial in an already long document. If the editors choose to offer us more space then we will, of course, be happy to offer a succinct elaboration of these potential translational applications. This paper does, however, describe the key resource that underpins such current and planned studies. Obviously these reports, where there will be more space to expound on translational relevance, this paper will be cited.

Of course, generating transgenic mice to test their model's predictive success in guiding "rational" treatment selection (insulin vs. activating insulin autoantibody) is beyond the scope of the current study. Yet, the authors should consider incorporating an in vitro metabolic endpoint to supplement AKT phosphorylation, such as insulin-stimulated glucose uptake, for at least some of the INSR variants highlighted in the paper (e.g., G333Q, D627A).

AUTHORS' RESPONSE: We refer the reviewer again to the space constraints we face. The comment is reasonable, but rather than layering further assays in the current crowded manuscript we have instead relied on reference to voluminous previous literature from many labs including ours. For example, in one of our recent publications (PMID 29700562) we studied a panel of convincingly pathogenic mutations for autophosphorylation, and AKT phosphorylation (in CHO cells) and insulin-stimulated glucose uptake (in 3T3-L1 adipocytes, which express GLUT4). Our results, expressed as % of WT INSR in the same cells/assay conditions, were as follows:

INSR variant		Response to 10 nM insulin stimulation			
		INSR autophosphorylation (% WT)	AKT phosphorylation (% WT)		Glucose uptake (% WT)
Mature Receptor	Pro-receptor		T308/9	S473/4	
WT	WT	100	100	100	100
P193L	P220L	23	79	94	99
F248C	F275C	3	16	57	51
R252C	R279C	27	108	111	86
S323L	S350L	9	46	50	37
F382V	F409V	19	75	52	99
D707A	D734A	0	11	26	29

R118C	R145C	65	Not assayed
I119M	I146M	99	Not assayed
K460E	K487E	95	Not assayed

The reviewer will observe that the general pattern, at least under these assay conditions, is that the biochemical defect in INSR autophosphorylation is greater than the defect in AKT phosphorylation, which in turn is greater than the effect on glucose uptake. This pattern reflects the way information is transmitted between different steps of the signalling pathway, each of which may exhibit a different dose-response relationship. We do NOT conclude that the variants with little or no discernable reduction in glucose uptake are not pathogenic, but rather that their abnormal signalling kinetics are only properly manifested in vivo, where interstitial insulin concentrations are low and likely pulsatile. This fascinating field of signalling dynamics and information theory really is beyond the scope of the current study, and for the immediate purposes of designing a discriminatory and scalable functional readout, we still believe that AKT phosphorylation was a sensible surrogate of biologically meaningful receptor function. Work is underway in the lab to refine our readout and increase the dynamic range of the assay, although current efforts focus on transcriptional reporter assays.

It would be better still, for future studies if not for the current one, to develop a high-throughput glucose-disposal metric (e.g. 2NBDG uptake) to complement their AKT assay.

AUTHORS' RESPONSE: Such a biological end point assay would of course be of great interest, but it is not readily apparent how one could be deployed on the scale required, quite aside from the issues of assay sensitivity discussed above. The reviewer is reminded that we have studied around 14,000 missense variants as well as controls expressed in a total of around 60 million cells per experiment. Moreover although 2-NBDG is mentioned as one possible tool, it is our understanding that this is not fit for the purpose suggested, as field consensus is that it doesn't report on glucose uptake: glucose enters the GLUT4 transporter binding site C-1 first, and the 2-NB group is too large to be accommodated. The assay thus does not measure glucose transport. (See, for example PMID: 34224807). As noted above, we are actively working towards developing readouts with greater dynamic ranges for the future. These will not, however, directly read out glucose uptake.

On a more specific point, this reader was left wondering if all known human INSR extracellular pathogenic missense mutations are represented in the MAVE. Although this is implied in the text, stating it explicitly would better cement its translational significance. Table S3 may help to answer this question but the table is not well explained, including the meaning of the gray shading (seemingly denoting extracellular missense mutations) and of "no data" in the columns marked "experimental scoring (this study)" for particular mutations. Although frameshift, nonsense, and intracellular-domain mutations listed in the table would not be tested in the MAVE as a matter of course, some extracellular missense mutations are also listed as associated with "no [experimental] data" (e.g. V93A, T211I, C293R, N489D, W642R). To this reader, "no data" suggests that that particular mutation was not tested in the MAVE system; if this impression is incorrect, it would be helpful for the authors to clarify the definition. It is important that all known INSR extracellular pathogenic missense mutations are tested because these variants are the true positive controls that back the in-vivo verisimilitude of the study's in vitro findings and thus are essential to evaluation of the MAVE.

AUTHORS' RESPONSE: Out of 88 unique missense extracellular variants described as causal for human severe insulin resistance, we have generated data for 77, or around 85%, in keeping with overall coverage in the MAVE of all missense variants. The drop out comes either from lack of coverage in the original mutagenesis library, or in the cellular library subsequently generated, as described explicitly. The 11 "clinical" variants which were missing

were C293R, T211I, R279H, N458D, V93A, N489D, D523N, Y606C, W642R, S835I, and V867F. Missing some variants is inherent in all MAVEs, but does not mean that no useful information is generated about these variants, albeit indirectly. In the graphs below (new **Supplementary Figure 6**) we plot the scores for all 77 pathogenic variants for which we have data against the harbouring individual clinical variants for which we do have scores, against the median score for all missense variants at that same amino acid.

The correlations shown demonstrate that the median score at a position predicts the effect of these pathogenic mutations with more than 50% accuracy, so even where specific variants of interest are missing from the MAVE, behaviour of different co-located variants still offers predictive value.

The median scores for the 11 variants missing from the MAVE are as follows (with the number of individual variants at each position included, N, in brackets):

Variant of interest	Median expression score at corresponding position (N) [LoF cut off -0.26]*	Median binding score at corresponding position (N) [LoF cut off -0.15]*	Median signalling score at corresponding position (N) [LoF cut off -0.10]*
C293R	-0.38 (13)	-0.19 (13)	-0.04 (13)
T211I	0.01 (14)	-0.35 (14)	-0.15 (14)
R279H	-0.13 (14)	-0.23 (14)	-0.03 (14)
N458D	-0.53 (12)	-0.17 (12)	-0.22 (12)
V93A	-0.30 (9)	-0.64 (9)	-0.14 (9)
N489D	-0.15 (17)	-0.28 (17)	-0.07 (17)
D523N	-0.27 (18)	-0.48 (18)	-0.31 (18)
Y606C	0.036 (10)	-0.26 (10)	-0.14 (10)
W642R	-1.09 (7)	-0.67 (6)	-0.36 (9)
S835I	-0.47 (13)	-0.38 (13)	-0.69 (15)
V867F	-0.36 (15)	-0.34 (15)	-0.10 (15)

*Loss of function (LoF) cut offs are conservative and approximate, as derived in Supplementary Figure 8. All scores below cut offs are red.

This Table is also now included as (new) **Supplementary Table 5**, with brief additions to *Results* also as follows:

Overall, 4,687 missense variants from 14,576 with scores (32%) significantly reduced expression, while 1,469/14,576 (10%) increased it (Figure 2H). Approximately 4,623/14,243 (32%) reduced insulin binding and 961/14,243 (7%) consistently increased it (Figure 2I). To validate findings, we curated all previous functional studies of individual variants, whether disease-associated or studied during structure-function investigations. Given variable experimental approaches used over 36 years we stratified variants pragmatically into 3 groups for each functional attribute: severe (<10% wild-type) or complete loss of function (LoF), intermediate LoF, and unimpaired function, also adjudicated on confidence in prior findings (Table S2).

Out of 88 unique pathogenic, missense variants, functional data were generated for 77, or 87.5%, in keeping with overall coverage of all missense variants. INSR expression and insulin binding scores from MAVE and literature strongly agreed. For cell surface expression, 11/12 reported severe LoF variants and 17/22 intermediate LoF variants were called as LoF with high confidence in the MAVE (Figure 2H). The only reported severe LoF variant discordantly classified was W659R, reported in Donohue syndrome in trans with V1054M³³. Prior evidence for W659R LoF was a single immunoblot after transient overexpression, showing severely impaired proreceptor processing³⁴. No difference was found for insulin binding or signalling in the MAVE, however, despite at least 6 barcodes representing the variant in all experiments. The median binding score for the 19 substitutions evaluated at this position was moreover only -0.06 (interquartile range -0.15-0.07). Our consistent findings in orthogonal assays do not exclude pathogenicity, but suggest that any LoF for W659R is not captured in static expression and binding assays of the overexpressed variant in MEFs.

Indirect evidence of loss of function was also obtained for the 11 documented pathogenic variants not directly studied in the MAVE: in Supplementary Figure 6 all function scores obtained for other 77 pathogenic variants are plotted against the position median score, showing that median scores predict the effect of these mutations with more than 50% accuracy. This means that even where specific variants of interest are missing from the MAVE, behaviour of different co-located variants still offers predictive value. Median scores per position for the pathogenic variants not directly assessed in this study are tabulated in Supplementary Table 5, showing that two residues show loss of function for one assay, four for two assays, and five for all three assays.

Similarly, it would be helpful to include known pathogenic (e.g., DS-/RMS-/TA-associated) mutations affecting receptor expression, binding, and/or function in the "manual" assays of Fig. 3C-E to better contextualize the data presented on G333Q and D627A.

AUTHORS' RESPONSE: We do accept the reviewer's point that including some well established loss-of-function variants in our "manual" assays would have been ideal presentationally. However in related assays we and others have found highly concordant deficits for a raft of such variants (we refer again to our previously published study of INSR variants in which we assayed INSR autophosphorylation, AKT phosphorylation and insulin-stimulated glucose uptake for 6 pathogenic missense mutations (see table above), as well as to the large literature that we document in **Supplementary Table 3**) and so we have no doubt at all what the findings here would have been. Unfortunately a continuing lab refurbishment prevents us from running these assays in the timescale of this resubmission. As the key comparison we make is with the WT receptor, we hope the reviewer will accept this.

Although it is alluded to in the materials section, the absence of both Igf1R and murine InsR in the transfected MEFs should be presented at the protein (+/- RNA) levels in order to formally exclude confounding of the binding and functional assays by the endogenous receptors.

AUTHORS' RESPONSE: The Igf1r KO MEFs that are the basis for our study were examined for MmInsr, MmIgf1r, and HsINSR expression when they were first bought into the lab, using qRT-PCR with species specific primers and immunoblotting. They were negative for MmIgf1r and HsINSR (showing no cross-contamination with other lines from the Wallace/Cosgrove laboratories from where they were procured), and positive, as expected, for MmInsr. After engineering to incorporate a landing pad permitting doxycycline-driven HsINSR expression, we further confirmed the desired efficient knockdown of endogenous MmInsr and re-expression of HsINSR in our final selected cell clone which was the basis of pur MAVE by immunoblotting. We agree these validation are important, and they are now included as new **Supplementary Figure 3:**

Supplementary Figure 3 Validation of Cellular Model used for Massively Parallel Functional Assays

A. Immunoblots demonstrating downregulation of endogenous mouse Insr and concomitant induction of myc-tagged human INSR on exposure of R-MmInsrKD + HsINSRmyc mouse embryo fibroblasts (MEFs harbouring inducible WT human myc-INSR and anti-mouse Insr shRNA) to doxycycline. Controls are R- (Igf1r^{-/-}) MEFs, R-MmInsrKD MEFs harbouring inducible anti-Insr shRNA only, and Hepa1-6 cells as positive controls. Dox = doxycycline. B. Immunoblots demonstrating absent expression of Igf1r in keeping with Igf1r gene knockout. β-actin is shown as a loading control. In this case Hek293T cells were positive controls. C. FACS trace demonstrating mAb83-7 binding as a surrogate for human INSR cell surface expression, demonstrating tight control of transgenic myc-INSR expression by doxycycline, with a high dynamic range of the expression assay.

Main text:

Study of INSR variant function requires a background free of endogenous insulin receptor and the homologous insulin-like growth factor 1 (Igf1) receptor. We used a mouse embryo fibroblast model in which *Igf1r* knockout²¹ is augmented by doxycycline-inducible, shRNA-mediated *Insr* knockdown¹⁴. We chose conditional knockdown as *Insr* knockout in this cell line reproducibly yielded cellular atypia, compromising viability. We lentivirally introduced a single copy of a published landing pad allowing efficient *Bxb1*-mediated integration of transfected plasmid¹⁹. Finally, the mutagenised, barcoded human *INSR* plasmid library was introduced to cells by electroporation (Figure 1B). **This generated a cellular system in which doxycycline simultaneously induced efficient endogenous mouse *Insr* knockdown and human *INSR* overexpression (Supplementary Figure 3), obviating concerns about clonal selection resulting from lethality in sustained culture of double *Igf1r/Insr* null cells.**

The new figure is also called out where appropriate in *Methods*

The seemingly discrepant binding of D627A in high-throughput binding assay (relatively high, Fig. 3B) vs. the gold-standard “manual” binding assay (relatively low, Fig. 3C) is concerning for the reliability of the system as a whole. How do the authors reconcile this as a non-systemic issue?

AUTHORS’ RESPONSE: Our interpretation does not align with that of the reviewer. The high-throughput assay (which employed 100nM AF647-labelled Insulin for binding) showed increased maximal insulin binding for both G333Q and D627A, with the former greater than the latter, and increased cell surface expression for D627A only. In Fig 3C, showing the results of low throughput binding of Eu-labelled insulin, G333Q is indeed confirmed to show greater binding (100 nM insulin (obviously) corresponds to the ‘-7’ (log10) datapoint in Fig 3C). Binding by D627A is indeed lower, but its expression is also clearly higher in the immunoblotting shown in Fig 3D. So we interpret the increased binding in the MAVE (which simply measures total insulin binding per cell) to be a consequence of increased cell surface expression that outweighs the smaller decrease in binding per receptor. In other words, the manual validation assays ARE consistent with the high throughput assay. Finally, we note that increased expression is itself an interesting readout, as all constructs are expressed from the same landing pad, meaning that changes in expression have a posttranscriptional basis.

Moreover, the highlighted G333Q appears not especially representative of most of the other G333 mutations, which seemed to have lower cell-surface expression and insulin binding. Was the goal of focusing extensively on G333Q in Fig. 3C-E to better discern an effect on binding independent of cell-surface expression?

AUTHORS’ RESPONSE: The reviewer is correct. Altering cell surface expression will inevitably commensurately alter maximal insulin binding, which is what we assayed in the MAVE, and so variants which reduced expression and binding to a similar degree were of much less interest to us than those which showed evidence of disproportionate or selective effects on binding. We thus focused on G333Q given the interesting observation in the MAVE of a selective increase in insulin binding. This was of significantly greater interest to us than variation at G333 *per se*, and so we felt no need to study a “representative” G333 variant.

Fig. 5A would benefit from addition of at least a subset of WT INSR alongside DS/RMS/TA

under Phenotypes. This might include, for example, synonymous mutations at the same residues associated with DS/RMS/TA.

AUTHORS' RESPONSE: This is a good suggestion – thank you. We have now computed a “synonymous variant score” essentially as for the missense variants, including all synonymous variants for each codon affected by the DS, RMS, and TA causal variants plotted. These scores were normalized by dividing by the mean barcode score of WT sequence and then log2-transformed, as before. In the plots below, **Patho Syn** represents the distribution of these normalized synonymous variant scores, all clearly centred on zero. Statistical comparisons were undertaken of the synonymous variant scores and each of the disease groups (DS, RMS, and TA). All resulting significant p-values are indicated using asterisks above the respective disease phenotype groups, with according amendment to *Methods* and *Results* text.

Figure 5: Evaluation of multiplexed assay results as an aid to genetic diagnosis A, Boxplots showing expression, insulin binding, and signalling score distribution, as well as composite sum of scores for different groups of variants. ClinVar variants designated benign or likely benign, as variants of uncertain significance (VUS), or as pathogenic or likely pathogenic are shown. Where conflicting dual designations are present in ClinVar both are shown. Data for all variants and homozygous variants from gnomAD are also shown, as well as scores for curated homozygous and compound heterozygous (CompHet) pathogenic variants associated with Donohue syndrome (DS), Rabson-Mendenhall syndrome (RMS), or type A insulin resistance (TA). Finally, scores for all synonymous variants at residues mutated in pathogenic mutations are plotted as “Patho Syn”. See *Methods* for average score computation for compound heterozygous variants. The Mann Whitney U test was used to test if DS, RMS, and TA variants differ from Patho Syn variants and if ClinVar VUS and pathogenic variants differ from ClinVar Benign variants. *, ** and * = $p < 0.05$, 0.01 and 0.005 respectively.**

Incidentally, the reviewer will note a small number of low synonymous variant scores. Most of these are represented by a single barcode and are not statistically significant, however some synonymous variants did show significant loss of function scores. The only ones with significant loss-of-function scores for at least 2 assays are tabulated below (from a total of 88 codons for which scores were computed):

Codon Change	Surface Expression	Insulin Binding	Signalling	FDR (any)
ACC488ACG	-0.61	-0.28	-0.22	FDR all <0.0025
GTT657GTG	-0.4	-0.33	-0.27	FDR all <0.02
CGC145AGG	-0.6	No data	-0.29	FDR both <0.02

*Loss of function (LoF) cut offs are conservative and approximate, as derived in Extended Data Figure 5A. All scores below cut offs are red.

We do not believe that these rare loss-of-function scores reflect an unidentified second mutation in *cis* given our long read sequencing and given their representation by multiple independent barcodes, so these may be true effects of these particular synonymous variants, perhaps destabilising mRNA. This is also pertinent to the point about interrogating synonymous variants raised by reviewer 3, and will be followed up in separate studies.

The many small heat maps are difficult to read and do not convey the figure's "message" to advantage. It would be more helpful perhaps to focus only on the boxed/highlighted areas of interest.

AUTHORS' RESPONSE: It is not completely clear to us which figure this refers to – we suspect it relates to the zoomed out complete variant effect maps (Figure 2A, 4A, 6A). If so, we respectfully disagree with the reviewer. The zoomed out views gives an excellent overview of the richness of the dataset, the mapping of function scores onto the domain structure, and differences between mAbs and insulin (now additionally quantified with a further statistical test, as described below). However if we have misinterpreted the comment and in fact the reviewer refers to other figure element(s) then we will be happy to respond further as required.

Figure quality is suboptimal in general. Although publication-quality figures are not required at this stage, some of them were difficult even to review. For example, Extended Data 1C is quite pixelated, the labels within the many plots and heat maps are so small as to require 200x (or greater) magnification to read unambiguously, and in some cases even the color-coded dots are difficult to tell apart at normal magnification (e.g., Extended Data 2C).

AUTHORS' RESPONSE: As noted in response to other reviewers, we agree and apologise for the extra effort involved in reviewing. We apologise specifically for pixelation of Extended Data Figure 1C, due to uploading of a suboptimal file type in error. We have made a series of changes to fonts and labelling throughout the figures to aid clarity, and will be happy to respond to any further concerns about this if required.

The proliferation of mutation labels associated with the dots in, e.g., Fig. 3A-B are jumbled or even overlapping, diluting the point the figure is trying to make. It would be helpful to better highlight the mutations specifically referred to in the text within these figures. This reader had to spend a good deal of time sorting through, e.g., Fig. 3B to find the exact missense mutations cited in the text (D627A, G333Q).

AUTHORS' RESPONSE: Again, we have done our best now to maximise readability, enlarging label fonts, reducing the number of labels, and highlighting variants called out in text.

Some figure cross-references are inaccurate (e.g. line 169 reference to Extended 2F) and should be checked.

AUTHORS' RESPONSE: We apologise and hope we have now corrected all such oversights

Reviewer #3

Overall, the manuscript reports a massive body of work that will be of broad interest.

AUTHORS' RESPONSE: We are pleased that the reviewer recognised substantial value and strengths of our study. We will not re-iterate the positive comments but instead address the constructive criticism point-by-point below:

Major points

The authors could improve the manuscript by including more statistical tests and direct quantification of differences they see. There is a lot of comparison where the effects are described as mild or strong but these qualitative terms can be hard to follow. In addition there are rough estimations throughout of the overall effects of mutations that do not include specific numbers or percentages to give a sense of the impact of those effects. I encourage the authors to be more quantitative in their presentation of the data.

AUTHORS' RESPONSE: Statistical analysis is obviously crucial in experiments of this scale, and we have tried already to undertake appropriate testing wherever possible. Indeed for every variant for which we have data we have not only function scores but also FDRs for each score, computed using the bootstrapping approach we describe. All scores and FDRs are available and interrogatable in MaveDB. Our manuscript also includes some secondary statistical analyses, each presented in figures (Figs 1E, 2A/E/F, 4A/B/C/D, 5A/C, 6A/E/F). We have checked all of these to ensure that the nature of statistical testing is clearly described (in figure legends in general). These appear to be us to be comprehensive. Finally, we also now conduct significant further analysis as suggested by the reviewer below to compare different MAVEs statistically, normalising data and applying paired non parametric testing in a sliding window, represented graphically in **Figures 2A, 4A, and 6A** and in new **Supplementary Figure 10**. Relevant changes to the manuscript are described in more detail in response to the Reviewer's specific comments later.

Beyond this we wonder additionally if the reviewer's concern may in part relate to the words we use in text to describe significant changes. Here we have applied a judgement that may differ from that of the reviewer. It is our view that inserting specific values for scores and FDRs in text tends cumulatively to compromise readability and thus obscure meaning. It is thus our taste to keep the text descriptors as they are, as all specific numerical data are easily visible in display items. If the editors take a different view we should be happy to alter this however.

One area where we agree we have not been able to undertake robust statistical testing is in our summarising of a large prior literature spanning 37 years of functional studies of different mutations. In referring to this prior data we indeed adopt semiquantitative terms such as "mild" and "strong", justifying this by laying out our approach clearly in Methods and displaying all the source evidence we have surveyed in extensive supplementary tables (**Supplementary Tables 3 and 4**), which in themselves are a significant resource for the community interested in the INSR.

I love the experiments with the antibodies and think they provide an opportunity for directly comparing the effects of binding and signaling across a range of ligands. However, these comparisons were not done, and in general, while there are some comparisons between insulin and antibodies – there are very few comparisons across the antibodies themselves and none in the context of the binding. This all leads this experiment to feel somewhat under-analyzed and does not feel like it does the beautiful work justice. I encourage the authors to dig a little deeper into the data.

AUTHORS' RESPONSE: We agree that an exciting opportunity afforded by the new resource we present will be to pan other atypical ligands against the mutant library for translational purposes, as we acknowledge in discussion already (*Discussion: "...It will be of interest to use the cellular INSR library generated in this study to extend analysis to dose response relationships and recycling kinetics. Future work could also further screen different ligands, different downstream pathways, or INSR expressed as a hybrid with the homologous IGF1R"*). These studies, using a wider range of ligands, are being planned but are beyond the scope of this already long manuscript. The reviewer also asserts that the opportunity to directly compare the effects of binding and signaling across insulin and antibodies here is not fully grasped, specifically pointing to "very few comparisons across the antibodies themselves and none in the context of the binding". We would first push back at this by highlighting the comparisons that are made in the initial manuscript:

Fig 2A and corresponding Supplementary Figure: Ins vs mAb binding

Figure 3A,B: Ins vs mAb binding for "gain of function" variants

All of Figure 6: signalling by insulin vs mAbs (now expanded to 2 figures per below)

Various Extended Data Figures

Nevertheless there is clearly potential for further analysis. We address this by splitting Figure 6 into 2 figures (new **Figures 6** and **7**). As elaborated on in response to specific comments below, the new space has permitted us to:

- Add more formal statistical comparison between MAVEs, including new tracks in overview figures (**Figures 2A,4A & 6A**), with collation of these in new **Supplementary Figure 10**, again all as detailed further below.
- Show the selective mAb sensitivity scores superimposed on the 4 insulin-bound receptor structure (new **Figure 7C**)
- Replace the current Venn diagrams with a more informative annotated scatter plot, representing the relative performance of the two mAbs studied at activating severely impaired receptors (new **Figure 7D**)

Minor points

Line 86: The authors mention conducting a multi-dimensional MAVE or deep mutational scan. They do not cite critical past studies in mutational, multi-dimensional, or mutational scans conducted on membrane proteins for signaling. I encourage the authors to cite and give credit to past relevant work in this space, as there are hundreds of prior studies they are building upon.

AUTHORS' RESPONSE: We are acutely aware of the prior studies on which we build. We assume the reviewer refers to the whole MAVE field when mentioning "hundreds" of prior papers. Instead it is those papers applying MAVEs to different classes of membrane-associated receptors that appear most pertinent to this study. Any citation is inevitably

somewhat selective, however we have now added several key examples of such prior MAVE papers. These address in particular GPCRs, with relatively few receptor tyrosine kinase MAVEs – if the reviewer can suggest other important citations we should be happy to add them. Additional citations are indicated below in revised *Introduction* text:

Multiplexed Assays of Variant Effect (MAVE) have emerged as a powerful approach in functional genomics¹⁸. By coupling comprehensive mutagenesis to pooled, sequencing-based assays, MAVE enable systematic evaluation of the functional consequences of many thousands of different amino acid changes simultaneously. The potential of this approach when applied to membrane-associated receptors is rapidly being demonstrated, having been used variously to delineate residues involved in ligand binding (e.g. ¹⁹), to determine mechanisms of signal transduction (e.g. ²⁰⁻²¹), and to stratify pathogenic variants for potential targeted therapy²².

L102: The authors mention that mutations were made by nicking mutagenesis, which yields a mutagenesis and barcoded library. To my understanding, this method involves two steps: nicking mutagenesis, which makes the mutations, and then introducing barcodes in a following step. I would suggest slightly altering the language to represent this nuance.

AUTHORS' RESPONSE: To our minds this is reasonably clear as illustrated in Figure 1 and described in the old text shown below:

(L100-104): We subcloned the *INSR* gene into an entry plasmid with an interposed *attB* site to permit integration into a genomic landing pad and incorporated 30mer barcodes upstream from the open reading frame¹⁸. Mutations were introduced into the *INSR* sequence by nicking mutagenesis based on a published method¹⁹ to yield a mutagenised, barcoded library.

Nevertheless we have endeavoured to clarify further by making the following textual adjustments:

Mutations were then introduced into the previously barcoded *INSR* plasmids by nicking mutagenesis, based on a published method²⁰, to yield a mutagenised, barcoded library.

*L107: Authors combined numerated *INSR* and synonymous mutations. I think it would be worth considering separating these out if possible - the effects of synonymous mutations are not well explored and here is a nice opportunity to directly compare them.*

AUTHORS' RESPONSE: Figure 1E (already in paper) shows WT and Synonymous variant barcode score distributions. The (new) table below shows the output of a Mann Whitney U test, showing no difference between WT and Synonymous variant barcode score distributions, but a p-value approximating to zero for comparison of WT and missense variants

	Comparison	U_statistic	p_value
Insulin Binding	WT versus Synonymous	4.2×10^7	0.803
	WT versus Missense	1.0×10^9	0
Expression	WT versus Synonymous	4.9×10^7	0.052
	WT versus vs Missense	1.1×10^9	0

Figure 1E has now been annotated with the above p values. Further interrogation of synonymous variants would indeed be of interest, perhaps identifying rare variants which reduce mRNA stability or cause aberrant splicing. We will indeed undertake such analysis on this whole dataset as suggested, but we take the view that the current manuscript is already crowded, and so have elected not to include such further analysis now. We do, however, now plot some limited synonymous variant data in updated Figure 5A, and offer some more discussion in response to the relevant question of Reviewer 2 above.

L112 The authors mention using doxycycline-inducible shRNA Insr knockdown in the background of a knockout. Simultaneously the authors included the INSR plasmid library in a landing pad - which requires doxycycline induction. This is a clever complementation strategy - perhaps worth explaining this a bit more as it allows doing a DMS in an essential gene which can be tricky.

AUTHORS' RESPONSE: This strategy has indeed worked well. It is already clearly described technically, but to highlight the point we have lightly edited the results as follows, as well as calling out the new extended data figures requested by reviewer 2 and described above:

Study of INSR variant function requires a background free of endogenous insulin receptor and the homologous insulin-like growth factor 1 (Igf1) receptor. We used a mouse embryo fibroblast model in which Igf1r knockout²¹ is augmented by doxycycline-inducible, shRNA-mediated Insr knockdown¹⁴. We chose conditional knockdown as Insr knockout in this cell line reproducibly yielded cellular atypia, compromising viability. Finally, we lentivirally introduced a single copy of a published landing pad allowing efficient Bxb1-mediated integration of transfected plasmid¹⁹. Finally, the mutagenised, barcoded human INSR plasmid library was introduced to cells by electroporation (Figure 1B). This generated a cellular system in which doxycycline simultaneously induced efficient endogenous mouse Insr knockdown and human INSR overexpression (Supplementary Figure 3), obviating concerns about clonal selection resulting from lethality in sustained culture of double Igf1r/Insr null cells.

L125 - The authors assessed signaling on permeabilized cells with a phospho-specific antibody. It's well known that PFA fixation makes sequencing quite difficult as a result most people do not do assays such as this. I did not notice a specific discussion of this issue in the methods. Was there specific optimization done to get this to work? If so it would be good to share this.

AUTHORS' RESPONSE: Our barcode PCR amplicon was only 120 bp, increasing the likelihood of amplification from fixed samples. Nevertheless we agree that PFA fixation can pose challenges for downstream sequencing. We used overnight proteinase K digestion to break down crosslinked proteins. We initially tried the QIAamp DNA FFPE Tissue Kit, which resulted in low DNA yield, so turned to traditional Phenol:Chloroform:Isoamyl alcohol (25:24:1) extraction. Phenol in particular helps to denature proteins further, and release DNA. This significantly improved yield. This is now mentioned in methods as follows:

PCR was performed to amplify gDNA barcodes. Optimisation of DNA preparation and PCR was required to enable efficient amplification of barcodes even from signalling assays, which employed PFA fixation prior to FACS. We used overnight proteinase K digestion to break down crosslinked proteins, and traditional Phenol:Chloroform:Isoamyl alcohol (25:24:1) extraction rather than kit-

based extraction. Phenol helps to denature proteins further, releasing DNA, while the short PCR amplicon of 120 bp also increased likelihood of amplification. This approach led to a reliably improved PCR yield. To avoid amplifying barcodes from any persisting extragenomic INSR plasmid, the reverse primer was designed to anneal to a gDNA sequence downstream from the plasmid integration site.

L141 - The authors discuss the coverage in their library as 88% of variants detected. However, it's probably worth including a threshold of reads that variants are detected at. For instance, it is not mentioned what percentage of variants have sufficient coverage to call a variant effect confidently or that are detected across sufficient number of samples to get a score.

AUTHORS' RESPONSE: We filtered out any barcodes that had fewer than 150 reads in total across 4 bins, as already mentioned in methods (L729 of original submission). All given % relate to numbers of variants that were present after applying this filter. We now reiterate our statement from *Methods* in the *Results* text as follows:

After removing barcodes with fewer than 150 reads across 4 bins, we observed 15,996 single missense and stop codon variants out of 18,560 possible (86% coverage), tagged by 80,956 unique barcodes. 81% of mutations associated with >1 barcode (Extended Data Figure 1C). 35% of the barcodes aligned with unmutated INSR or synonymous mutations only, and were taken as wild-type (WT) in downstream analyses (Extended Data 1E, Tables S1,S2).

We'd additional observe that for all experiments we undertook 2-5 independent replicates, and concordance for scored variants across replicates was universally excellent, as below. Indeed >94% variants were seen in all 5 replicates for the signalling experiment.

	insulin binding	83-07 binding	83-14 binding	83-07 signalling	83-14 signalling	insulin signalling
Number of replicates	2	2	2	3	3	5
Total scored variants	14,243	14,237	14,416	14,377	14,418	14,544
Observed at least in 2 replicates	13,389	13,561	13,774	13,834	13,915	14,230
Observed at least in 2 replicates (%)	94	95	96	96	97	98

We mention this as for some of the later translational analyses in Figure 6E-H we applied additional filters, mandating presence of variants in more than one replicate. All this is explicitly described.

L153: The authors compare insulin and antibody binding. They discuss areas where this is different as is apparent due to the 'blue blush.' While I appreciate the beautiful language I think it would be good to have a quantitative comparison here as well.

AUTHORS' RESPONSE: We are heartened by the reviewer's admiration for our use of language... But we also take their important point and have now added a further statistical analysis track to **Figures 2A, 4A and 6A**, as well as adding new **Supplementary Figure 10** to address it. This track features a line indicating $-\log_{10}p$ values arising from application of the Wilcoxon Signed Rank Test to test the null hypothesis that mAb and insulin stimulation have the same effect, with the alternative hypothesis that responses to mAb are less impaired than

to insulin. Paired missense variants for sliding windows of 15 residues were tested. This nicely visually represents regions such as the α CT “blue blush” where mAbs score systematically higher than insulin (i.e. show a relatively less impaired signalling response), as well as revealing other aspects of comparison worthy of comment. Figure legends, *Methods*, *Results* and *Discussion* have been updated accordingly:

Methods

Regional Comparison of Variant Scores Between Different Assays

To compare consequences for different functional readouts of mutating different regions of the receptor, we applied the Wilcoxon signed-rank test. Prior to statistical comparison, scores from each dataset, which show different ranges between assays, were normalized using the empirical cumulative distribution function (ECDF), which transforms scores into quantile ranks (ranging from 0 to 1), enabling robust, distribution-free comparisons between scores from different assays. We aligned the two datasets being compared, matching amino acid substitution and position, before applying the Wilcoxon signed-rank test to a 15-residue sliding window. Windows were tested only if they contained at least five variants. The test assessed whether the distribution of ECDF-normalized scores in condition A (e.g. insulin binding) was significantly lower than in condition B (e.g., expression), i.e. whether mutations had more deleterious effects in assay A than assay B, as indicated in figure legends. To aid interpretability, only one-sided tests were used (alternative = ‘less’), focusing on reduced variant tolerance in the first dataset. The resulting $-\log_{10}(\text{p-value})$ values were plotted against the window centre position to visualize regions of statistically significant sensitivity shifts. Regions for which $p < 0.05$ are shown as shaded areas on the plots.

Results

We then compared median antibody binding scores with median insulin binding scores for 80 residues implicated structurally in insulin binding in sites 1 and 1' (henceforth “site 1”), based on proximity to insulin in available structures or prior alanine scanning mutagenesis studies^{4,5,27–32} (Supplementary Table 2), comparing them to cysteine-rich domain (CRD) cysteines (Figure 2E). Mutation of site 1 residues slightly reduced cell surface INSR expression compared to WT receptor, while severely reducing insulin binding (Figure 2E). This was in contrast to CRD cysteine mutations, which nearly all reduced expression and insulin binding (Supplementary Figure 5A). This assessment was confirmed by agnostic statistical comparison of expression and binding scores, which revealed a close correlation between regions showing selective loss of binding, and site 1 residues (Figure 2A, red boxes). Mutation of amino acids implicated structurally in insulin binding at sites 2 and 2' (“site 2”)^{4,5} (Supplementary Table 2) had no effect on receptor expression, and only mildly reduced insulin binding on average. Moreover although mutation of a small number of residues severely impaired both (e.g. I512, L514, W516, F530, I561, K583) (Figure 2F, Supplementary Figure 5D), very little correlation was seen between experimentally determined selective loss of binding and regions enriched for Site 2-implicated residues (Figure 2A, red boxes).

....

To assess maximal insulin signalling (E_{max}), we used a fluorescently labelled antibody to quantify Akt phosphorylation (on Ser473/474) in response to 100 nM insulin. High level results are illustrated in Figure 4A, with insulin binding results as a comparator. While there was broad similarity between binding and signalling maps, two hotspots where signalling was relatively more impaired than binding were seen, in the L2 and at the junction of the L2 and CR domains (orange hatched boxes, Figure 4A). Several regions were also seen where binding was relatively more impaired than signalling, especially including elements of binding site 1 in the L1 and FnIII-1 domains (purple hatched boxes, Figure 4A). The α CT helix was not prominent in this analysis, likely because the effect of its mutation on insulin binding is severe. Analysed together, mutations in insulin binding site 1 residues indeed had less effect on signalling than on binding (Figure 4B). These findings are in keeping with the “spare receptor” concept, whereby maximal signalling is seen at relatively low levels of receptor occupancy³⁷.

Appropriate alterations to Figure Legends have also been made

L175: The authors mention that overall they see 5,000 variants significantly decrease expression and 1,500 increase it. I think more specific numbers are useful here as are giving percentages.

AUTHORS' RESPONSE: We are not convinced that giving precise numbers truly enhances clarity (as opposed to introducing pseudo-precision), but we have made this change in the text

L229 The authors mention their signaling assay had far less dynamic range than was previously seen. What was the difference? Why might that be?

AUTHORS' RESPONSE: In our original *Results* section we simply observe that “While insulin- and antibody-binding assays showed a >100-fold dynamic range, the dynamic range of the signalling assay was less than 5-fold. We thus undertook 5 replicates.” In Discussion we further observe that “We also used a single very high insulin concentration, reporting only maximal response to receptor activation, or efficacy, and missing potentially clinically significant rightward shifts in the sigmoidal insulin dose-response curve.” Indeed this aspect of assay design is by far the dominant factor in determining the constrained dynamic range we see. Additional potential reasons include the requirement to detach and permeabilise cells before anti-phosphoAkt antibody binding to the labile phosphoAkt, which inevitably leads to some loss of signal compared to binding of mAb or labelled insulin to stable cell surface receptor. As discussed in response to reviewer 2, gradual attenuation of the magnitude of the signalling effect is also seen on progressing down the signalling pathway, meaning that the effect at the level of Akt phosphorylation may intrinsically have a lower dynamic range than at the level of receptor autophosphorylation. We would note that the cellular model used, combining *Igf1r* knockout and conditional endogenous *Insr* knockdown showed some evidence of low level residual endogenous *Insr* expression, but although this is likely to have narrowed the dynamic range of both signaling and insulin binding assays, without affecting antibody stimulation and binding (as the antibodies used are human INSR-specific), excellent dynamic range is seen for insulin binding, so we do not believe that this is a major source of reduced dynamic range in signalling, and so we do not discuss this in the manuscript.

For these technical and biological reasons, and because we use a single high concentration of insulin, the signaling readout we use likely has greater specificity than sensitivity for detection of impaired signaling. In use of our findings to aid genetic diagnosis this problem is mitigated by use of composite scores incorporating data from all three assay types.

Nevertheless our signaling findings are likely to be of greatest value for providing evidence for loss of function.

Given limited space, we have made only the following limited edits to capture some more of the nuance around the issue of dynamic range:

Discussion

We also used a single very high insulin concentration, reporting only maximal response to receptor activation, or efficacy, and missing potentially clinically significant rightward shifts in the sigmoidal insulin dose-response curve. **Loss of phosphoAkt signal during cell processing may further contribute to loss of dynamic range in signalling assays.**

L235: The authors compare different effects of variants based on their location in site 2. They discuss the effects of mutations in ‘outlying’ amino acids. It is unclear what these are referring to. Perhaps consider making a

AUTHORS’ RESPONSE: The comment appears to be truncated, so our apologies that we cannot see the reviewer’s recommendation. Our use of “outlying” in indeed somewhat ambiguous. In fact our point is rather simple, namely that although use of the Wilcoxon signed rank test as described in the figure legend to test Site 2 mutants against WT produces only a modest p value, a significant subset of variants do show a much more severe effect. We have refined wording and added some more annotation as follows:

Mutation of site 2 residues only mildly reduced binding and signalling in group-based analysis, but with much more deleterious effects **for a small subset of amino acids** (Figure 4C). **Loss of binding/signalling was explained for most of these variants** by reduced cell surface expression (Supplementary Figure 5D), **although mutation of 529 and 530 reduced binding but not expression.** The effect of mutating site 2 mutations in aggregate was thus much smaller than for site 1 mutations (Figure 4D). Well-expressed variants with the largest loss of signalling affect residues clustered around site 1, with others in the “legs” of the receptor involved in unliganded receptor compaction (Figure 4E).

L242 The authors say around 5,000 individual variants reduced signaling. In general I think it would be useful to include specific numbers (throughout the manuscript) but also put this into context of the number of variants that have scores. Finally, what is reduced signaling? What was the cutoff that was used to make this call, and what was its rationale?

AUTHORS' RESPONSE: Specific numbers have now been added to text as follows:

5,358 individual variants from 14,544 with scores (37%) reduced signalling (Figure 4A,F; Tables S2,S3).

Similar edits have been made where the number of variants reducing expression and binding are discussed. As discussed in methods, we define reduced signalling simply as a score <0 and an FDR <0.05. Different cut offs for diagnostic use developed from consideration of known pathogenic variants are later suggested (Extended data figure 5A).

L244: The authors mention lower confidence score - what confidence is this?

AUTHORS' RESPONSE: We simply meant FDR <0.05 and have amended the corresponding sentence:

Nine of the remaining variants **had scores that were not significantly different to WT (FDR ≥0.05) (Figure 4F).**

L248: Authors mentioned T221 emerging as a residue of interest based on the average positional score then say but it's only based on 4 mutations. It's unclear what I am supposed to learn from this. In addition, it would be good to be a bit more specific about what average positional score the authors are referring to. Additionally, in the insets the score is referred to the median positional scores not the mean positional score.

AUTHORS' RESPONSE: T221 is called out simply as it is an outlier in Extended Figure 4A (now Supplementary Figure 8A). With only 4 scores it is difficult to evaluate how important this is, but we thought that reviewers may see it and want to know more. We now try to be more precise regarding the average positional score as below, and finally have clarified at all relevant points that we indeed mean median scores.

Threonine 221 emerged as a residue of potential interest based on the median score for all codon 221 variants (Supplementary Figure 8A), but this was based on only 4 variant scores.

L299: The authors compare the effects of partial agonist antibodies to insulin at 10nM. It is challenging to directly compare one compound to another because the effective concentration may be different. Why was this concentration chosen? Why is it an order magnitude less than insulin?

AUTHORS' RESPONSE: It is well established from our own work and that of many others that 100nM insulin produces maximal binding to and signalling from WT INSR, and is moreover at the very highest reported range of plasma insulin even in rare infants with effectively no functional insulin receptor. 10nM mAb was used based on evidence that this concentration also elicits the maximal mAb response at WT receptors: *Siddle et al Biochem Soc Trans* 15:47–51; *O'Brien et al. EMBO J* 6:4003–4010). For many but not all mutant

receptors that we have studied previously 10 nM mAb also solicited the greatest response which then plateaued. Precise comparison of mAb vs insulin potency will require future dedicated study for variants of interest.

Figures—The figure labels are small, making interpreting them hard. The figure labels in the main figures should be more consistent.

AUTHORS' RESPONSE: We have endeavoured to improve clarity throughout, increasing font sizes where possible and harmonising styles among other alterations.

Figure 6: I would have loved to see some of this data mapped on the structures. I had a really hard time interpreting the data here which is a shame as I think it's the most interesting experiment done in the manuscript.

AUTHORS' RESPONSE: We now include a structure in new **Figure 7** showing Selective Sensitivity Scores computed as below projected onto the four insulin-bound receptor. We hope the reviewer agrees that this, among the wider remodelling of the final figures and corresponding text described below, has improved clarity.

Figures 6B-D: The authors show the median score per position across the different signaling datasets. However, for most of the comparisons, they discuss individual variants. I think it would be worthwhile to show the variant distributions.

AUTHORS' RESPONSE: This is a reasonable suggestion. In new Figure 6 we show scatterplots for both position median scores and individual variant scores, limiting points plotted to those with FDR <0.01 to denoise. Selected variants have also been marked up.

Figure 6E—It's unclear why this panel is not included in an earlier figure, as this figure's primary focus is the partial agonist antibodies. It seems this is here mostly to explain figures 6F-G (as described below I found this extremely confusing). Figures 6F-G: The authors show the effects of an antibody signaling-insulin signaling. The y-axis denotes this as (average mab signaling)-(insulin signaling). But I am a bit confused by what is being calculated - is this the average across replicates? Is there no average in the insulin score? In the corresponding figure legend it says that only certain variants were included. I commend the authors in trying to be rigorous but I ended up very confused. I encourage them to include a section on the methods on this section of the manuscript.

AUTHORS' RESPONSE: Our graphics were complex and we accept on review that we neither illustrated nor explained our key findings clearly enough. We have addressed these and other points related to Figure 6 with an extensive redraft and inclusion of a further figure 7. We offer a summary and a rationale for our changes, with some additional clarification to respond to comments that are still relevant in the light of the new format:

Because of the translational relevance of this final analysis we increased its stringency by employing additional filters on top of those applied in previous figures. Specifically, we plotted only variants for which:

1. Signalling scores were generated by all 5 insulin signalling replicates AND all 3 mAb signalling replicates
2. Insulin signalling scores were in the lowest 4 of 5 equally spaced bins dividing the loss-of-function scores (corresponding to <0.204)

These stringent filters ensured that we addressed only variants with the most robust evidence for impaired insulin signalling, and the highest confidence scores. We then computed the difference between the Ab signalling score and the insulin signalling score for each variant to generate a new score which we refer to as the “Selective mAb Sensitivity Score”. It is this score which we plot on Manhattan plots with the additional mark ups described.

Rather than adding these details to Methods as suggested, we felt they were sufficiently important to figure interpretation that we preferred to include details in main results text as follows:

Responsiveness of mutated receptors to antibody partial agonists

The translational aim of this study extended beyond addressing the diagnostic “VUS problem”. We also sought to identify variants showing severely impaired insulin responsiveness but that respond to stimulation by the monoclonal antibodies tested. Our reasoning was that that patients with extreme IR harbouring at least one such variant may benefit from future antibody-based therapy, as previously suggested¹⁴⁻¹⁷. We tested this by assessing responses of the whole library to 10 nM antibodies *in lieu* of insulin. Both antibodies elicited a pattern of signalling overlapping heavily with that of insulin (Figure 6A-D), however key areas of difference were apparent on the variant effect maps, including but not limited to the α CT region noted previously.

To focus only on the variants with the strongest evidence for impaired signalling we selected variants with scores in all five insulin signalling assay replicates and with median score in the lowest 80% of the observed loss-of-function range. We further filtered these variants to include only those with scores in all three antibody signalling assay replicates. For each of these robust loss-of-insulin-signalling variants we then calculated the difference between antibody and insulin-stimulated signalling scores, which we call the “selective sensitivity score”. These selective sensitivity scores were then represented as Manhattan plots for each antibody. Finally, areas identified on variant effect maps as showing selective activation by antibodies were boxed for reference, and the mean sensitivity scores for sliding windows of 15 residues were plotted (Figure 6F,G).

Figure 6H: The authors have a Venn diagram comparing the two mAb partial agonists but not insulin. It would be worthwhile having the Venn diagram also include insulin. Instead, I think this is based on the data from the Manhattan plots which is mab-insulin.

AUTHORS’ RESPONSE: This Venn diagram did not simply show the variants that respond to mAbs, but, more specifically, those that **A.** show severely impaired insulin responsiveness AND **B.** respond better to the mAb than insulin. In other words insulin responsiveness is implicitly factored into the data shown. It is clear that we did not explain this well enough in the original manuscript. We have now replaced the Venn diagram with the new scatterplot below which we think is more informative than the Venn diagram, as it illustrates effect sizes as well as binary responsiveness:

We amend corresponding *Results* text as follows:

Areas identified as showing selective activation by antibodies are boxed for reference, and the mean sensitivity scores for sliding windows of 15 residues are also plotted. D734A, an antibody-responsive variants that was the focus of previous proof-of-concept studies, was confirmed to signal on antibody exposure, but was far from the most antibody-sensitive variant (Figure 7A,B). Selective sensitivity scores for all variants with positive scores for both antibodies tested are plotted in Figure 7D, showing strong correlation and identifying variants with the most robust responses to both antibodies. The list of variants generated in this analysis offers good candidates for future translational studies of humanised versions of the antibodies studied (Supplementary Table 6).

Finally, we have augmented and marked up **Supplementary Table 6** so that it is more informative, now computing selective sensitivity scores for mAbs over insulin, additionally expressing the mAb signalling response as a % of the insulin signalling deficit.

We respond to reviewer 1's residual comments below:

Responses to reviewers' comments

Reviewer #1

The authors have on the whole addressed the concerns I raised in my first review, and the manuscript is now largely suitable for publication. I will leave assessment of the authors' response to the other reviewers to the latter's and the Editor's discretion.

- *However, there remain a few (minor) points which need to be addressed by the authors or which I recommend be subject to editorial discretion. These are detailed below, wherein I copy my original query and then state my response to that of the authors.*
- *Responses by the authors to my remaining queries are acceptable and are not discussed further herein. I do not need to see a further corrected manuscript.*

Original Query by Reviewer #1:

Extended Data Figure 2: the panel captions do not match the panels themselves, and likewise the references to panels in this figure in the main text appear incorrect. This makes it difficult to disentangle, and there is likely a panel missing from this Figure.

Reviewer's comment on Authors' response:

The authors' corrections are adequate, except that Panel E is now labelled "F" in the caption, this needs to be corrected.

AUTHORS' RESPONSE: Thanks for pointing out this residual oversight. Now corrected.

Original Query by Reviewer #1:

I am also concerned about the authors' use of incredibly small font in some of the Figures, there may be editorial requirements in this regard.

Reviewer's comment on Authors' response:

The authors have improved the font size. I will leave it to editorial discretion as to whether further improvements are needed or can be achieved.

AUTHORS' RESPONSE: We note the comment and await editorial guidance, if any.

Original Query by Reviewer #1:

The authors do not discuss the impact of variants on receptor glycosylation. Obviously, mutation of residues involved in N- and O-linked glycosylation (either the residues that form the linkage to the sugar or those that form part of the associated motif) will remove glycosylation at the relevant site. It would be of interest to see whether, for example, mutation of the asparagine residues involved in N-linkage of sugars has impact on folding and/or insulin binding and whether such impact concurs with literature in this regard.

Reviewer's comment on Authors' response:

The authors have addressed this question in detail in their amended Results and are to be commended for doing so. For completeness, I recommend inclusion in the final manuscript of the four additional panels that they offer as additional Extended Data Figures / Panels, together with appropriate cross-referencing of these panels in the main text.

AUTHORS' RESPONSE: The suggested plots have now been added as (new) Supplementary Figure 6 with due cross referencing

Original Query by Reviewer #1:

L81. The authors should also reference the class of receptor activating peptides pioneered by Schäffer et al. (2003, <https://doi.org/10.1073/pnas.0830026100>), which have also more recently been subject to structural studies to elucidate their mode of action. The latter could also be referenced at L365 (p15).

Reviewer's comment on Authors' response:

The authors add an appropriate reference at original line L365 to the Schäffer manuscript (authors' reference #13 in the current draft). However, the inclusion of the reference at original L81 (now line L83) does not make sense, as it suggests that the Schäffer study refers to activating antibodies, whereas it deals with insulin mimetic peptides. This needs to be corrected.

AUTHORS' RESPONSE: The reviewer is correct to spot this. The first, inaccurate, instance of this citation is now removed while retaining the later appropriate citation.

Original Query by Reviewer #1:

Extended Data Figure 1A. There is something amiss in coloring of the schematic on the right-hand side of this panel. In particular, the two FnIII-2 domains (in the middle of the structure, second domains up from the bottom) show a multitude of colors and not the brown color indicated on the left-hand side. There are also "brown" strands within the yellow FnIII-1 domains. Furthermore, there is no indication in the panel or in its caption that the four insulins are colored green. I recommend that the panel be redone in clearer and more contrasting colors. Also, in the left-hand component of the panel, the authors should label as such the respective α - and β -chains, to assist subsequent interpretation of Fig. 2A, as well as label the chain components as ID α , ID β FnIII-2 α and FnIII-2 β for consistency with the text

Reviewer's comment on Authors' response:

The Figure is improved. However, the label "L2" appears detached above the right-hand side part of Panel B, I suspect that it is intended to be part of the left-hand part of Panel B, this needs to be corrected. Also, in Panel A, I require explicit inclusion of segments α CT and α CT', colored as in Panel B.

AUTHORS' RESPONSE: The reviewer is correct regarding the "L2" label - now resited to the correct left hand panel. We have also now labelled and recoloured the alphaCT segments in Panel A as requested.

Original Query by Reviewer #1:

L158. The authors should consider including a Supplementary Table that lists the 80 residues they considered as part of Site 1, and to remove any doubt be clear that the 80 refers to 80 for each of site 1 and site 1' (if that is indeed what they mean).

Reviewer's comment on Authors' response:

The authors have now included such a Table. I request that the table include in addition to the residue numbers their respective three-letter amino acid codes to help prevent any confusion.

AUTHORS' RESPONSE: Amino acid codes have now been added as requested to the relevant supplementary table.

Original Query by Reviewer #1:

L251-L254. A number of the residue location descriptions here are wrong. For example, I see no “remodelling” of Y839 upon overlay of PDB structures 4ZXB (apo IR ectodomain) and 6PXV (four-insulin-bound IR ectodomain); note that Y839 is equivalent to Y800 in the PDB files as the latter number from the start of the mature protein and are based on IR-A, the exon 11 minus isoform. Likewise, the authors state that F862 is “not resolved”: this residue is equivalent to F823 in the above PDB files and it is indeed present in both, i.e., not “not resolved”. It is also present in PDB file 8U4B (the IR-B apo ectodomain structure). The authors should therefore check (and correct if necessary) all of their residue locations claims throughout the manuscript.

Reviewer’s comment on Authors’ response:

The authors now have the following text:

Threonine 221 emerged as a residue of interest based on the average positional score (Supplementary Figure 8), but this was based on only 4 variant scores in the MAVE. Other variants warranting future assessment based on selectively increased signalling include residues forming part of insulin binding site 1 (L2 domain: E343I, N376W; FnIII-1: N568C, P576C), residues involved in interdomain interactions that change on insulin binding (e.g. CR domain: C293F; FnIII-2a: R661W; FnIII-2b: Y839V; Extended Data Figure 4A-G), and residues not resolved in available structures (e.g. FnIII-3: F862Y).

I note in the last part of the last line above that F862 is still claimed to be “not resolved”, even though the authors now agree that it is “resolved”. This needs to be fixed.

AUTHORS’ RESPONSE: Our apologies. This change had been made but was lost by a mistake in version control in editing. The erroneous comment about F862Y here is now removed.

In terms of Y839, I agree that we had a different interpretation of their word “remodelling”, but I recommend as a fix that the authors use the more informative word “re-arrange” as opposed to the non-descript word “change” in the above text.

AUTHORS’ RESPONSE: We are happy to use “re-arrange” instead, and have made this change (although in truth we now appear to be in a rather grey world of semantics)